



# Review Article: Global Monitoring of Snow Water Equivalent using High Frequency Radar Remote Sensing

Leung Tsang[1], Michael Durand[2], Chris Derksen[3], Ana P. Barros[4], Do-Hyuk Kang[5], Hans Lievens[6], Hans-Peter Marshall[7], Jiyue Zhu[1], Joel Johnson[8], Joshua King[3], Juha Lemmetyinen[9], Melody Sandells[10], Nick Rutter[10], Paul Siqueira[11], Anne Nolin[12], Batu Osmanoglu[13], Carrie Vuyovich[13], Edward Kim[13], Drew Taylor[14], Ioanna Merkouriadi[9], Ludovic Brucker[13], Mahdi Navari[13], Marie Dumont[15], Richard Kelly[16], Rhae Sung Kim[13], Tien-Hao Liao[17], Xiaolan Xu[17]

[1] Department of Electrical Engineering and Computer Science, University of Michigan, Ann Arbor, MI, 48109, USA
[2] School of Earth Sciences & Byrd Polar and Climate Research Center, The Ohio State University, Columbus, OH, 43210, USA
[3] Climate Research Division, Environment and Climate Change Canada, Toronto, Canada
[4] Civil and Environmental Engineering University of Illinois at Urbana-Champaign, Urbana, IL, USA
[5] ESSIC, University of Maryland, College Park, MD, 20740, USA.
[6] Division of Soil and Water Management, KU Leuven, Leuven, Belgium
[7] Department of Geoscience, Boise State University, Boise, Idaho, USA
[8] Department of Electrical and Computer Engineering, Ohio State University, Columbus, OH 43212 USA
[9] Arctic Research Centre, Finnish Meteorological Institute, Helsinki, Finland
[10] Geography and Environmental Sciences, Northumbria University, Newcastle, UK
[11] Electrical and Computer Engineering, University of Massachusetts, Amherst, MA, USA
[12] Department of Geography, University of Nevada-Reno, Reno, NV, USA
[13] NASA Goddard Space Flight Center, Greenbelt, MD, USA
[14] Remote Sensing Center, University of Alabama, Tuscaloosa, AL, USA
[15] Centre d'Etudes de la Neige, Meteo-France, Grenoble, France
[16] Department of Geography and Environmental Management, University of Waterloo, Waterloo, Canada
[17] NASA Jet Propulsion Laboratory, Pasadena, CA, USA

*Correspondence to*: Leung Tsang (leutang@umich.edu)

**Abstract.** Seasonal snow cover is the largest single component of the cryosphere in areal extent, covering an average of 46 million square km of Earth's surface (31% of the land area) each year, and is thus an important expression of and driver of the Earth's climate. In recent years, Northern Hemisphere spring snow cover has been declining at about the same rate (~ -13%/decade) as Arctic summer sea ice. More than one-sixth of the world's population relies on seasonal snowpack and glaciers for a water supply that is likely to decrease this century. Snow is also a critical component of Earth's cold regions' ecosystems, in which wildlife, vegetation, and snow are strongly interconnected. Snow water equivalent (SWE) describes the quantity of snow stored on the land surface and is of fundamental importance to water, energy, and geochemical cycles. Quality global SWE estimates are lacking. Given the vast seasonal extent combined with the spatially variable nature of snow distribution at regional and local scales, surface observations will not be able to provide sufficient SWE information. Satellite observations presently cannot provide SWE information at the spatial and temporal resolutions required to address science and high socio-





economic value applications such as water resource management and streamflow forecasting. In this paper, we review the potential contribution of X- and Ku-Band Synthetic Aperture Radar (SAR) for global monitoring of SWE. We describe radar

interactions with snow-covered landscapes, characterization of snowpack properties using radar measurements, and refinement of retrieval algorithms via synergy with other microwave remote sensing approaches. SAR can image the surface during both day and night regardless of cloud cover, allowing high-frequency revisit at high spatial resolution as demonstrated by missions such as Sentinel-1. The physical basis for estimating SWE from X- and Ku-band radar measurements at local scales is volume scattering by millimetre-scale snow grains. Inference of global snow properties from SAR requires an interdisciplinary

approach based on field observations of snow microstructure, physical snow modelling, electromagnetic theory, and retrieval strategies over a range of scales. New field measurement capabilities have enabled significant advances in understanding snow microstructure such as grain size, densities, and layering. We describe radar interactions with snow-covered landscapes, the characterization of snowpack properties using radar measurements, and the refinement of retrieval algorithms via synergy with other microwave remote sensing approaches. This review serves to inform the broader snow research, monitoring, and

applications communities on progress made in recent decades, and sets the stage for a new era in SWE remote-sensing from SAR measurements.

## 1 Introduction

Seasonal snow on land is responsible for a number of important processes and feedbacks that affect the global climate system, freshwater availability to billions of people, biogeochemical activity including exchanges of carbon dioxide and trace gases,

and ecosystem services. Despite this importance, snow mass (commonly expressed as the 'snow water equivalent' or SWE) is a poorly observed component of the global water cycle. Given the vast area of northern hemisphere snow extent (exceeding $45 \times 10^6$ km$^2$ each winter), surface observing networks are insufficient as a sole source of information for snow monitoring. Satellite remote sensing is the only means to monitor SWE consistently and continuously at continental scales. Optical satellite imagery acquired under cloud-free conditions can provide information on where and when snow is on the ground, but does

not support the retrieval of SWE. Long time series of snow mass information are available from satellite passive microwave measurements (Luojus et al., 2021), but at coarse spatial resolution (gridded at 25 km spatial resolution), mountain areas across which high values of SWE occur are excluded, and bias correction is required under deep snow conditions (>150 mm SWE; Pulliainen et al., 2020). Land surface models driven by meteorology from atmospheric reanalysis can produce hemispheric-scale SWE information at coarse spatial resolutions (e.g. Kim et al., 2021), but there is a large spread between products due to

differences in the meteorological forcing data (especially precipitation) and a pronounced negative bias in mountain areas (Wrzesien et al., 2019a; Cao and Barros, 2020; Lundquist et al., 2019). Differential airborne and ground-based lidar altimetry (Deems et al., 2013; Meyer et al., 2021) and spaceborne stereo photogrammetry (Deschamps-Berger et al., 2020) can provide snow depth information at high resolution by differencing repeat digital elevation models but are limited to small spatial domains and sparse temporal sampling. C-band radar has recently been applied to retrieve snow depth in mountainous regions

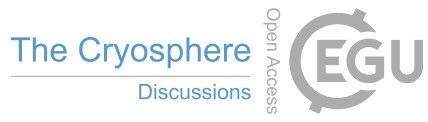

(Lievens et al., 2019) using empirical relationships derived from ground-based measurements; however, this approach is not demonstrated for the comparatively shallow snowpack found across large regions of the northern hemisphere. Airborne and tower-based measurements have also identified the possibility of retrieving snow parameters from L-band interferometric SAR (Deeb et al., 2011), P-band signals of opportunity (Shah et al., 2017; Yueh et al., 2021), wideband auto-correlation radiometry (Mousavi et al., 2019), and FM-CW radar (Yan et al., 2017).

Ku- and X-band radar measurements, in contrast, provide a viable pathway to produce SWE information at the temporal and spatial scales necessary to advance operational environmental prediction, climate monitoring, and water resource management across the northern hemisphere (see Section 2 for an overview of the scientific requirements for snow mass information). Significant progress was made over the past decade in understanding the Ku-band and X-band radar response to variations in SWE, snow microstructure, and snow wet/dry state. The ESA Cold Regions Hydrology High-Resolution Observatory

(CoReH2O) mission (dual-frequency X- and Ku-band; completed Phase A at ESA in 2013; Rott et al., 2010) was a major impetus. Previously, although the potential for Ku-band radar was also explored at NASA as part of the Snow and Cold Land Processes Mission and supporting Cold Land Processes Experiment (Yueh et al., 2009). Experimental tower and airborne measurements have been used to advance understanding of the physics of backscatter response to snow microstructure and SWE (Lemmetyinen et al., 2018; King et al., 2018), including the complicating effects of forest cover (Montmoli et al., 2016;

Cohen et al., 2015). Innovative new field measurement of snow microstructure parameters (Löwe et al., 2013; Kinar and Pomeroy 2015) now provide the quantitative observational basis for radar modelling of layered snowpacks (Tsang et al., 2018) and radar retrieval algorithms (Zhu et al., 2018), which have both advanced dramatically in the past decade.

The purpose of this review is to summarize the status of all the components necessary to fully develop the pathway towards a potential future radar mission focused on seasonal snow mass. This includes the theoretical sensitivity to SWE via volume

scattering processes, the influence of surface and ground contributions, and approaches to SWE retrieval as supported by physical snow and radiative transfer modelling. Results from previous ground, tower, and airborne measurement campaigns are also reviewed. Ku-band SAR measurements are limited to snow depth of about 1 meter because of saturation of radar volume scattering. Thus, synergism with C-band Sentinel 1 data for snow depths beyond 1 meter (Lievens et al., 2019) is also explored. Other synergies with interferometric SAR, radar tomography, passive microwave, and L- and C- band SAR

measurements are also described to provide ancillary information and to improve retrieval performance.

## 2 Scientific Objectives of Global Remote Sensing of SWE and Spatial and Temporal Requirements

High priority science objectives require snow mass information at moderate spatial resolution (250-500m) and frequent revisit (~3-5 days), a measurement paradigm that is currently not available. As outlined below, these science requirements support applications related to climate services and operational environmental prediction including quantifying snow mass

contributions to water, energy, and geochemical cycles, better prediction of spring flooding, shallow landslide activity, and adaptation of cold-regions water resources to climate change.





1. Inventory how much water is stored as seasonal snow, and how it varies in space and time.

The amount, distribution, and variability of terrestrial SWE across the northern hemisphere is poorly quantified because surface networks are inadequate, and existing gridded SWE datasets have divergent climatologies (Wrzesien et al., 2019a) and anomalies (Mudryk et al., 2015). Alpine regions are particularly problematic because the course spatial resolution of existing products (typically 25 km grid spacing or more, with some new analyses available at 9 km) is incompatible with the scale of SWE variability (100's of metres). SWE estimates derived from models at continental scale are subject to uncertainties in both meteorologic forcing data and model parameterizations (e.g. Kim et al., 2021). SWE is highly sensitive to changing temperature and precipitation in a warming climate; confident projections of resultant changes are uncertain because we lack baseline SWE estimates. SWE can change rapidly from day to day and across local areas due to the influence of individual weather events, but we currently do not have any means to track these changes with sufficient spatial or temporal resolution. The lack of a baseline snow inventory negatively impacts many aspects of hydrological resource management. With projections of continued climate warming and shifts to snow cover resources (including precipitation phase changes and timing of spring melt), addressing this capability is more pressing than ever.

2. Properly initialize snow in environmental prediction systems including numerical weather prediction (NWP) and streamflow forecasting.

Land surface data assimilation is an important component of state-of-the-art environmental prediction systems, which provide initial land surface conditions (such as snow conditions, soil moisture and temperature) for numerical weather prediction and other forecasting systems. Satellite data from the SMOS and SMAP missions are presently assimilated to improve the characterization of soil moisture (e.g. Carrera et al., 2019). Parallel activities have not been sustained for seasonal snow because assimilation of existing satellite measurements does not sufficiently improve land surface model performance (de Lannoy et al., 2010). Addressing this gap is important because evidence shows that a more realistic initialization of SWE can improve streamflow forecasts, especially during extreme events (Vionnet et al., 2020) and at lead times greater than 2 weeks (Abaza et al., 2020; Wood et al., 2016). The current inability to plan and respond to snow-related runoff events is costly: if effectively managed, runoff from snow melt has a global economic value in the trillions of dollars (Sturm et al., 2017), but also poses a risk through loss and damage associated with flood events. For example, the devastating floods in the Canadian Rockies, foothills and downstream areas of southern Alberta and south eastern British Columbia during June 2013 provide a compelling case for the impact of improved snow information on hydrological modelling of costly extreme events (Pomeroy et al, 2016). Additionally, high-resolution satellite-derived snow distributions could support development of improved downscaling techniques for existing coarsely gridded products (Manickam and Barros, 2020).

3. Validate and support improvement of the representations of snow processes and feedbacks in regional and global climate models.

Gridded SWE datasets are required for the verification of models used for seasonal prediction (e.g. Sospedra-Alfonso and Merryfield, 2017), and the validation of historical climate model simulations which form the basis of climate projections (e.g.





Mudryk et al., 2020). Earth observation-derived products make a small contribution to the current suite of available gridded SWE products: reanalysis and snow models form the primary basis for the evaluation of seasonal prediction and coupled climate model simulations. The first assessment of CMIP6 model simulations by Mudryk et al. (2020) identified two key findings: (1) excessive snow mass at the hemispheric scale is a feature of CMIP6 models, and (2) nearly all models increase

snow extent too slowly during the accumulation season and decrease snow extent too slowly during the snowmelt period. These findings would be strengthened through the support of appropriate moderate resolution satellite SWE datasets. Furthermore, more detailed analysis at the grid point scale is needed to effectively link the model parameterizations of the (usually diagnosed) snow cover fraction to the prognostic snow mass. Prediction between the snow-on and snow-free conditions is especially challenging due to repeat cycles of melting and refreezing that alter the vertical microstructure of the

snowpack and lateral SWE redistribution.

    4.   Address the role of snow properties across high latitudes in influencing terrestrial carbon cycling, trace-gas exchanges, and permafrost.

Snow is an important insulator of the underlying soil, influencing the thermal regime and corresponding carbon fluxes in winter

(Natali et al., 2019). Permafrost is warming across the northern hemisphere (Biskaborn et al., 2019) with implications on vegetation, surface hydrology, landscapes, and the carbon cycle. Addressing the drivers of these changes requires a sound understanding of the role of seasonal snow, but current snow mass datasets do not meet the requirements of state-of-the-art permafrost models (Obu et al., 2019) which provide continental scale estimates of permafrost extent, thermal state, and active layer thickness.

**3 Radar interaction with snow covered landscapes**

**3.1 Theoretical descriptions of radar-landscape interactions**

In this section, we describe volume scattering from a snowpack, rough surface scattering from the snow-soil interface, and the attenuation of radar waves by forest canopies.

**3.1.1 Interaction of radar waves with snowpack by the Radiative Transfer Model (RTM)**

Microwave signals emitted from a SAR system are scattered by the mm-scale ice grains that make up the snow and at boundaries between snowpack layers with different dielectric properties. The SAR system measures the portion of the signal returned to the sensor (i.e. the backscatter). Because volume scattering increases with snow mass, measurement of backscatter allows estimation of snow mass. Structural changes in snow that impact snow backscatter are densification and metamorphism that introduce vertical heterogeneity in snow grain sizes and snow density, and thus impact snow depth and SWE.

Historically, the first model developed for microwave scattering of snow was by Chang et al. (1976), which assumed Mie scattering from a collection of ice spheres in a single layer to solve a radiative transfer equation for passive remote sensing





applications. The first active remote sensing model (Zuniga et al, 1979) used the Born approximation with snow represented by a random medium characterized by a correlation function and associated correlation length. Since these early models, a variety of radiative-transfer microwave scattering models have been developed with representations of (i) snow microstructure,
(ii) absorption coefficient, (iii) effective permittivity, (iv) scattering phase matrices, and (v) layering effects. Analytical and numerical solution methods are used to solve these equations (Tsang et al. (1985), Ulaby et al. (1986), and Fung et al. (2010)). A historical review of different models is given in Shi et al., (2016). Rapid progress has been made recently due to the advancement of high-performance parallel computations and efficient computation methods for characterizing the complex microstructure of snow and full wave solutions of Maxwell's equations (Ding et al., 2010; Xu et al., 2012; Tan et al., 2017;
Tsang et al., 2018).

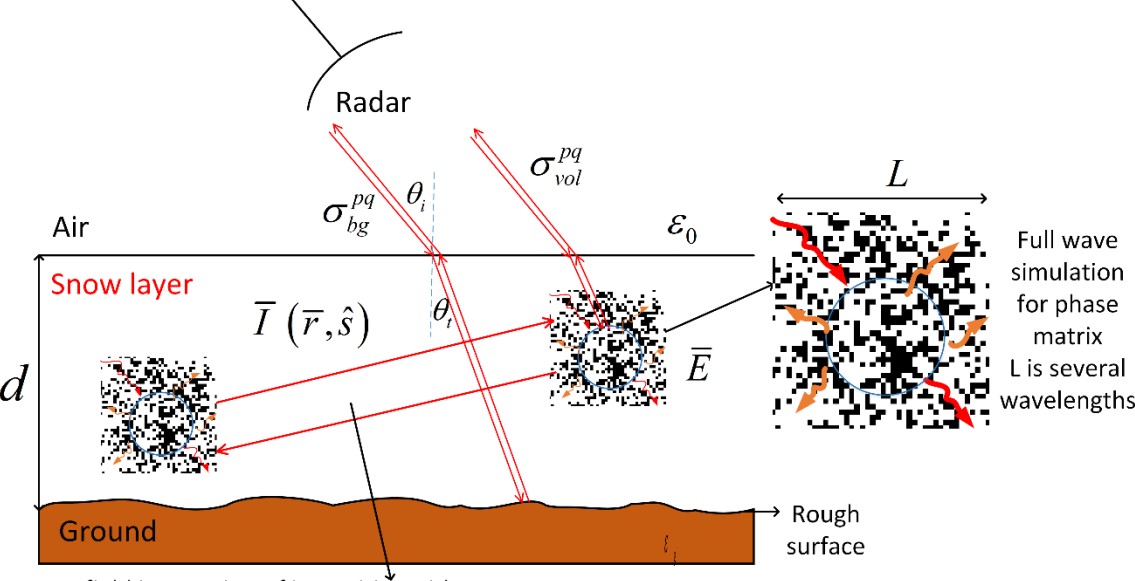

**Figure 1**: **Main contributions to radar scattering from snow covered ground. Scattering at air/snow interface is neglected, and the snow layer is assumed homogeneous. The snow depth is $d$. $\varepsilon_0$ is the air permittivity and $\varepsilon_g$ is the soil permittivity. $\theta_i$ is the incident angle of radar. $\overline{E}(\overline{r})$ is the internal electrical field and $\overline{I}(\overline{r}, \hat{s})$ is the specific intensity within the snowpack.**

Consider an incident wave from the radar at an incident angle $\theta_i$ A simple physical model represents total backscatter $\sigma_{tot}^{pq}$ from the snowpack over the ground arising from two contributions as shown in Figure 1: i) the volume scattering component $\sigma_{vol}^{pq}$ from the snowpack and ii) the rough surface scattering $\sigma_{bg}^{pq}$ from the underlying soil. The expression is:

$$\sigma_{tot}^{pq} = \sigma_{vol}^{pq} + \sigma_{bg}^{pq} exp(-2\tau sec\theta_t) + \sigma_{surf}^{pq} \quad (1)$$

where p and q refer to the polarization state e.g. $\sigma_{tot}^{HV}$ represents backscatter emitted at vertical polarization and received at
horizontal polarization. Scattering from the underlying rough soil surface is attenuated by the snow layer, represented by the two-way attenuation factor of $exp(-2\tau sec\theta_t)$. The quantity $\tau$ is the optical thickness of the snowpack and $\theta_t$ is the refraction





angle in snow which is related to $\theta_i$ by Snell's law  Because of the low permittivity contrast between air and snow, the scattering ($\sigma_{surf}^{pq}$) from the air/snow interface (Rott et al., 2010)  can be neglected except for wet snow . Rough surface scattering is considered in Section 3.1.2.

Multiple scattering effects within snow can be calculated by the dense medium radiative transfer equation (DMRT). Here we describe DMRT for illustration of the retrieval technique, but this could equally be applied to other electromagnetic theories. The DMRT is a partially coherent model having coherent and incoherent parts. The incoherent interactions are based on the classical radiative transfer equation (RTE):

$$\frac{d\bar{I}(\bar{r},\hat{s})}{ds} = -\kappa_e \bar{I}(\bar{r},\hat{s}) + \int_{4\pi} d\Omega' \bar{\bar{P}}(\bar{r},\hat{s},\,\hat{s}')\bar{I}(\bar{r},\hat{s}') \quad (2)$$

where $\bar{I}(\bar{r},\hat{s})$ is the specific intensity at the position $\bar{r}$ in direction $\hat{s}$, $\bar{\bar{P}}(\bar{r},\hat{s},\,\hat{s}')$ is the phase matrix, and the extinction coefficient $\kappa_e$ is the sum of the scattering and absorption coefficients, $\kappa_e = \kappa_s + \kappa_a$. The distinctive difference of DMRT from classical RTM is the coherent part in which the extinction coefficients $\kappa_e$ and the phase matrix $\bar{\bar{P}}(\bar{r},\hat{s},\,\hat{s}')$ are obtained by solutions of Maxwell equations including coherent wave interactions among the ice grains that are in the near field and intermediate fields distance ranges from each other (Liang et al., 2008; Ding et al., 2010). In solving Maxwell's equations, the

dense medium effects and snow microstructure are accounted for. Both coherent and incoherent wave interactions are included.

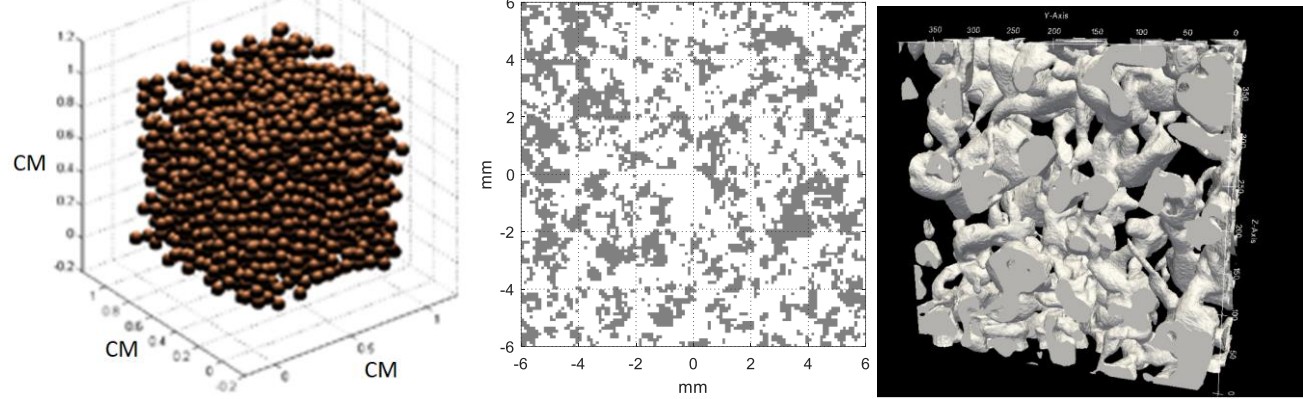

Figure 2: Microstructural descriptions of the snowpack: (a) sticky hard sphere theoretical model, (b) a bi-continuous medium with ice crystals in dark and air in white based on parameters ⟨ζ⟩= 1 mm, and *b*=1.0 and (c) 3-D image from x-ray microtomography with ice crystals shown in white and air voids in black.

The volume scattering $\sigma_{vol}^{pq}$ depends on snow microstructure; the response of microwave radiation to snow microstructure has been represented in at least five different ways in electromagnetic models. 1) In Chang et al., (1976), collections of spheres are used for the microstructure, and the scattering by individual spheres is added incoherently. However, it is not appropriate for snow as particles are densely packed, meaning electromagnetic (EM) waves scattered from individual grains interact coherently within distance scales of several wavelengths. 2) DMRT has been applied to the cases of hard spheres (Tsang et al.,

1985), sticky hard spheres (Tsang et al., 2007) and distributions of sphere sizes (Tsang et al., 1992). The coherent interactions are described analytically by the quasi-crystalline approximation (QCA) of Mie scattering for closely  packed spheres. Figure

2 (a) gives a visual representation of the sticky hard sphere microstructure. 3) The approach of Hallikainen et al. (1987) empirically relates grain size directly to scattering coefficient. In this case, the model was derived from experimental observations of extinction behaviour and the relations to traditional grain size measurements. The direct connection between

scattering and grain size means the model is simpler to apply, albeit with potentially large errors due to limited observations and varies with snow types. 4) A different representation of snow is a random medium of ice and air (Figure 2 (b)). Mätzler (1998) treats scattering by characterizing the microstructure autocorrelation length, using the improved Born approximation and the random medium assumption. 5) The more generalized bicontinuous medium approach uses two parameters to characterize snow microstructure: a mean grain size $\langle\zeta\rangle$ and an aggregation parameter $b$. The aggregation parameter $b$

represents the adherence of ice grains together to form clusters. Smaller $b$ parameter values produce greater aggregation. The $b$ parameters chosen for X- to Ku-bands fall in the range of 1.0 to 2.0 (Chang et al., 2014; Tan et al., 2015; Xiong & Shi, 2019). With development of computational EM, numerical solutions of Maxwell's Equations in 3D are also used (Ding et al., 2010; Xu et al., 2012; Tan et al., 2017).

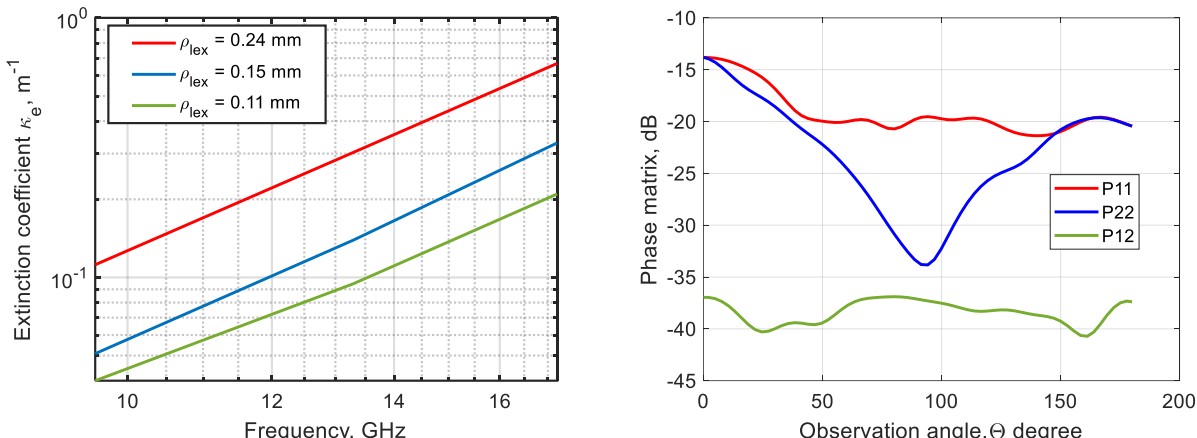

**Figure 3**: **(a) Frequency dependence of the extinction coefficient from X- to Ku-band and (b) phase matrices at 13.3 GHz for different values of microstructure correlation length $\rho_{lex}$. Snow parameters are volume fraction $f_v = 0.20$, aggregation parameter $b = 1.2$, and correlation length 0.15 mm. P11 and P22 are co-polarized phase matrix elements and P12 is a cross-polarized phase matrix element.**

EM models thus have evolved in part due to knowledge advances from our improved ability to measure snow microstructure.

Stereological approaches led to advances in treating snow as a random medium using correlation functions (Wiesmann et al., 1998). X-ray micro-computed tomography (μ-CT) has emerged to image the 3-dimensional structure of snow (Kerbrat et al. 2008), as illustrated in Figure 2 (c), which has fed advances such as the dual active/passive Snow Microwave Radiative Transfer (SMRT) model (Picard et al., 2018). SMRT was developed to understand how to represent microstructure faithfully at scales relevant for microwave scattering and has the potential to allow direct use of correlation functions from μ-CT.

Application of μ-CT-derived microstructure parameters in SMRT removes the need for empirical grain scale factors with frequency-dependent model performance governed by the quality of microstructure model fit (Sandells et al, 2021). EM models



have been adapted to work with field-derived measurements as well. Field methods to measure microstructure are more fully discussed in section 3.3.1. These advances reduce the uncertainties in interpreting remote sensing observations and support the design of remote sensing missions to observe seasonal changes in snow storage.

Figure 3 illustrates the volume scattering of snow with bicontinuous DMRT. Figure 3 (a) shows the frequency dependence of the extinction coefficients $\kappa_e$ for different values of microstructure correlation length and with aggregation parameter $b = 1.2$. The results show the increase of extinction with increasing correlation length. The exponential of the frequency dependence is 3.3 from X- to Ku-band, which is less than the fourth-power law of Rayleigh scattering. This difference is due to the dense media effect, and the exponent is found to depend on the aggregation parameter $b$. The phase matrices at 13.3

GHz for snow are shown in Figure 3 (b). The phase matrix gives bistatic scattering as a function of the angle $\theta$ between the incident direction $\hat{s}$ and scattered direction $\hat{s}'$. In Figure 3 (b), P11 and P22 are co-polarization phase matrix elements and P12 is the cross-pol phase matrix. The phase matrix exhibits a dipole scattering pattern. The cross-polarization P12 is much larger than would be calculated by the sphere models because of the irregular shapes of the aggregates in the bicontinuous medium. The computed phase matrix and the extinction coefficients are substituted into the RTE which is then solved to calculate

backscattering. The results as a function of SWE for a homogenous snow layer are shown in Figure 4 for 6 channels with VV polarization at 10.2 GHz, 13.3 GHz, and 16.7 GHz in Figure 4 (a) and VH polarization for the same frequencies in Figure 4 (b). The results are compared with the Finnish NoSREx backscattering dataset within which tower measurements were taken over snowpacks with SWE up to 120 mm. The bicontinuous-DMRT model results are in good agreement with backscatter observations over the 6 channels of multiple frequencies and polarizations, albeit with a frequency-dependent bias for the

cross-pol results. Both the model predictions and measurements show high correlations with SWE with stronger correlations at 13.3 and 16.7 GHz than at 10.2 GHz.

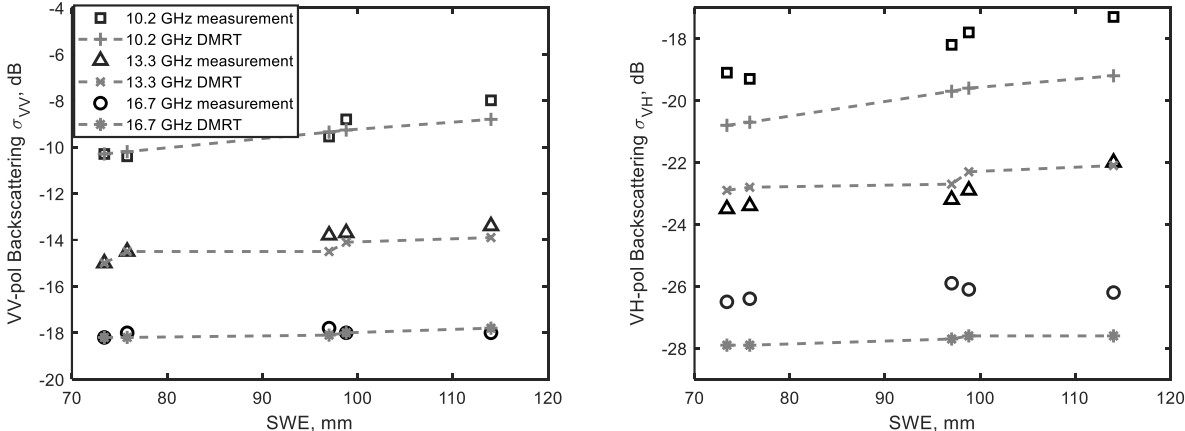

**Figure 4**: **Comparison of the DMRT/ bicontinuous media model using NoSREx 2010-2011 data backscatter against SWE for vertical co-pol (left) and cross-pol (right) at 10.2GHz, 13.3GHz, and 16.7GHz. Figures are adapted from Tan et al., 2015.**

Because of snow accumulation events and weather patterns, snow cover can have layering structures that correspond to variations of snow densities and grain sizes. Extensive work has been done in studying multi-layered models of snow using



the DMRT model (Liang et al., 2008), the HUT model (Lemmetyinen et al., 2010), the DMRT-ML model (Picard et al., 2013), and the MEMLS model (Proksch et al., 2015).

### 3.1.2 Interaction of radar waves with the ground surface beneath snowpack

Rough surface scattering from the snow/soil interface contributes to radar observations as indicated by equation (1), and should be removed when attempting to retrieve SWE. In this section, we discuss methods of determining the scattering of the snow/soil interface at X- and Ku-bands.

Physical models of rough surface scattering have been studied with the two classical methods of the small perturbation method (SPM) and the Kirchhoff approach (Ishimaru, 1978; Tsang and Kong 2001). Advanced analytical methods include the

advanced integral equation model (AIEM, Chen et al., 2003) and small slope approximation and its extensions (Voronovich, 1994; Elfouhaily and Johnson 2007). A description of roughness is $kh$ which is the product of the EM wavenumber $k$ and the surface rms height $h$. For soil surface scattering, the results of analytical models are typically limited to $kh < 3$. This means the upper limits of $h$ are 11.46cm, 2.65cm, 1.49cm, 1.05cm, and 0.83cm, respectively, for L-band (1.25GHz), C-band (5.4GHz), X-band (9.6GHz), low Ku-band (13.6Ghz), and high-Ku band (17.2GHz). Thus, surface rms heights must be at the

level of 1 cm or less to allow use of analytical models for surface scattering at Ku-band.

The soil surface can be described by a stationary Gaussian random process, so that knowledge of its covariance function is sufficient to describe its properties. The covariance function is often further parametrized in terms of its rms height and correlation length. Ground measurements have been made of these properties (Oh et al., 1992; Oh and Kay, 1998; Ulaby and Long, 2015). The measured correlation lengths have a maximum of 10cm. We label these roughness measurements as "limited

correlation length up to 10cm". However, in global retrieval of soil moisture using six months of the NASA Soil Moisture Active Passive (SMAP) radar data at L-band, the roughness was modelled as having a constant ratio of correlation length to rms height (Kim et al., 2012; Kim et al., 2014). The ratios used ranged from 5 to 20. The two assumptions are widely different for rms heights beyond 2cm. The rms heights in mountainous regions are large and can be up to 6cm. Surface scattering also depends on the soil permittivity which in turn depends on soil moisture (which describes the volume of water present per unit

volume of soil) and texture (which describes soil composition). Given these parameters, empirical models (Mironov et al., 2004; Peplinski et al., 1995) are available to calculate the soil permittivity. In addition to soil properties, land-cover including litter and vegetation, as well as rock outcrops impact the spatial variability of surface permittivity.





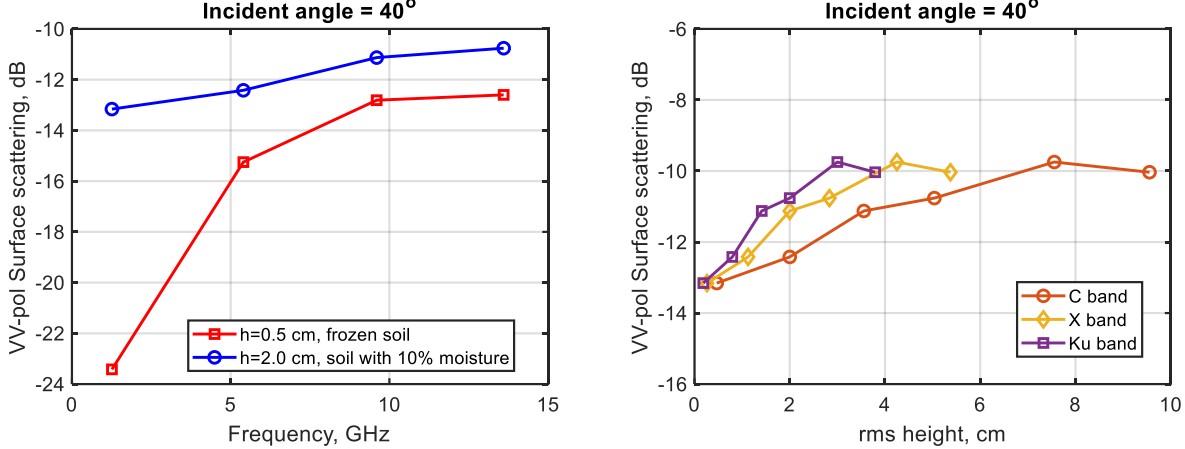

**Figure 5: (a) Backscattering at VV-polarization as a function of frequency: red curve is results with frozen soil (permittivity 4+1i) and rms height 0.5 cm and blue curve is results with soil of 10% moisture and rms height 2 cm. (b) Backscattering at VV-polarization as a function of rms height with soil of 10% moisture at C-, X-, and Ku-band.**

Given the limitations of the analytical models, full wave simulations based on numerical solutions of Maxwell's equations (NMM3D) were applied to L-band radar backscatter for the SMAP mission (Huang et al., 2010; Huang and Tsang, 2012). Look-up tables (LUT) were generated (Liao et al., 2016). However, these past NMM3D results were limited to $kh < 1.5$. To model rough surface scattering at X-band and Ku-band, results are needed with $kh$ up to 22. Recently NMM3D calculations were made with $kh$ up to 15 ($h = 4.16\ cm$ for 17.2 GHz). In Figure 5(a), we show the VV backscattering as a function of frequency for $h = 0.5cm$ and $h = 2cm$. In Figure 5(b), we show the VV backscattering as a function of rms heights at C-, X-, and Ku-bands. Both figures show saturation effects meaning that rough surface scattering saturates at large rms heights (~ 3-6 cm) and at high frequencies. The new results of $kh$ up to 15 are useful for studying rough surface radar backscattering at X- and Ku- bands.

We can use radar observations at X- and Ku-bands at a specific location before the snowfall to estimate the surface backscattering (Rott et al., 2010). Such an approach neglects any changes in soil properties and background land-cover during the snow-on season. Our proposed method for estimating and subtracting surface scattering contributions assumes that surface roughness is constant irrespective of weather and soil moisture. The method consists of using 4 channels, L-, C-, X-, and Ku-band, before snowfall and 2 channels, L- and C-band, after snowfall. Snell's law is used to adjust the incident angle at the snow-soil interface to account for the refraction angle at the air/snow interface. The approach is to use co-polarized radar observations at L- and C-band, which have much larger surface scattering than volume scattering, to estimate soil permittivity and surface roughness. The parameters obtained are then used to predict surface scattering contributions at X- and Ku-bands using the NMM3D model. The proposed approach assumes the availability of matchup L- and/or C-band SAR observations with revisits of 10 days. The revisits provided by the Sentinel-1 and NISAR systems, as well as future proposed continuation missions, suggest that such datasets are likely to be available during the time frame of a future snow observing mission.



To demonstrate the retrieval of rms heights and soil permittivity, we consider L-band UAVSAR radar full polarization observations under snow-on conditions using the NMM3D LUT (Liao et al., 2016). The UAVSAR dataset was collected from February to March, 2017, in the SnowEx 2017 campaign using 5 flights over the Grand Mesa region in Colorado, United

States. In-situ soil moisture measurements were also collected throughout 2017 from an installed meteorological observation station. We apply the time series retrieval algorithm developed for the SMAP mission (Kim et al., 2012) at the station location, that is at the point-scale. From the retrieved soil permittivity, the soil moisture is derived using Mironov's empirical model (Mironov et al., 2004). The retrieved rms height for the validation site is 1.9cm, and the comparison of retrieved and measured soil moisture is shown in Figure 6 (a). The retrieval achieves a root mean square error (RMSE) of $0.047 m^3/m^3$, a correlation

of 0.95 and a bias of $0.039 m^3/m^3$, demonstrating good agreement with the in-situ measured data. Based on the measurements, the soil moisture at this site remained relatively constant during the dry snow season (February 6 to March 8, 2017).

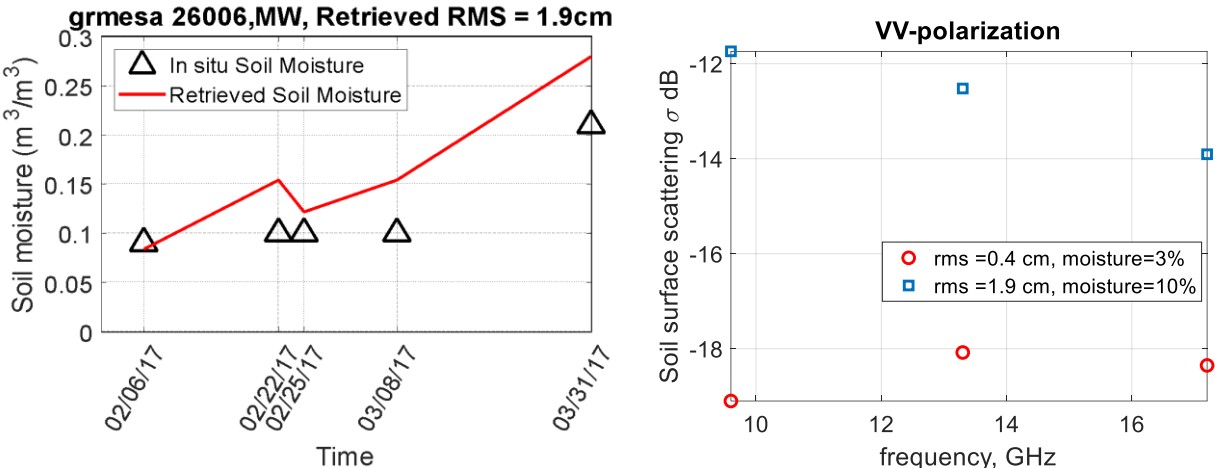

**Figure 6 (a) Retrieval of soil moisture compared with in situ-measurements. The retrieved rms height is 1.9 cm. The retrieval is based on based on L-band UAVSAR data from SnowEx 2017 campaign. The measured soil moisture is from SnowEx 2017 campaign**
**meteorological observations with a measured soil temperature of 0.6° C. The location of the station is 39.03388° N, 108.21399° W with an elevation of 3033m. (b) Simulated surface scattering with snow attenuation from X- to Ku-band at VV polarization. Snow parameters are with depth of 54 cm, density of 183 kg m⁻³, ⟨ζ⟩ = 1.2 mm, and b=1.2. For soil properties, blue marks are for the NoSREx 2010-2011 and red marks are based on the SnowEx 2017 campaign data shown in (a).**

Next, we apply the retrieved soil properties from Figure 6 (a) and soil measurements from the Finnish NoSREx dataset of

2010-2011 to calculate the surface scattering contributions with snow attenuation, $\sigma_{bg}^{pq} exp(-2\tau sec\theta_t)$, in equation (1) at X- and Ku-bands. In the Finnish NoSREx dataset, the measured soil moisture is 0.03 $m^3/m^3$ and the rms height is assumed to be 0.4 cm (Tan et al., 2015). To calculate attenuation from snow, snow properties measured in a Feb. 1, 2010 Finnish snowpit (snow density of 183 kg/m³ ) were applied (Tan et al., 2015) was applied to both cases. Figure 6(b) then shows surface scattering including snow attenuation. Higher frequencies typically experience higher volume scattering and greater

attenuation of the surface contribution. For the NoSREx dataset, surface scattering contributions at X-band are comparable to the total backscatter (shown in Figure 4) while smaller than volume scattering at Ku-band. The co-polarization contributions



of surface scattering can still be large at Ku-band. This approach of subtraction of surface scattering can be used to improve the accuracy of SWE retrieval (Zhu et al., 2018).

Continued studies are required to refine and validate this approach. These studies include extending NMM3D surface
modelling studies and LUT into cases with rms heights of 4 wavelengths or more so that the LUTs can be applied to Ku-band of 17.2 GHz. Unlike volume scattering of snow, rough surface scattering has a stronger dependence on incidence angle. Thus, the changes in incidence angle caused by topographical slopes, particularly, in mountainous regions must be considered.

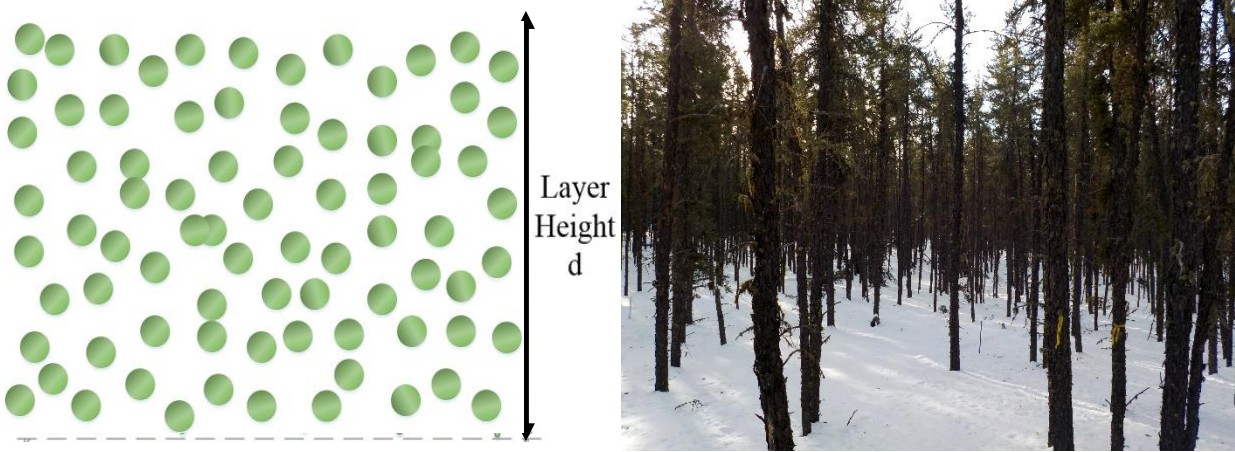

**Figure 7. (a) RTE assumption: uniformly randomly-positioned scatterers which are statistical homogeneous. (b) Illustration of trees**
**for a forest with snow cover beneath. The picture was taken on March 14th 2017 in a Jack Pine stand situated in the Boreal Ecosystem Research and Monitoring Sites (BERMS), Saskatchewan, Canada.**

### 3.1.3 Interaction of radar waves with forests above snowpack

In CoReH2O (Rott et al., 2010), studies on the effects of vegetation for snow retrievals indicate that, during the winter period, the presence of dormant herbaceous or short vegetation have small contributions to backscattering and do not affect the
sensitivity to SWE. For the effects of coniferous forests (CF), simulations were shown using the radiative transfer equations (RTE). The simulation results show that in the case of low fractional cover of CF (<25%), contributions of snow volume scattering are the dominant contributions to the radar signal, with the radar signals correlating with SWE. When the forest density or the fractional cover increases, the sensitivity to SWE decreases. The sensitivities are much affected by CF larger than 75%. However, the studies in Rott et al. (2010), were based on simulations using the RTE equation (2). In the RTE model,
the assumptions are independent scattering and uniform random positions of scatterers. The assumption of uniform random positions is the same as homogenization, meaning that there is an effective attenuation rate $\kappa_e$. For a forest height of $d$, the Beer-Lambert Law or the Foldy approximation states that the transmission through forest is given by the expression $e^{-\tau sec\theta_i}$ where the optical thickness $\tau$ is equal to $\kappa_e d$. Recently, full wave simulation results (Huang et al., 2017; Huang et al., 2019; Gu et al., 2021) have shown that the RTE is not valid for modelling microwave propagation and scattering in forests. In addition



to forest density and structure, snow interception in the canopy can vary widely in time and depending on snow type and canopy architecture and modify the transmissivity and scattering characteristics.

Microwave modelling of vegetation and forests initially began with the water-cloud model (Attema and Ulaby, 1978) (Figure 7 (a)). It was then extended by using RTE to include scattering effects in addition to absorption. RTE, like in the MIMICS model (Ulaby et al., 1990) and in the Torgata model (Ferrazzoli and Guerriero, 1995) have been used for modelling vegetation

and forests for several decades. There are two assumptions in RTE models. The first assumption is that the scatterers are uniformly random in positions which are valid for clouds and rainfall (Figure 7 (a)). However, in forests, such as coniferous forests (Figure 7 (b)), aspen forests and deciduous forests, the scatterers are aggregated in trees. Unlike clouds, which do not have gaps, there are gaps between the trees. Thus, waves, such as those in the Ku-band at 17.2 GHz with wavelengths of 1.74cm, can pass through gaps when gap sizes are larger than these wavelengths. The consequence of the assumption is that

RTE underestimates the transmission due to neglecting these gaps. The second assumption is that the leaves and branches are assumed to be single scatterers, and the extinction coefficients and phase matrices of equation (2) are calculated by adding the scattering cross section of the branches and leaves. This assumption is valid for cloud and rainfall as the water droplets can be assumed to be single scatterers. Bindlish and Barros (2001) applied the water cloud model formulation to the parameterization of vegetation backscatter from C- and L-band radar measurements using three vegetation parameters (a measure of vegetation

density, a measure of vegetation architecture, and a dimensionless vegetation correlation length) to characterize different types of vegetation in rangeland, winterwheat crops and pasture. They found that the estimation of land-cover and land-use class specific parameters resulted in significant improvements in retrieval of soil moisture, which suggests that a similar approach could be used for snow retrieval using multifrequency data along with detailed ancillary vegetation data sets to estimate the place-based parameters for the water-cloud model. In the case of forests, the scattering cross sections of each branch and each

leaf are strongly dependent upon frequency with the scattering increasing with frequency from the L- to Ku-bands. Such strong increase with frequency means the scattering at Ku-band is much larger than that at L-band, making the optical thickness at Ku-band large. Zoughi et al. (1986) conducted X-band radar measurements to identify the contributions from leaves, petioles, twigs, and branches of pine, oak, sycamore and sugar maple trees to backscatter and attenuation. In addition to quantitative differences related to tree architecture and vegetation moisture content, they reported that the backscatter is mainly produced

by the top layers of the canopy, petioles (tree microstructure) can significantly affect backscatter depending on their size relative to wavelength, leaves play an equally important role in attenuation and backscatter, whereas twigs and branches dominated in terms of backscatter with weak attenuation when leaves were not present. They did not consider the effect of tree trunks. However, the geometry (Figure 7(b)) is that the branches and leaves are attached to the tree. For a coniferous forest, there are primary branches and secondary branches attached to a tree. The needle leaves are aggregated and are attached to

branches. Thus, the entire tree itself should be treated as a single scatterer rather than an individual branch or an individual leaf. Therefore, the phase matrix of a tree should be used rather than incoherently adding the scattering cross sections of branches, leaves, and the trunk for a tree. Using a tree as a single scatterer will have weaker frequency dependence than that predicted by RTE.



Full wave simulations to solve Maxwell equations among trees or plants were deemed to be computationally formidable.
Recently, a computationally efficient hybrid method (HB) has been developed to perform full wave simulations (Huang et al.,
2017; Huang et al., 2019; Gu et al., 2021). The hybrid method is a combination of the Foldy-Lax multiple scattering equations
and commercial-off-the-shelf software of computational electromagnetics. The hybrid method consists of two steps. In the
first step, a plant or a tree is treated as a single scatterer. Commercial-off-the-shelf software is used to calculate the scattering
matrix of a single plant or a single tree. In the second step, coherent wave interactions among the plants and trees are calculated
by using the Foldy-Lax multiple scattering equations (Tsang and Kong, 2001). The coherent wave interactions among trees
consider the aggregating properties of branches and leaves and the gaps among the trees. Simulations were performed for the
transmission through a simulated forest (Huang et al., 2019) consisting of 196 cylinders of 20 m height and 12 cm diameter
arranged as shown in Figure 8. The results are tabulated in Table 1. The results show that the transmission is almost twice that
of RTE.

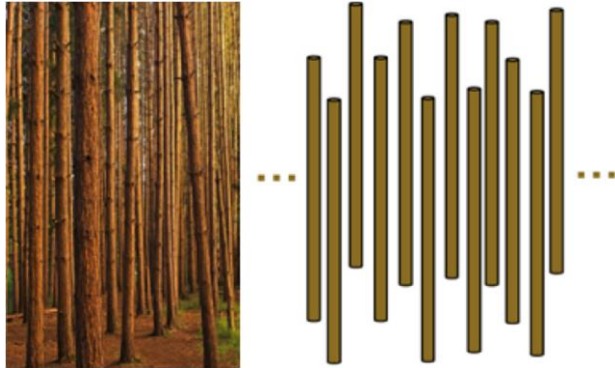


Figure 8: Tree trunks (left) are modelled as dielectric cylinders (right). The figure is adapted from Huang et al., 2019.

**Table 1: transmission coefficient from RTE based on distorted Born approximation (RTE/DBA) and the hybrid method from Figure 8. The table is adapted from Huang et al., 2019.**

|  | RTE/DBA | Hybrid method |
| --- | --- | --- |
| Transmission | 0.35 | 0.66 |

To consider frequency dependence, we calculate the transmission through a field consisting of 196 wheat plants (Figure 9 (a))
at L-, S- and C-bands (Gu et al., 2021). Results of Figure 9 (b) are compared with RTE. Firstly, the results show the
transmission of full wave simulations is much higher than RTE. Secondly, the transmission at C-band is only slightly less than
that at S-band while RTE shows big drop in transmission from S- to C-band. The results show that full wave simulation has
weaker frequency dependence than RTE.



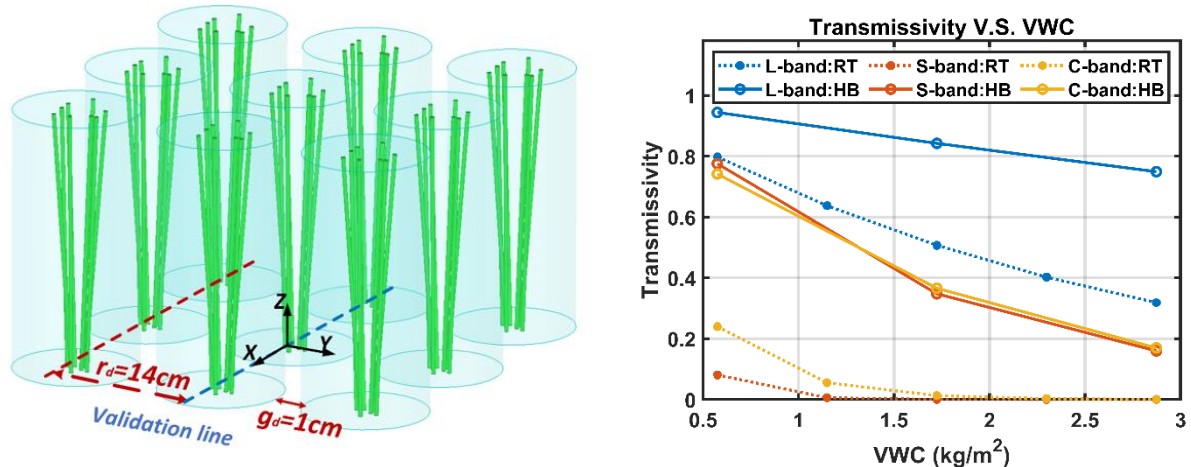


**Figure 9: (a) Scattering from wheat plants are the radius of the circumscribing cylinder 6.5 cm, distance between the centers of 2 circumscribing cylinders is $r_d$ =14 cm, the closest distance between 2 circumscribing cylinders is $g_d$ = 1 cm. Figure is adapted from Gu et al., 2021. (b) Transmission of microwave through wheat of different calculated using the hybrid method and the RTE vary with volume water content (VWC).**

Ray tracing can also be used for the high-frequency channels of Ku-band. At Ku-band, because the wavelengths are smaller than object size and object spacing of trees, results of ray tracing may be acceptable. However, ray tracing, in the opposite extreme of RTE, has no frequency dependence although frequency dependence can be introduced in an ad hoc manner such as by only keeping the dominant scatterers at the operating frequency. .

At Ku-band (17.2 GHz), the wavelength is 1.74 cm, which is much smaller than the gaps in trees. The wave can travel in
straight lines as rays through the gaps. In such scenario, the Ku-band waves will travel like the case of lidar which has been shown to be able to penetrate forest canopies. In wireless communication, ray tracing has been performed as a path-loss model in forests (Ling et al., 1989; Kurt et al., 2017). Inter-comparisons among results of full wave simulations, RTE and ray tracing will be made in future work.

**3.2 Experimental measurements of radar-landscape interactions**

Collection of experimental data is a prerequisite for the development of Earth Observation satellites. Ground-based and airborne sensors provide means to collect observations of the geophysical parameter of interest in a relatively controlled environment. These measurements provide the basis for the validation of forward modelling approaches and development of retrieval algorithms prior to launch of the space-borne mission. The measurements also help to understand the spatial resolution and temporal requirements for a spaceborne mission.

Ground-based sensors deployed on tower structures allow near-continuous observations over extended periods, which are critical for understanding both slow and seasonal processes as well as rapid phenomena induced by diurnal changes at the sensor foot-print. Such temporal features are of particular importance for seasonal snow cover. Airborne observations, or the



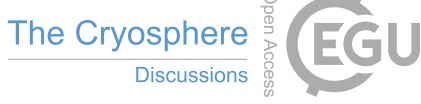

deployment of ground-based sensors on other mobile platforms, provide the ability to expand localized observations to a larger scale, allowing to observe the effect of heterogeneous land cover and vegetation on Earth Observation signatures. Seasonal

snow presents a particularly challenging target for observations, due to the high variability of snow over both temporal and spatial scales. Hence, several both localized, ground-based campaigns as well as airborne sensor deployments have been conducted in recent years, in an attempt to understand radar signatures from seasonal snow cover. These campaigns have covered diverse snow and climatological conditions.

**Table 2: Summary of ground-based active microwave sensors for snow studies.**

| Ground-Based Campaign | Dates | Sensor | Freq. | Pol. | Location/snow regime | Data availability |
|---|---|---|---|---|---|---|
| Can-CSI, CASIX | 2009-2011 | UW-Scat | 9.6, 17.2 GHz | VV, VH, HV,HH | Churchill, Manitoba, Canada Tundra, Taiga, Lakes, Sea Ice | Available from Dr Richard Kelly rejkelly@uwaterloo.ca |
| NoSREx | 2009 - 2013 | ESA SnowScat | 10-17 GHz (stepped frequency) | VV, VH, HV, HH | Sodankylä, Finland / boreal forest | ESA EO campaign portal |
| SnowEx '17 | 2017 | UW-Scat | 9.6, 17.2 GHz | VV, VH, HV, HH | Grand Mesa, Colorado, USA | NSIDC |
| APRESS | 2019 - 2021* | ESA WBScat | 1-40 GHz | VV, VH, HV, HH | Davos-Laret, Switzerland / Alpine Sodankylä, Finland / boreal forest | Not yet available |

**3.2.1 In situ radar experiments and signatures**

The ground-based campaigns are summarized in Table 2. Ground-based campaigns of Can-CSI and CASIX were conducted between 2009 and 2010, and multiple field campaigns were completed near Churchill, Manitoba, Canada, as part of the Phase A science activities of CoReH2O. These campaigns aimed to evaluate the potential for dual-frequency X- and Ku-band snow properties retrievals in subarctic environments. Central to Churchill campaigns was deployment of the University of Waterloo

Scatterometer (UW-Scat), a novel ground-based radar system analogous to the proposed configuration of CoReH2O (King et al., 2012). In Europe, ESA initiated the deployment of SnowScat (Werner et al., 2010), a stepped frequency, fully-polarimetric





ground-based radar in a series of campaigns in the boreal forest zone in Northern Finland (Lemmetyinen et al., 2016). The campaign was called NoSREx and operated SnowScat over four winter seasons, complemented by passive microwave radiometry and regular snow microstructural observations. These campaigns have been instrumental in enhancing our
understanding of snow-microwave interactions and providing data to develop and evaluate forward models simulating backscattering from snow cover (King et al., 2015; Tan et al., 2015; Proksch et al., 2015a), as well as developing retrieval approaches (Cui et al., 2016; Lemmetyinen et al., 2018).

Previously, in section 3.1.1, we have made use of the NoSREx campaign 6 channels of backscattering data of VV at 10.2 GHz, 13.3 GHz and 18.7 GHz, and HV at 10.2 GHz, 13.3 GHz and 18.7 GHz in comparisons with the simulation results of
bicontinuous-DMRT models. The comparisons have validated both the ground campaign measurements and the physical models (Tan et al., 2015).

Recent ground campaigns include APRESS in which the ESA WBScat instrument, a full-polarization radar operating at 1-40 GHz, was deployed for a full winter season in 2019-2020 measuring an Alpine snowpack in Davos, Switzerland. For the winter of 2020-2021, the instrument was set up in Sodankylä, Finland, to collect data over a sparsely forested site. These sensors and
deployment have generated and will continue to generate critical datasets for characterizing the radar backscattering of snow, the effects of rough surface scattering, and forests. They will help to advance retrieval development when coupled with advancements in field methodology and forward modelling capabilities.

### 3.2.2 Airborne experiments and signatures

Airborne campaigns (listed in Table 3) provided the first experimental demonstration of the sensitivity of Ku-band polarimetry
scatterometer (POLSCAT) backscattering to SWE (Yueh et al., 2009). CoReH2O provided further impetus for the development of new airborne sensors. The ESA SnowSAR (a dual-polarization, airborne, side-looking SAR operating at X- and Ku-bands (Coccia et al., 2011; Meta et al., 2012) was deployed at several sites in Northern Finland, the Austrian Alps, Northern Canada and Alaska between 2011 and 2013. The purpose of the flight campaigns was to collect data over a range of climatological snow classes and land cover regimes. All flight campaigns were supported by extensive measurement of snow
properties, including vertical profiles of snow stratigraphy and microstructure. The campaigns have enabled the further assessment of e.g., vegetation effects on backscatter (Cohen et al., 2015; Montomoli et al., 2016), the effect of spatially variable microstructure (King et al., 2018) and further elaboration of modelling and retrieval capabilities (Zhu et al., 2018). Figure 10 demonstrates the effect of changing snow conditions on the observed Ku-band co-polarized backscatter during two of the SnowSAR flights in Finland.




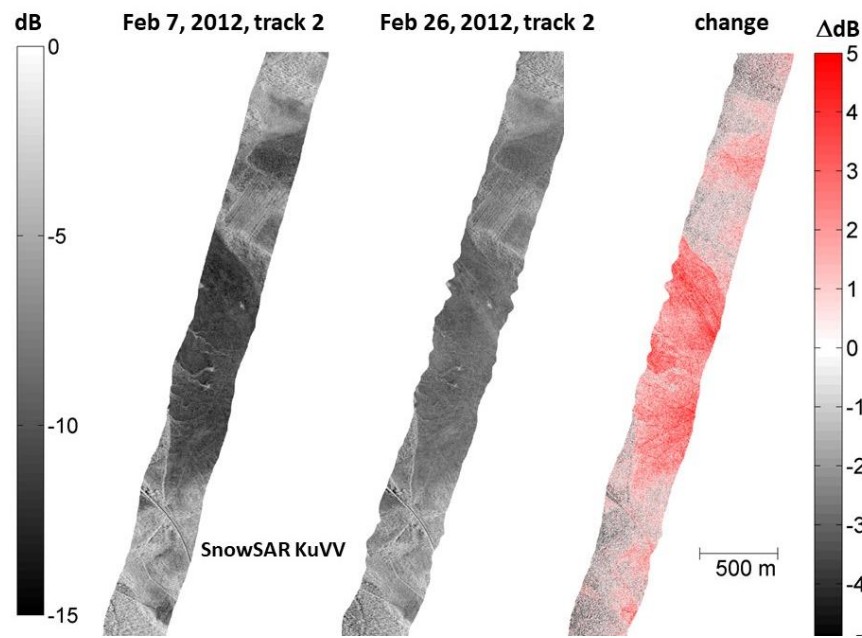

**Figure 10: Demonstration of observed Ku-band VV-pol backscattering from two consecutive SnowSAR flight campaigns in Sodankylä, Finland. The difference in backscatter is depicted on the right, with red implying an increase. Increases of measured backscattering are correlated with the increase of SWE. The measured SWE between the flights increased on average by 41 mm in non-vegetated areas, which show the highest increase (Figure adapted from Lemmetyinen et al., 2014).**

Recent airborne campaigns including SnowEx 2017, SnowEx 2020, and TVCExp 2019 have deployed a new generation of airborne systems to address known uncertainties including penetration in dense vegetation, background interactions, and InSAR applications (Table 3). As part of ongoing research at Trail Valley Creek, a new Ku-band InSAR (13.285 GHz) developed by the University of Massachusetts was deployed during the winter of 2018-2019. This system was developed to allow rapid deployment aboard common commercial platforms, leading to three successful acquisition periods throughout the

winter. Coupled with objective measurements of snow microstructure and a distributed network of soil permittivity sensors, these data are now being used to develop InSAR and backscatter retrieval methods for future missions. The Snow Water Equivalent SAR and Radiometer (SWESARR) is a tri-band synthetic aperture radar (SAR) and a tri-band radiometer. Both the active and passive bands utilize a highly novel current sheet array (CSA) antenna feed. SWESARR has three active (9.65, 13.6, 17.25 GHz) and three passive (10.65, 18.7, 36.5 GHz) bands. Radar data are collected in dual polarization (VV, VH)

while the radiometer makes single polarization (H) observations. During SnowEx 2020, NASA Goddard's SWESARR demonstrated for the first time that X-band, low Ku-band and high Ku-band SAR acquisition can be made through a single antenna feed. Data collected during this campaign coincided with detailed measurements of vegetation structural properties and under canopy snow properties that will be critical to address the effects of vegetation and forests in SWE retrieval.

Airborne campaigns are planned for 2022-2023 for both Canada and US SnowEx. These future campaigns will address the

following questions: (a)What is the maximum forest density for retrievable SWE at Ku-band? Recent full wave simulations using Maxwell equations suggest that penetration through forests is higher than predicted by past models of radiative transfer.

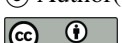



(b) What is the saturation maximum depth retrievable SWE at X and Ku-bands? Using cross polarizations, can we have higher depth of penetration such as 1 meter to 3 meters? (c) What is the impact of stratigraphy and how snow physics models can help to retrieve SWE in the presence of stratigraphy? (d) How permafrost and the changing freeze depth in some areas would
affect the surface scattering contributions and the ability to subtract surface scattering estimated under snow-free conditions. With 6 measurements in SWESARR which are co-polarization, and cross polarization at three frequencies, what are the optimum combinations of polarization and frequencies for SWE retrieval?

**Table 3: summary of airborne deployments of active microwave sensors for snow studies.**

| Airborne Campaign | Dates | Sensor | Freq. | Pol. | Location/snow regime | Data availability |
|---|---|---|---|---|---|---|
| CLPX-I, -II | 2002, 2003 | JPL POLSCAT | 13.95 GHz | VV, VH, HV, HH | Colorado, USA / Alpine & prairie | NSIDC |
| SnowSAR | 2010 -2011 | ESA SnowSAR | 9.6, 17.2 GHz | VV, VH | Sodankylä and Saariselkä, Finland / Taiga & tundra | ESA EO campaign portal |
| AlpSAR | 2011 -2012 | ESA SnowSAR | 9.6, 17.2 GHz | VV, VH | Leutasch, Mittelbergferner, Rotmoos, Austria / Alpine | ESA EO campaign portal |
| TVCExp | 2012 -2013 | ESA SnowSAR | 9.6, 17.2 GHz | VV, VH | Trail Valley Creek, NWT, Canada / Tundra | Not yet available |
| SnowEx '17 | 2017 | ESA SnowSAR | 9.6, 17.2 GHz | VV, VH | Grand Mesa, Colorado, USA / Alpine | NSIDC |
| TVCExp | 2018-19 | UMass Ku-InSAR | 13.285 GHz | VV | Trail Valley Creek, NWT, Canada / Tundra | Not yet available |
| SnowEx '20 | 2020 | NASA SWESARR | 9.65, 13.6, 17.25 GHz | VV, VH | Grand Mesa, Colorado, USA / Alpine | NSIDC |

*ongoing





### 3.3 Linking field measurements of snow with theoretical models of snow-radar interactions

Quantification of physical snow processes through objective measurements has revolutionized our understanding of microwave interactions with complex snowpacks at multiple scales (millimetre to kilometre). Objective methods of determining snow grain size in the field have only been available over the last decade. Prior to this, grain size was typically quantified visually with a hand lens or microscope (Fierz et al., 2009) and could be subject to errors of up to 1 mm in grain size estimation (Leppänen et al., 2015). Laboratory processing methods include gas absorption (Legagneaux et al., 2002) and thin section imaging techniques (H. Bader et al., 1939). As described in section 3.1.1, µ-CT measurements of snow samples obtained in the field have revolutionized our ability to model EM interaction from snow. Today, field-based instruments can quantify the snow specific surface area (SSA) rapidly in the field either through near-infrared reflectance (e.g. Gallet et al., 2009) or penetrometry (Proksch et al., 2015b). SSA is often related to an effective sphere diameter, that is the diameter of a sphere taken with the measured SSA (Mätzler, 2002). In practice, these metrics must often be scaled to match output from radiative transfer models (Montpetit et al., 2012). Here, we describe how this new knowledge of microscale variability over seasonal timescales can be leveraged to inform algorithms applied at landscape-scales.

### 3.3.1 Spatial variability of field measurements

Geolocated measurements are vital to quantify variability of snowpack properties within sensor footprints (airborne or tower). Figure 11 (a) suggests an optimal configuration of snow depth and Snow MicroPenetrometer (SMP) measurements to create representative distributions of snowpack properties within airborne swaths. The main 222 m transect of snow depth and SMP profiles, located along an airborne swath centreline, has variable spacing ($10^{-1}$, $10^0$, and $10^1$ m) on either side of a central pit to capture different horizontal length scales of variability. A shorter 22 m orthogonal transect with $10^{-1}$ and $10^0$ m spacing bisects the main transect at the central pit, and a spiral of snow depths extends from the central pit out to an 11 m radius (Figure 11 (a)). Measurements additional to the main transect allow omnidirectional analysis of snow depth variability and bi-directional analysis of snow microstructural properties, both of which may be influenced by dominant prevailing wind direction or irregular patterns of subnivean vegetation. Ideally, snow depths and positions are measured using automatic depth probes with integrated GPS providing position accuracy to ±2.5 m, e.g. Magnaprobes (Sturm and Holmgren, 2018). Where forest canopies obscure GPS satellite connection or snowpacks are deeper than 180 cm (current maximum magaprobe length), hand probes are used in measured grids arranged relative to a known absolute position. Accuracy of snow depths range from nearly 0 cm on hard ground to ~5 cm in soft subnivean vegetation. Where snowpit locations are not predetermined, measured snow depth distributions can be subsequently used to determine the location of snowpit(s) to match mean depth or multiple pits spanning interquartile ranges.



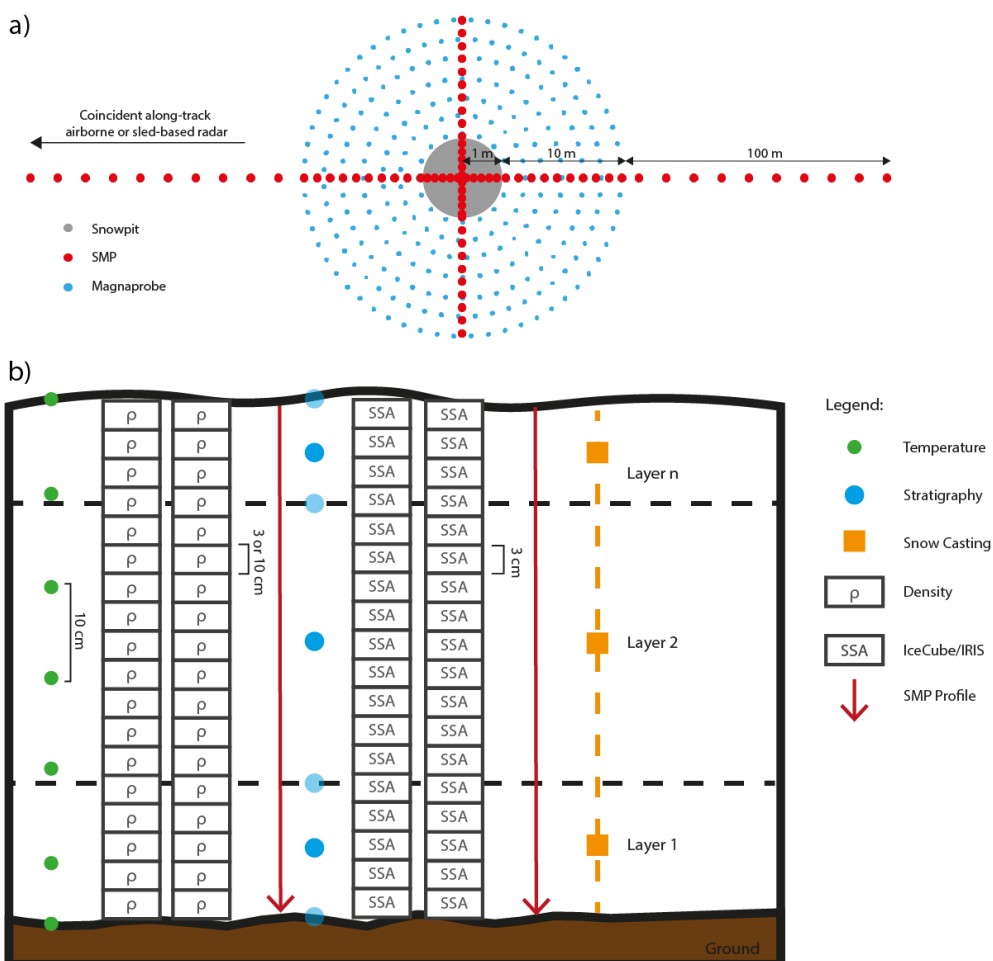

**Figure 11: Optimal measurement configurations for evaluation of snow properties in sensor footprints.**

One-dimensional (vertical) measurement of snowpack properties for SAR retrievals require objective measurements of snow microstructure. SMP allows rapid (< 1 minute) force profile detection at millimetre resolution (Proksch et al., 2015b), which is used to derive density, correlation length and Specific Surface Area (SSA); quantities directly or indirectly used by radiative transfer models (Chang et al., 2016; Picard et al., 2018). The speed of data acquisition allows for SMP measurements to be

used in a distributed manner along transects, but SMP also supports other coincident snowpit measurements using different techniques. Figure 11 (b) shows a schematic of optimal snowpit measurements. Double-sampled volumetric density measurements, 100 cm$^3$ box-cutter with 3 cm vertical resolution for shallow snowpacks or 1000 cm$^3$ wedge-cutter with 10 cm vertical resolution for deep snowpacks (Proksch et al., 2016), are averaged at each vertical position. Following principles presented in Gallet et al. (2009), 3 cm vertical resolution double-sampled SSA measurements are made using an InfraRed

Integrating Sphere (IRIS) (Montpetit et al., 2012) or an A2 Photonic Sensors IceCube (Zuanon, 2013). Micro-CT analysis of snow casts (Schneebeli and Sokratov, 2004; Lundy et al., 2002), consisting of entire profiles or samples of critical layers, are





used as a benchmark for corroboration of all other measurements. However, in-practice, in-situ snow casting and subsequent cold-laboratory micro-CT analysis requires a much higher level of expertise and processing time than SMP or IRIS/IceCube measurements, meaning field application of micro-CT is often limited.

Profiles of snow temperature using well-calibrated stem thermometers at 10 cm vertical resolution are important parameters for radiative transfer models, in conjunction with stratigraphic identification of snow layer boundaries and ice lenses using hand hardness. Visual identification of grain type (Fierz et al., 2009) using a hand lens or macroscope is an important complimentary measurement for layer classification and understanding the seasonal history of snowpack processes. However, using similar visual methods to quantify snow 'grain size' are too subjective to create a microstructural metric for further use

in radiative transfer models. The SSA measurements provide much better accuracy for use in radiative transfer models.

Near-infrared (NIR) photography allows two-dimensional analysis of layer boundary position (Tape et al., 2010) and layer thickness variability (Rutter et al., 2019) in snow trenches, quantifying spatial variability of stratigraphy around a single snowpit profile. It also enables measurements of layer boundary roughness, particularly of the snow-air and snow-ground interfaces. Other methods to characterise snow-air surface roughness use photographic image contrast analysis of dark boards

placed behind snow (Fassnacht et al., 2009; Anttila et al., 2014) and subnivean roughness of areas cleared of snow using pin profilers, LiDAR scanning (Chabot et al., 2018; Roy et al., 2018) or structure from motion photogrammetry (Meloche et al., 2020). The subnivean roughness between snow and soil give significant contributions of rough surface scattering because of the contrast of dielectric constants between snow and soil.

At the landscape scale, understanding snow spatial variability is of critical importance with respect to the development of

methodologies that can observe and model discrete and bulk properties of snowpack with low uncertainty. Mountains, hills, and valleys exert aerodynamic roughness controls on snowfall trajectory, enhancing snow accumulation and redistribution processes often dominated by blowing snow and sublimation. Exposed topography (e.g. alpine areas or open upland plateaus) are typically scoured of snow while enhanced accumulation is found in gullies or on the lee-side of plateaus (Pomeroy et al., 1993; Liston and Sturm, 1998). Once accumulated, the persistence of snow on the landscape is influenced by terrain slope and

aspect which control a snowpack's energy budget; the incoming heat energy to north-facing slopes is radically different to that for south-facing slopes that can lead to pronounced variations of SWE at the landscape scale (Lopez-Moreno et al., 2014). Local slope angles also cause changes in the incidence angle of radar backscattering, which may impact assumptions underpinning microwave retrieval algorithms because rough surface scattering has a strong angular dependence.

The distribution of tree canopy, woody biomass, and the fragmentation characteristics of the vegetation stands play a significant

role in how snow accumulates in a forested landscape. Metrics describing plant functional types (e.g. deciduous/coniferous or broadleaf/needleleaf, canopy densities and heights, etc.), are used to quantify spatial difference in simulations of sub-canopy snow and microwave radiative transfer. Correcting for forest microwave attenuation has shown that forest transmissivity plays an important role in the observability of the sub-canopy snow. However, transmissivity changes through the season as the woody biomass undergoes progressive cooling at temperatures below 0° C (Li et al., 2020). Moreover, observed transmissivity

of a tree stand is also impacted by the forest gap fraction or forest fragmentation. Landscape metrics can be used to characterize





these ecological factors using high-spatial-resolution active and passive optical observations (Vander Jagt et al., 2013). The impact of low stand shrub vegetation is also important for snow accumulation in subalpine, tundra and sub-tundra regions, and in the understorey of forested environments. Shrub-dominated landscapes retain snow more effectively than graminoid plant cover but less than forest-covered regions (Marsh et al., 2010).

The soil type and state affect microwave observations of snow at the landscape scale because, at microwave wavelengths, the soil relative permittivity can play an important role in how reflected or backscattered energy is attenuated at the snow-ground interface. Generally, relative permittivity of soil is controlled by the texture, moisture content, and thermal heat content of the soil. Seasonal change in soil water state (liquid or frozen), is also important since relative permittivity changes significantly as the surface soil moisture changes state. This is all complicated by the variability in soil type at the landscape scale, and

especially the mix of organic and inorganic content; peat soil landscapes are very complex in their microwave response whilst inorganic soils are somewhat simpler to characterize. In agricultural landscapes, especially post-harvest, the surface soil layer tends to be more spatially uniform and freeze earlier, making the variations in relative permittivity of the soil relatively constant.

In SWE retrieval, the snow-soil rough surface scattering and the forest effects on transmission and backscattering give bias in
radar measurements. It is important to evaluate the magnitudes of these effects and how such bias in radar measurements vary with time.

### 3.3.2 Seasonal variability

Temporal change in snowpack properties have important implications for radar backscatter, in particular: 1) snow mass change, 2) metamorphism of snow microstructure, and 3) liquid water content and refreeze (ice lenses). High temporal resolution
(hourly) measurement of snow mass accumulation and ablation using snow pillows, e.g. SNOTEL (Yan et al., 2018), or passive gamma radiation SWE sensors (Smith et al., 2017) provide excellent evaluation data to test SWE retrieval algorithms. However, such point measurements of SWE are spatially limited and seasonal variability in SWE is more commonly estimated through depth measurements, at a point using an acoustic sounder or spatially distributed from lidar, and periodically measured or modelled snow density. Uncertainties in modelled snow densities commonly dominate uncertainties in measured depth
(Raleigh and Small, 2017).

Seasonal change in snow microstructural properties can strongly influence scattering of radar backscatter, especially in snowpacks of which depth hoar is a significant component. Constraining the proportions of snowpacks that have different scattering properties (e.g. surface hoar, wind slab, consolidated layers, indurated hoar or depth hoar) is required to prevent the retrieval of SWE from backscatter becoming an ill-posed problem. Frequent (weekly) objective profiles in snow pits (section
5.3) are optimal, however, in lieu of in-situ pit measurements, thermistors situated at different heights above the ground that become sequentially buried in accumulating snow allow calculation of temperature gradients within the snowpack. Consistent temperature gradients can be used as a proxy for likely snow crystal type (Domine et al., 2008): rounded ($<10°$ C m$^{-1}$), facets





(10-20° C m$^{-1}$), depth hoar (>20° C m$^{-1}$). Where internal snowpack temperatures are not available, 2 m air temperatures and near surface soil temperatures can provide a bulk estimates of snow temperature gradients.

Profiles of liquid water content (LWC) are measured in-situ through insertion of dialectic devices (Denoth, 1994; Sihvola and Tiuri, 1986) into a snowpit wall. LWC can also be retrieved through non-invasive techniques using only GPS signal attenuation (Koch et al., 2019) or electrical self-potential (Thompson et al., 2016). Where mid-winter melt events are observed, either directly from LWC measurements or via inference from meteorological inputs, the chances increase of ice lens formation within the snowpack, which an important consideration for SAR backscatter retrievals. Snow wetness affects the radar

backscattering (Stiles and Ulaby, 1980). The dielectric constants of wet snow have been modelled as a function of snow wetness (Ulaby and Long, 2015).

## 4 Characterization of snowpack properties using radar measurements

### 4.1 Describing the retrieval problem

In early studies, empirical models were proposed for SWE retrieval (Ulaby and Stiles, 1980; Drinkwater et al., 2001). These

are unsuited for all snow types—e.g., ephemeral, prairie, maritime, and mountain snow, etc. (Sturm et al., 1995). Later investigators applied multiple channel measurements to determine snow parameters (Shi and Dozier, 2000; Rott et al., 2010). Recent algorithms (Cui et al., 2016; Xiong and Shi, 2017; Lemmetyinen et al., 2018; Zhu et al., 2018; King et al., 2019) are based on physical models in which RTM are used. Physical model-based retrieval algorithms consist of three parts: 1) a physical model of snow volume scattering, 2) estimation of a priori parameters, and 3) a cost-function inversion of the physical

model to obtain SWE. In this section, we will describe the estimation of *a priori* parameters and SWE retrieval procedures.

Volume scattering of snowpack is a function of parameters including SWE, density, snow microstructure, and stratigraphy. Paramount for the success of the retrieval is the ability to predefine or constrain some of these unknown parameters, in particular the parameters that are used to characterize the snow microstructure. Based on the RTM, volume scattering is a function of snow depth, density, snow microstructure and layering structure (Rott et al., 2010; Zhu et al., 2018; King et al.,

2018). A challenge is the non-uniqueness in inversion as different combinations of SWE and parameters of snow microstructure can give similar backscattering (Tsang et al., 2004; King et al., 2018). *A priori* estimates of parameters of snow microstructure can be used to improve the accuracy of retrieval by constraining the cost function with estimated statistical uncertainties.

Assuming normal distributions for the errors in forward simulations and observations of backscatter, the cost function of a

maximum likelihood estimate for SWE and snow microstructure can be formulated as follows:

$$F = \left\{ \sum_{i=1}^{N} \frac{w_i}{2s_i^2} \left( \sigma_i^{obs} - \sigma_i^{model}(SWE, x) \right)^2 + \frac{w_x}{2s_x^2} (x - \bar{x})^2 \right\} \quad (3)$$

where $\sigma_i^{obs}$ are radar observations from the $i$th channel, and $N$ is the total number of channels for measurements. In CoReH2O, $N=4$ for VV and VH polarizations of X- and Ku-band. The backscattering predictions of snowpack are given by $\sigma_i^{model}$. In





the above, $s_i^2$ are the error standard deviations of the radar measurements. The parameter $x$ is related to snow microstructure,
such as single scattering albedo, correlation length, and grain size. $s_x^2$ is the variance of *a priori* constraint. In Cui et al. (2016)
and Zhu et al. (2018), $s_i$ are assumed to be 0.5, which are based on the error standard deviations of radar measurements. $w_i$
and $w_x$ are the weighting factors in the retrieval. Dual frequency retrievals are using either X- (9.6GHz) and Ku-band of 17.2
GHz as in CoReH2O or dual Ku-band of 13.6 GHz and 17.2 GHz as currently being proposed (see Section 6). However, cross
polarizations have not been fully utilized and algorithms have been using the two co-polarizations of the dual frequency
measurements. The two frequency measurements exploit the frequency dependence of volume scattering in snow. The two
parameters that strongly influence the backscattering measurements are SWE and snow grain size. Then the problem becomes
retrieval of two parameters from two measurements.

**4.2 Constraining the retrieval problem with prior information**

Some of the challenges in retrieving snow properties from radar measurements can be addressed by using so-called "prior
information" in a Bayesian sense, such as is sometimes done for passive microwave retrievals (e.g. Pan et al., 2017). In actual
implementation of the algorithm for global monitoring, there are needs of auxiliary information and *a priori* parameters. There
are also needs on how to combine all these *a priori* information effectively in a retrieval algorithm. The amounts of satellite
SAR data are massive, and computational efficiency needs to be achieved so that the retrieval algorithm can be operated in
real time. As described in the retrieval algorithm, each *a priori* parameter $x$ associated with the 2nd term in the cost function
$\left(\frac{w_x}{2s_x^2}(x - \bar{x})^2\right)$ with mean $\bar{x}$ , weight $w_x$, and uncertainty $s_x$.

**Table 4: Classifications of snow type. From Sturm et al. (1995). Note that the "no data" label refers to data available in the study of Sturm et al., rather than availability in general.**

| Snow Class | Depth range (cm) | Average bulk density (g/cm$^3$) | Number of layers | Grain size: new snow, fine, medium, coarse grained, depth hear |
|---|---|---|---|---|
| Tundra | 10-75 | 0.38 | 0-6 | Wind slab, depth hoar |
| Taiga | 30-120 | 0.26 | >15 | New snow, depth hoar |
| Alpine | 75-250 | No data | >15 | New, fine, medium, coarse, depth hoar |
| Maritime | 75-500 | 0.35 | >15 | New, wet |
| Ephemeral | 0-50 | No data | 1-3 | Recent snow |
| Prairie | 0-50 | No data | <5 | Wind slab, recent, fine |
| Mountain | Deep snow up to 400 cm | No data | variable | |



### 4.2.1 Leveraging snowpack information and snow classes

Sturm et al. (1995) used snowpit measurements from Alaska to objectively define seven snow classes. These classes can be
predicted based on available land cover and meteorological information, globally. Snowpack properties are listed in Table 4, above, with information compiled from Sturm et al. (1995), rearranged from the point of view of *a priori* information for X- and Ku-band radar backscattering and retrieval. Retrieval algorithms could, for example, leverage the predicted snow class to define *a priori* estimates to guide SWE retrievals.

### 4.2.2 Snow microstructural models

**4.2.2.1 Background**

Across the electromagnetic spectrum, the interaction of radiation with snow cover is mediated by snow microstructure (West et al., 1993; Wiscombe and Warren, 1980; Nolin and Dozier, 2000). However, due to differing penetration depths, visible and near-infrared measurements are sensitive to grain size at the surface, whereas microwave measurements are sensitive to grain size at depth (Hall et al., 1986). The sensitivity of radar measurements to microstructure properties has been demonstrated by
both models (Xu et al., 2012; Proksch et al., 2015a) and experiments (King et al., 2015; Rutter et al., 2019). Algorithms to retrieve SWE from radar backscatter typically solve for both SWE and some measure of snow microstructure (e.g. single-scattering albedo, correlation length) and regularization terms or prior information on microstructure is often included in the retrieval cost function (as described in the previous section; see Rott et al., 2010). In addition to important advances in measuring snow microstructure in the field, as described in previous sections, new work to simulate the evolution of snow
grain size has indicated great potential for improving radar retrieval algorithms of SWE. In the cost function of retrieval algorithm, the *a priori* estimate term is $\frac{w_x}{2s_{\bar{x}}^2}(x - \bar{x})^2$ with grain size being the most important *a priori* parameter.

To prevent confusion, it is important to begin with clear definitions for snow microstructure. Snow is a continuous, granular, bonded medium, composed of irregularly-shaped ice crystals, water vapor, liquid water, and void areas. The term "microstructure" commonly refers to snow grain size, shape, bonding, and distribution. Snow microstructure evolves both
vertically within a snowpack and temporally throughout the snow season (as well as exhibiting significant horizontal spatial variability). The dendritic forms of new snowflakes sublimate, leading to a wide range of rounded or faceted shapes, depending on snowpack conditions. For many years, snow microstructure was referenced by the term "grain size". However, because "grain size" is an imprecise term, and because several metrics of snow microstructure play a role in radar backscatter, we will here simply refer to "snow microstructure" to encompass all measures of the snow medium; see Mätzler (2002) for a formal
description of the various microstructural quantities. The most important and objective measure of snow microstructure is the snow specific surface area (SSA), which is often defined as the surface area of the ice-air interface to the mass of a control volume of snow. SSA provides an objective measure that explains much of the microwave scattering processes. However, microwave interaction with microstructure cannot be entirely summarized by SSA; instead, radiative transfer is controlled by the spatial autocorrelation function (SAF) of the ice-air interface (Löwe and Picard, 2015). In the random medium model, the



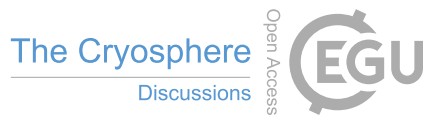

microstructure is described by a correlation function (Mätzler, 2002). In the bicontinuous medium model and the spherical
scatterers models with stickiness, spatial correlation functions (SAF) have also been derived and have been related to the grain
size and the aggregation parameter or the stickiness parameter (Chang et al., 2016). Thus, all microstructure characterizations
in electromagnetic models can be related through the SAF. The SSA is a measure of the SAF at only the shortest spatial lags
(e.g. ~10 µm); radar waves also respond to SAF at longer lags, (~100s µm). The SAF can be estimated using laboratory

methods, such as micro-CT. Grain shape can play an important role in scattering, especially for the cross-polarization radar
terms (Yueh et al., 2009). In summary, SSA has emerged as a practical and important microstructural property that both
explains much variability in microwave scattering and can be measured in the field.

### 4.2.2.2 Microstructure evolution schemes: how they work

SSA evolves based on well-understood physical properties, providing a source of information to better inform SWE retrieval

from Ku-band radar. Accurately modelling snow microstructure is not trivial, but decades of pioneering work by Colbeck
(1982), Sturm (1989), Brun (1989), Brun et al. (1992), and Jordan (1991), among others, led to the development of snow
metamorphism laws rooted in mass and energy conservation. The gravitational settling and metamorphism laws govern the
temporal evolution of snow microstructure and its mechanical (e.g., snow stratification and shear stresses) and thermal (e.g.,
albedo and emissivity) properties. The metamorphism laws describe three types of grain growth mechanisms including: kinetic

growth, equilibrium growth, and melt metamorphism (Lehning et al., 2002; Huang et al., 2012); kinetic and equilibrium growth
are sometimes called constructive and destructive metamorphism, respectively.

Destructive metamorphism describes movement of water vapor from small grains with high curvature to larger grains with
lower curvature. As a result, small grains and dendritic branches of large snow crystals evaporate and form larger and more
spherical crystals. Constructive metamorphism is primarily driven by temperature gradients within the snowpack, resulting in

direct vapor transport from warmer to colder surfaces. Sturm and Benson (1997) documented these processes in the Arctic.
The snowpack temperature gradient, in turn, is simply the difference in temperature between the ground and the air, divided
by the snowpack depth; typically snow covers insulate the ground, leading to soil being warmer than the ground. When snow
is wet, the growth rate accelerates, compared with dry snow conditions. These fundamental physical processes have been
explored for decades and solutions to the governing equations exist in a range of physical models.

### 4.2.2.3 Microstructure simulation accuracy

Snow microstructure model accuracy has improved significantly in recent decades, driven by improvements of field
measurements and model techniques. Morin et al. (2013) used objective field-based measurements of snow specific surface
area using the DUFISS instrument to evaluate the many-layer CROCUS snow model. Figure 12 clearly shows that the model
captured the seasonal evolution of SSA. The $r^2$ fit values between observations and simulations ranged from 0.6 to 0.74.



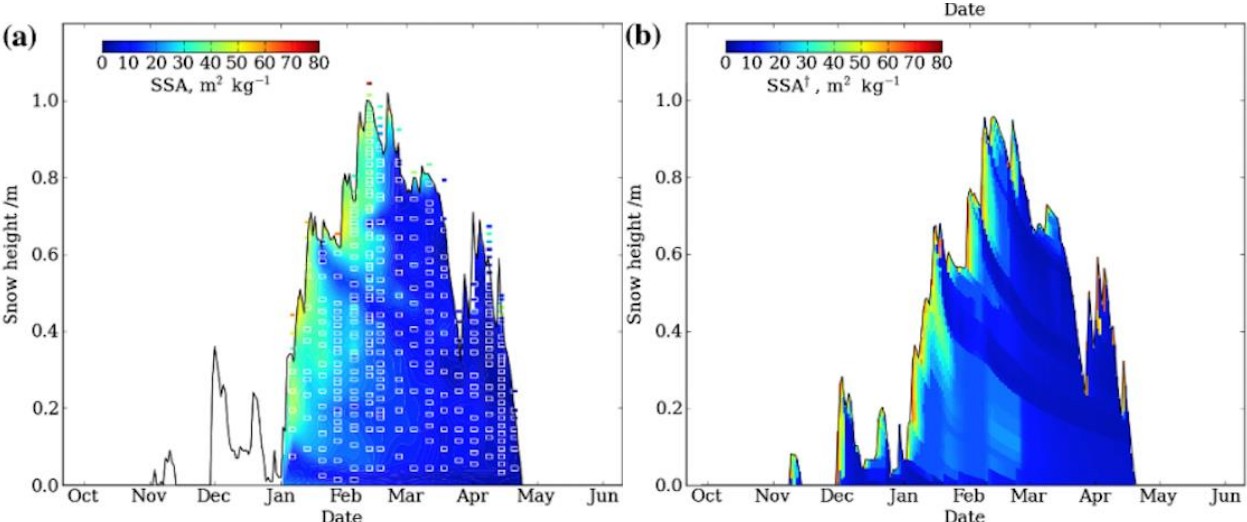


**Figure 12: Observed and interpolated SSA observations (a) and SSA simulations (b). From Morin et al. (2013). The small white marks are the observations.**

Use of microstructure simulations to constrain SWE retrievals require the additional step of coupling the microstructure simulation scheme and the RTM. Such studies have been explored more extensively in the context of passive microwave

remote sensing (Kontu et al., 2017; Langlois et al., 2012; Larue et al., 2018). Exploring the accuracy of a coupled radar backscatter and snow physics model is an area for future work.

**4.2.2.4 Implications for microwave remote sensing retrieval**

Given the advances in microstructure modelling skill, retrieval of SWE from radar backscatter stands to benefit from incorporation of prior information on snow microstructure provided by snow physics models. Prior information could be

provided in at least two ways. First, the radar backscatter could be assimilated directly into a coupled snow physics and RTM. This approach has been shown to be effective in assimilating passive microwave radiance. Assimilating backscatter has been demonstrated by Bateni et al. (2013, 2015) in the context of assimilating in situ backscatter observations. The second way that models could provide information to constrain SWE retrievals would be simply as a regularization term. In other words, the simulation model can produce a single estimate of a microstructure parameter such as SSA and the cost function could

incorporate this as *a priori* information in Bayesian retrieval (as shown in equation 3 above).

**4.2.2.5 Challenges and Future Work**

There are challenges associated with coupling snow physical models and RTMs. First, SWE simulations from snow physical models are often inaccurate, due to biases in the precipitation forcing. This happens mostly when information from weather stations is not available, and snow models must rely on atmospheric reanalysis data that are subject to biases, especially in the

precipitation amounts (Lindsay et al., 2014; Wrzesien et al, 2019b). Recent modelling work in Grand Mesa, CO shows however that biases in the simulations and analysis of higher resolution (~ 1km) numerical weather prediction models are significantly reduce leading to modelled SWE within ±10% of the observations (Cao and Barros, 2020). Changes in SWE have an immediate



effect on several other snow properties, particularly snow microstructure, a critical parameter in radiative transfer models. Second, in snow physical models, SWE is inversely related to the grain size. When SWE decreases, temperature gradients increase and grain growth is accelerated. In radiative transfer models, radar backscattering increase with SWE and with grain size. Backscattering is directly proportional to SWE and grain size. In SWE retrieval, SWE and grain size are independent parameters. When SWE increases and if the grain size stays constant, radar backscattering increases. However, in snow physical models, when SWE decreases, grain size increases and radar backscattering can increase or decrease due to the combined effects of SWE and grain size. This ambiguity emerges due to the nature of SWE, microstructure and backscatter relations. An ongoing study has found that even for small changes in simulated SWE (+/- 10%), snow microstructure is affected enough to mislead the retrieval algorithm and deteriorate the SWE retrievals even further (Merkouriadi et al., accepted). These challenges can be addressed by introducing appropriate physical constraints to the retrieval algorithms.

There are multiple areas where simulations must continue to improve, including demonstrating skill in the context of varying degrees of forest cover. Additionally, most simulations focus on estimation of SSA. While SSA has been shown to be an adequate summary of radiative transfer properties most of the time for visible and near-infrared remote sensing of clean snow, this is not the case in the microwave spectrum. To couple SSA to input to the RTMs, SSA must be related to SAF correlation length. Another subject of study is to use micro-CT to extract more information such as to relate to SAF. There are potentially useful synergies to be explored with retrieval of surface microstructure from visible and near-infrared measurements, in the context of radar retrievals of SWE. The retrieval algorithm shown earlier requires a classification rather than a precise value of grain size. Thus, the study of the error tolerance for estimations of SSA should also be pursued. Retrievals using a two-layer physics model are the next step (King et al., 2018). Significant effort will be needed to address the question of how many layers are necessary to capture small changes in snowpack backscatter behavior and SWE, and how to solve the inverse problem in multi-layered snowpacks the number of and vertical structure of which changes in time.

### 4.3 Solving the retrieval problem: three example algorithms

There are presently three physical model-based retrieval algorithms that have been applied successfully (Lemmetyinen et al., 2018; Zhu et al., 2018, King et al., 2019). All three algorithms utilize the frequency dependence of snow volume scattering for X-band and the two Ku-bands.

Lemmetyinen et al. (2018) first proposed a SWE retrieval scheme using both radar and radiometry measurements. The retrieval is based on the model of the expanded Microwave Emission Model for Layered Snowpack (MEMLS3&a) for simulation of both radar and radiometry observations (Proksch et al., 2015a). In the algorithm, the snow microstructure is represented by the effective correlation length, which is first retrieved with radiometry and in situ snow depth measurements. Then, retrieved correlation lengths are applied to constrain the SWE retrieval with radar observations. The algorithm has been validated with the Finnish NoSREx dataset.

King et al. (2019) proposed a SWE retrieval algorithm based on two-layer snow modelling for dual Ku-band radar observations. The two-layer model accounts for the small grain size of new fallen snow versus the larger grain size beneath





the new fallen snow; the forward physical based model of the retrieval is the SMRT model (Picard et al., 2018) for predictions of radar measurements. The ancillary configuration parameters and *a priori* parameters of snowpack for the retrieval are provided by the snow hydrology model and the snow physics model. The correlation length is based on *a priori* information from the snow physics model. The algorithm relying on the coupled snow physical model and SMRT  was applied successfully
to SWE retrieval from data taken in the tundra environment.

Zhu et al. (2018) developed the X- and Ku-band dual frequency radar SWE retrieval algorithm. The algorithm has three features: 1) surface scattering are subtracted from radar observations; 2) a parameterized bicontinuous-DMRT model derived from regressions is developed to simplify the retrieval with only two unknown parameters: scattering albedo and optical thickness, which are related to the snow depth and the grain size; and 3) classification of snowpack into two classes (low
albedo and high albedo) to mitigate the non-unique inversion problem. The algorithm has been applied and validated with three sets of airborne SnowSAR data. Two datasets are from the 2011 and 2012 campaigns in Finland (Meta et al., 2012; Chang et al., 2014) and the third dataset is from the 2013 campaign in Canada (King et al., 2018). Recently, the *a priori* estimate of scattering albedo has also been based on passive radiometry measurements (Zhu et al., accepted) following the earlier work of Lemmetyinen et al. (2018).

**4.4 Retrieval results**

In the Zhu et al. algorithm (2018), the co-polarization of X-band (9.6 GHz) and Ku-band (17.2 GHz) are utilized. Thus, the cost function is:

$$\text{F} = \text{MIN}\left\{\frac{w_X}{2s_X^2}\left(\sigma_{X,VV}^{obs} - \sigma_{X,VV}^{obs,bg} - \sigma_{X,VV}^{model}(\tau_X,\omega_X)\right)^2 + \frac{w_{Ku}}{2s_{Ku}^2}\left(\sigma_{Ku,VV}^{obs} - \sigma_{Ku,VV}^{obs,bg} - \sigma_{Ku,VV}^{model}(\tau_X,\omega_X)\right)^2 + \frac{w_p}{2s_p^2}(\omega_X - \bar\omega_X)^2\right\} \quad (4)$$

where the two unknown parameters to be retrieved are the scattering albedo at X-band ($\omega_X$) and the optical thickness at X-
band ($\tau_X$). The input parameters are: $\sigma_{X,VV}^{obs}$ and $\sigma_{Ku,VV}^{obs}$ are radar observations at X and Ku band, $\sigma_{X,VV}^{obs,bg}$ and $\sigma_{Ku,VV}^{obs,bg}$ are background scattering, and $\bar\omega_X$ is a priori scattering albedo. Background scattering can be obtained by radar observations under snow free conditions or X band observations at small SWE. It is also possible to apply other observations at low frequencies such Sentinel 1 as mentioned in section 3.1.2 to refer background scattering at X and Ku bands. In the retrieval, $\bar\omega_X$ is based on a binary choice. For snowpack with large SWE and small albedo, $\bar\omega_X = 0.65$. Otherwise, $\bar\omega_X = 0.35$. If better estimation
of $\bar\omega_X$ can be obtained by ground measurements, snow physical model (Xiong and Shi, 2017), or passive observations (Zhu et al., 2021), the retrieval performance can be improved. $w_X$, $w_{Ku}$, $w_p$, and $s_p$ are user defined parameters. $s_X, s_{Ku} = 0.5\ dB$ are instrument parameters. In this retrieval, both SWE ($\tau_X$) and scattering albedo $\omega_X$ are retrieved.





### 4.4.1 Steps of Retrieval algorithms

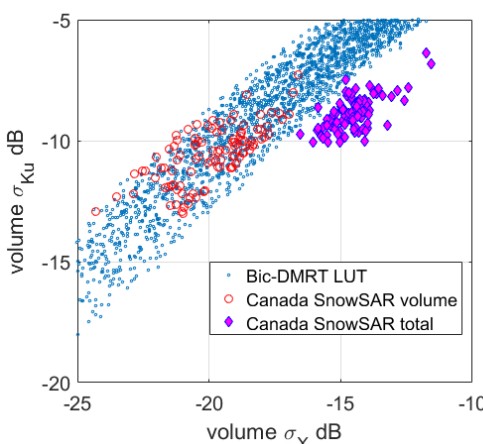

**Figure 13: Canadian SnowSAR measurements compared with the Bic-DMRT LUT. Figure from Zhu et al. (2018).**

In step 1, the rough surface contributions are subtracted from the radar measurements. Subtraction of rough surface scattering of the snow/soil interface improves the accuracy of SWE retrieval (Cui et al., 2016; Xiong and Shi, 2017; Zhu et al., 2018). An example is given with the SnowSAR data collected from the 2013 Canadian TVC campaign (King et al., 2018; Zhu et al., 2018) in Figure 13. SnowSAR flights were performed on April 8 and 9, 2013 crossing each of the TVC sites. Since there are no measurements on the underlying soil in the 2013 Canadian TVC campaign, rough surface scattering are determined by differences between radar observations and snow volume scattering calculated based on snowpit measurements. The SAR data show a bias when compared with X- and Ku-band DMRT simulations. The bias is attributed to the contributions from surface scattering. The bias in the Ku-band data are 1 to 2 dB while, in the X-band data, the bias is about 4 to 6 dB, which indicates that the surface scattering has more influence at X-band. After subtraction of surface scattering from the SnowSAR data, the data fall into the range covered by the DMRT simulations. The dynamic range of SAR data is also larger, meaning that the obtained volume scattering components have better correlations with the SWE.

In step 2, regression training is used to reduce the number of unknowns (Cui et al., 2016; Zhu et al., 2018; Zhu et al., accepted). The look-up table (LUT) based on DMRT simulations for snow volume scattering is generated for various snow properties of depth, density, and snow microstructure. The output-dependent variables for the LUT are backscatter ($\sigma_X$ and $\sigma_{Ku}$), the effective single scattering albedo ($\omega_X$ and $\omega_{Ku}$), and optical thickness ($\tau_X$ and $\tau_{Ku}$) for X- and Ku-bands. Regression trainings are performed for $\omega_X$ versus $\omega_{Ku}$ and $\tau_X$ versus $\tau_{Ku}$ to utilize the frequency dependence of scattering between the X- and Ku-bands.

In Zhu et al. (2018), there are two trained relations. The first regression trained relation for volume scattering is $\sigma = A + B \log(\sigma^{1st})$, where $\sigma$ is total volume scattering of snow and $\sigma^{1st}$ is the first-order scattering. The coefficients A and B are given in Zhu et al. (2018) for X- and Ku-bands. In the second trained relations, the relations between albedo at X- and Ku-band and between optical thickness at X- and Ku-band are trained to give $\omega_{Ku}(\omega_X)$ and $\tau_{Ku}(\tau_X)$. After these two sets of



training, we have $\sigma^{1st} = 0.75cos\theta_t\omega(1 - \exp{(-2\tau/cos\theta_t)})$ (Cui et al., 2016; Zhu et al., 2018) and the regression relations,

$\sigma_X = A_X + B_X \log\left( 0.75cos\theta_t\omega_X(1 - exp(-2\tau_X/cos\theta_t)) \right)$  and  $\sigma_{Ku} = A_{Ku} + B_{Ku} \log\left( 0.75cos\theta_t\omega_{Ku}(\omega_X)(1 - exp(-2\tau_{Ku}(\tau_X)/cos\theta_t)) \right)$. Thus, the volume scattering $\sigma_X$ and $\sigma_{Ku}$ depends only on two parameters, $\omega_X$ and $\tau_X$. The approach is labelled a "parameterized bicontinuous-DMRT model". We apply two channel observations to retrieve the two parameters, $\omega_X$ and $\tau_X$, from which the SWE is obtained by the relation SWE = $a(1 - \omega_X)\tau_X$, where a = 9745 for SWE retrieval (Zhu et al., 2018).

In step 3, *a priori* estimates of parameters and classification are applied to determine the mean, the weight, and the uncertainty $\frac{w_p}{2s_p^2}(\omega_X - \bar{\omega}_X)^2$ of the cost function. Several methods have been applied to determine *a priori* parameters of snow microstructure. *A priori* information are obtained from co-located field stations or historical ground measurement data (Rott et al., 2010; Cui et al., 2016; Zhu et al., 2018). In the study of Xiong & Shi (2017), snow physical models are applied to derive *a priori* information. *A priori* information have also been derived from passive observations (Lemmetyinen et al., 2018; Zhu

et al., accepted). In parametrizing microstructure, the snow grain size, correlation length (Proksch et al., 2015a) or scattering albedo (Cui et al., 2016; Zhu et al., 2018) are used.

To mitigate the non-uniqueness problem in retrieval, classification is also performed. We use the dataset from the Canadian SnowSAR 2013 campaign as an example. The *a priori* information are obtained from co-located ground measurements. The backscattering $\sigma$ are classified into two groups: snowpack with high albedo and snowpack with low albedo. The threshold

$\omega_X = 0.49$ is the average of all *a priori* albedos. Figure 13 shows that the sensitivities of backscatter to SWE are enhanced by classification of snowpack based on *a priori* scattering albedo. The classification scheme accounts for the heterogeneity of snow cover from location to location.

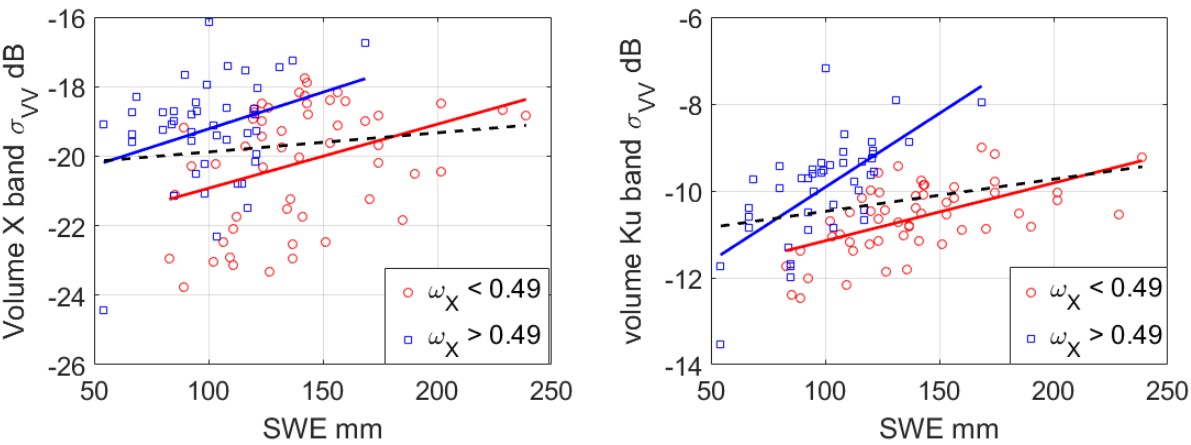

**Figure 14: volume scattering of the Canada SnowSAR data as a function of SWE based on the retrieved scattering albedo $\omega_X$ at (a,**
**left) X-band and (b, right) Ku-band. Radar observations are from a single flight. *A priori* estimates are used to classify $\omega_X$ into two classes. They are plotted by different coloured markers. The black dashed curve is the regression line of all Canada SnowSAR data. The blue and red solid curves are the regression lines for backscatter with $\omega_X$ larger and smaller than 0.49, respectively. Figure from Zhu et al. (2018).**



### 4.4.3 Retrieval results

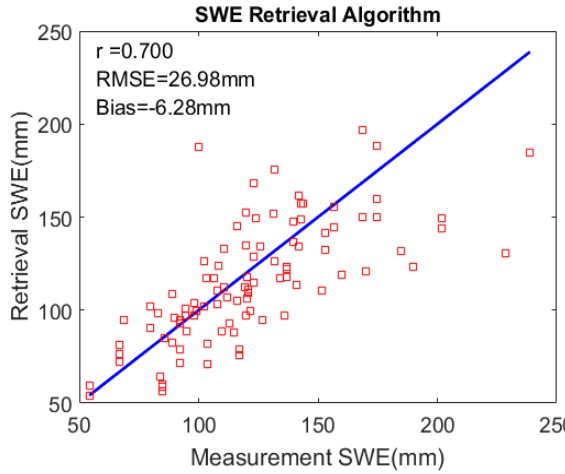
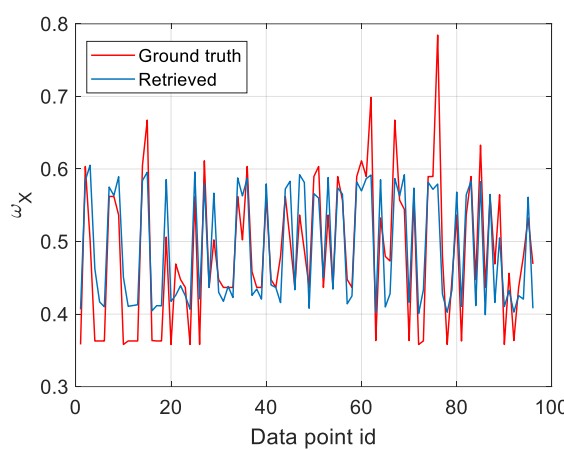

**Figure 15: (a, left) SWE retrieval performance and (b, right) Scattering albedo retrieval using Canadian SnowSAR 2013 X- and Ku-band radar data. The figures are from Zhu et al. (2018) and Zhu, 2021.**

In step 4, the cost function of equation (4) with least squares is applied to the SAR data at X- and Ku-bands in Figure 15. In this example, $s_x$ is set to 0.1, and the weight factors are set to be unity. Figure 15 (a) shows the performance of the retrieval algorithm. The retrieval results have RMSE of ~27 mm of SWE, a correlation of 0.7, and a bias of 6.3 mm. The results show that the retrieval algorithm is particularly successful for SWE values below 150 mm. It also satisfies requirements from the CoREH2O are the RMSE is less than 30 mm for SWE below 300mm and the RMSE is less than 10% of SWE for SWE above 300 mm. In Figure 15 (b), the retrieved scattering albedo $\omega_X$ are in good agreement with albedo derived from ground measurements (grain size or correlation length). It indicates the snowpack has two types: larger SWE and small albedo and small SWE and large albedo.

In the equation of cost function, the weight $w_x$ is the measure attached to the *a priori* parameter. In the example of Figure 15, $w_x$ is set to be 1. In Zhu et al. (2018), we have studied the retrieval performance variation by reducing $w_x$. The performance is still acceptable down to $w_x = 0.14$ with RMSE equal to 40 mm. Next, we study the performance with variations in surface scattering subtractions was also studied by adding noise to the best guess of surface scattering in Zhu et al. (2018). At the best guess with 0% noise, the RMSE of SWE is 27 mm. The RMSE of SWE retrieval increases with noise increase. However, even with 50% additional noise (absolute error of 3 dB in surface scattering), the performance is with RMSE of 45 mm.

### 4.4.4 Future work

In the Zhu et al. (2018) algorithm illustrated above, a single layer is used. A multilayer snow model should also be considered. Studies have already been conducted using a two-layer model (King et al., 2018; King et al., 2019; Rutter et al., 2019) to account for small grain size in the upper layer and a larger grain size for the bottom layer (depth hoar). Accuracy requirements of *a priori* estimates to achieve desired retrieval skill have not yet been quantified. Future work should include systematic



usage of cross polarizations in the retrieval algorithms. The inclusion of cross polarizations is important for deep snow layers as the co-polarization K-band backscatter saturate for SWE larger than 300 mm.

## 5 Improving SWE retrieval estimations via synergy with other datasets

In satellite remote sensing, with the vast amount of satellite data, there can be data fusion of synergistic use of other data sets to refine the retrieval algorithms and improve the SWE estimations. The combined active and passive microwave remote sensing using data of the same frequencies has been an active area of research. Recent work shows the use of C-band Sentinel-1 data to retrieve SWE for deep snow layers. Interferometry and tomography are also studied for future launches of complementary missions. P-band GNSS-R has also been proposed for satellite retrieval of SWE.

### 5.1 Passive microwave

GCOM (Global Change Observation Mission) was launched in 2012 and carries the AMSR2 (Advanced Microwave Scanning Radiometer 2) instrument, measuring microwave brightness temperatures at 6 frequencies. The channels at 18.7 GHz and 36.5 GHz with both V and H polarizations have been used to retrieve global SWE. The spatial resolutions are coarse and are respectively at 22 km by 14 km for 18.7 GHz and 12 km by 7 km for 36.5 GHz. With overlapping footprints, interpolations

algorithms have been applied (Long and Brodzik, 2016) to downscale resolution to 3 km. Uncertainty increases when downscaling resolution increases. Because the passive emissivity is related to the radar bistatic scattering (Tsang et al., 1982), the brightness temperatures are related to the radar backscattering cross sections. Since SAR observations (< 500 m) have much finer spatial resolutions than passive radiometry, a synergy is to use SAR at the X- and Ku-bands to downscale coarse resolution passive data or SWE products (Takala et al., 2011) to finer spatial resolutions using data assimilation or other

algorithms.

Several other methods have been explored to combine the active and passive microwave observations to retrieve SWE. In Hallikainen et al. (2003), satellite microwave observations were used to demonstrate its feasibility to retrieve SWE by using passive-only, active-passive, and active-only algorithms in sub-arctic snow in Finland. Tedesco and Miller (2007) evaluated SWE retrieval performances by using active (Ku-band) and passive (X-band) microwave observations. Recently, Bateni et al.

(2015) conducted SWE retrieval studies using passive (Ku- and Ka-bands) and active (L- and Ku-bands) ground-based microwave observations. The SWE retrieval was obtained within a data assimilation framework by comparing simulated microwave observations against the corresponding observations at multiple frequencies.

In the study of Lemmetyinen et al. (2018), the correlation length of snowpack is derived by matching both active and passive microwave observations against the simulations of the MEMLS3&a model (Proksch et al., 2015a) with ancillary data from

snowpit measurements and weather stations. Next, the correlation length is used for the active radar algorithm to retrieve SWE. The derived correlation length from both active and passive data demonstrates an improvement of the SWE retrieval performance over the SWE retrieval with the active-only algorithm. Cao and Barros (2020) used a multilayer snow hydrology



model coupled to MEMLS3&a to investigate the signature of the variability of snow physics on the microwave behavior at seasonal scales. They ound that a combined approach using active microwave sensing in the accumulation season and passive

sensing in the melting season would yield in the best sensitivity to capture the temporal evolution of seasonal SWE by taking optimal advantage of microwave hysteresis (Ulaby et al., 1981, Kelly and Chang, 2003). The generalization of this approach to available active and passive microwave measurements with large resolution gap poses a significant challenge

Recently, passive observations have been used to enhance the performance of the active algorithm (Zhu et al., accepted). The active-only algorithm was described earlier by using a cost function between simulated and observed radar observations as a

function of the two parameters of scattering albedo and optical thickness. In this enhancement, passive observations at Ku- and Ka-bands at the collocated and coincident snow scene are used to determine the range of the scattering albedo. The bicontinuous DMRT model is applied for both passive and active model simulations. X- and Ku-band radar data are then used with the determined scattering albedo to obtain SWE. This active and passive combined method is applied to the NoSREx dataset. Comparison statistics show that the combined active-passive method has improved performance over the active-only

method. The retrieval does not require *a prior* information and ancillary data from ground measurements.

## 5.2 C-band SAR

C-band radar is typically used to detect wet snow by the strong decrease in backscatter (Shi et al., 1995; Baghdadi et al., 1997; Koskinen et al., 2009; Bateni et al., 2013; Nagler et al., 2016; Marin et al., 2020). However, the use of C-band for SWE retrieval has not been investigated systematically until recently. The snow grain sizes of 1 mm are more than 50 times smaller than the

C-band wavelength of 5.5 cm, so the volume scattering by snow grains was believed to be small at C-band. Based on Radarsat and ERS observations, Shi et al. (2000) and Pivot et al. (2012) showed a significant increase in co-polarized $\sigma^0$ (several dB) in areas where snow volume scattering was dominating over ground surface scattering. Arslan et al. (2006) revealed a stronger increase in cross-polarized than in co-polarized $\sigma^0$ from airborne data. Radiative transfer modelling at C-band remains relatively unexplored. Kendra et al. (1998) applied RTE to simulate the co- and cross-polarized $\sigma^0$ response to artificial snow.

Veyssière et al. (2019) applied the MEMLS3&a model to simulate Sentinel-1 (S1) observations, showing limited impact of SWE on co-polarization, a result confirmed by Cao and Barros (2020).

Lievens et al. (2019) developed an empirical change detection algorithm to retrieve snow depth at 1-km spatial resolution from C-band S1 observations over all northern hemisphere mountain ranges. The retrievals are based on the reasoning that 1) an increase in snow depth causes an increase in snow volume scattering in both co- and cross-polarized $\sigma^0$,

2) the snow volume scattering contribution in cross-polarization is comparable to or larger than the ground surface scattering contribution, 3) the ground surface scattering contribution remains relatively constant due to the insulating properties of snow, and 4) the ground scattering will decrease with increase of snow depth due to snow attenuation. It is hypothesized that use of a ratio between cross- and co-polarization $\sigma^0$ observations can reduce the impacts of ground, vegetation, and surface geometry properties, and also enhance the sensitivity to snow depth (Bernier et al., 1999). With algorithm improvements, retrievals were

performed at sub-kilometre (100 m, 300 m, and 1 km) resolutions over the European Alps in Lievens et al. (submitted), and





below, 300-m retrievals are shown for Idaho, Montana and Wyoming, US. Figure 16 (a) shows a density plot, comparing weekly S1 retrievals for the periods between August and March from 2017 to 2020 with in situ measurements from 203 available SNOTEL sites within the study region. The spatio-temporal Pearson correlation between retrievals and measurements is 0.81 and the mean absolute error (MAE) is 0.27 m. Figure 16 (b) shows the MAE relative to the snow depth (in %),

illustrating that for shallow snowpacks (< 1 m) The MAE is on average ~ 50% with large variance. From a 1-m depth onwards, the relative average MAE remains constant at ~30%. Figure 16 (c) compares time series of snow depth retrievals and measurements for six selected SNOTEL sites. The ESA Sentinel-1 mission is the first C-band radar constellation that measures consistently (not regularly tasked) with the exact same orbit revisited every 6 days at different times of day. Such consistent observation scenario benefits the change detection approach and could explain why limited success has been found with the

CSA Radarsat C-band system, which must be tasked.

The recent snow depth retrieval results from C-band SAR over mountainous regions with deep snow indicate a strong complementarity with applications at higher frequencies – for example, Ku-band is much more sensitive to shallow snow, while C-band performs better in relative terms for deep snow.

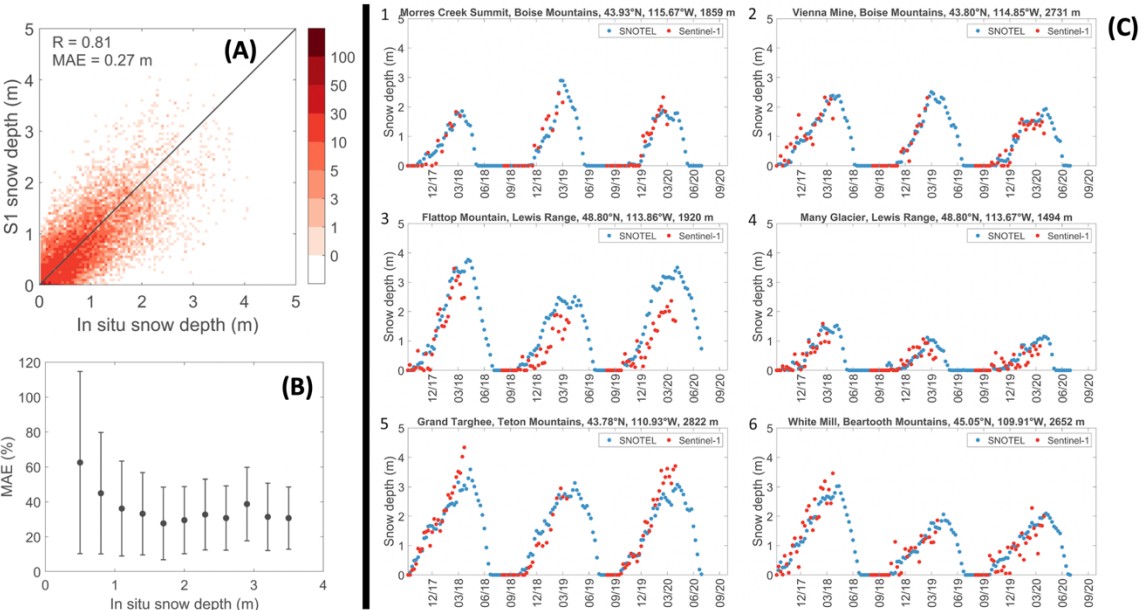

**Figure 16: Snow depth retrieval from Sentinel-1, 2017-2020 in Idaho, Montana, and Wyoming. (a) Density plot of weekly 300-m resolution S1 snow depth retrievals, and coincident SNOTEL observations from 203 sites. (b) The corresponding mean and standard deviation of the absolute error between S1 retrievals and in situ measurements, stratified by the measured snow depth. (c) Time series comparison between snow depth from S1 and SNOTEL in situ measurements at six locations for three winter seasons. Note that the analysis includes snow-free conditions but excludes wet snow conditions as detected with the S1 algorithm (Lievens et al.,**
**submitted).**

Despite the observational evidence of C-band sensitivity to deep snow, the exact underlying physical mechanisms are still not fully understood. Snow crystals can form larger-scale clusters that are more similar in size compared to the C-band wavelength. Snow crystals are highly anisotropic (irregular in shape), that, in principle, results in a stronger scattering in cross-polarization.





Other contributions to snow scattering at C-band that can be investigated are larger-scale contrasts in snow density (incl.
between snow layers) and the formation of ice layers with rough boundaries in deep snowpack. Recent progress in snow
radiative transfer modelling is addressing the anisotropic shape of snow crystals and the clustering of crystals (Zhu et al.,
accepted) using a bicontinuous model that introduces a two-parameter correlation function (Ding et al., 2010). Initial results
indicate that a situation with large clustering of grains can cause cross-polarization signals from volume scattering that are in
the range observed by S1, and that the volume scattering signal at cross-polarization can dominate over surface scattering for
large snow depths. This occurs for a clustering parameter $b = 0.4$. However, previous airborne (Yueh et al., 2009; Zhu et al.,
2018) and tower-based measurements (TomoSAR, NoSREx) at the X- and Ku-bands are better explained by a clustering
parameter of $b = 1.0$ to $2.0$ (Chang et al., 2014; Tan et al., 2015; Xiong and Shi, 2019). It is likely that the clustering parameter
changes with snow conditions, and in particular compaction mechanisms. More work is needed to improve our understanding
of the physical mechanisms that cause C-band sensitivity to snow mass.
Manickam and Barros (2020) showed that Sentinel-1 backscatter exhibit different scaling behavior depending on land-cover
and landform and specifically topography that is dynamic at sub-seasonal scale reflecting changes in weather. Specifically
they identified an ubiquitous scaling break at scales of 180-360 m that separates the small-scale range with invariant single
scaling from scaling highly sensitive to snow wetness at larger scales. They found similar results for L-band UAVSAR data.
This scaling behavior should be very useful to integrate multifrequency measurements at different resolutions both for
upscaling and downscaling applications (e.g. Bindlsih and Barros, 2002; Kim and Barros, 2002)

### 5.3 Phase-based approaches

Approaches to estimating snow characteristics using microwave remote sensing almost invariably make use of some
combination of the radar cross section $\sigma^0$, sensitivity to polarization, and frequency dependence; the techniques previously
mentioned are all based on microwave scattering, which focus on the amplitude response. Approaches that use the signal phase,
related to the electrical path length and time-of-flight information, while more challenging, can provide additional information
about the snowpack, which is synergistic with the Ku-band backscatter approach that is the focus of this paper. These
approaches provide independent information about snow properties through travel-time through layers, and information about
where within the snowpack the major sources of amplitude (e.g. layer boundaries) are originating. Three techniques are
described: 1) ultra-wideband radar, 2) tomography, and 3) interferometry.

### 1000 5.3.1 Ultra-wideband radar

The range resolution of a radar system is inversely proportional to the bandwidth. Ultra-wideband radar can be used to resolve
reflections from different depths within the snowpack. This not only provides insight into which locations within the snowpack
are contributing most to the backscatter but also estimates the depth-dependent refractive index of snow, which can be used to
independently estimate snow depth, SWE, and stratigraphy. However, the ultra-wideband radar technology does not have a
straightforward path to space, due to bandwidth and frequency allocation limitations. Ultra-wideband approaches have been





demonstrated for decades from the ground, using broadband radar at nadir incidence angles to estimate depth and SWE (for a review, see Marshall and Koh, 2008), using L-band GPR systems (Lundberg et al., 2000; McGrath et al., 2019) and at higher microwave frequencies (Gubler and Hiller, 1984); multi-channel L-band radar has recently been used to map density profiles in polar firn (Meehan et al., 2021). Nadir ultra-wideband FM-CW radar (2-18 GHz) has been flown from an aircraft platform

with success in the polar regions as part of NASA Operation IceBridge, mapping snow depth on sea ice and stratigraphy in firn on the ice sheets (Panzer et al., 2013; Arnold et al., 2019). More recently, airborne FM-CW experiments have been done over seasonal snow in the mountains (Yan et al., 2017), with additional efforts being carried out at the University of Alabama in developing airborne tomography mountain applications (Taylor et al., 2020).

### 5.3.2 Tomography

Recently, the Synthetic Aperture Radar (SAR) Tomography (TomoSAR) has been used in monitoring the snowpack at frequencies from X- to Ku-band (Wiesmann et al., 2019; Rekioua et al., 2017; Xu et al., 2018). This technique provides unique access to the structure of the imaged scene, and in the case of the snowpack, enables the separation of multiple snow layers as well as the detection of and compensation for soil and vegetation layers. The addition of polarimetric capabilities brings in the ability to detect spatially varying shapes, sizes, and permittivities, to decompose the backscattered signal into volumetric and

surface scattering components, and to distinguish between snow, soil, and vegetation. Some ground-based field experiments have been carried out that demonstrate the focused image recovery of the layering structure of the snowpack with different densities. In Figure 17, the results of tomography carried out at three frequencies in the X- (9.6 GHz) and Ku-bands (13.5 GHz and 17.2 GHz) and for co- and cross-polarization are shown where a layered structure is detected. As the frequencies increase from the X- to Ku-bands, the volume scattering from the snow becomes more prominent. The coherency distribution also

corresponds to the vertical profile of the snow density and snow grain size. The ability to identify snow layers and density changes are expected to significantly improve SWE retrieval.

The tomographic results shown in Figure 17 were achieved using an FM-CW radar measuring the complex scattering coefficient at multiple incidence angles, and frequencies. By using the frequency and angular-correlation functions (Zhang and Tsang 1998; Tsang and Kong, 2001) of the complex amplitudes, time domain back projection is used to obtain the tomograms.

To gain insight of the scattering within a complex snowpack such as shown in Figure 17, 3D full waveform simulations are necessary for analysis. Because of the complexity of such analyses and the need for additional data, the work in modelling and producing tomographic results is ongoing.


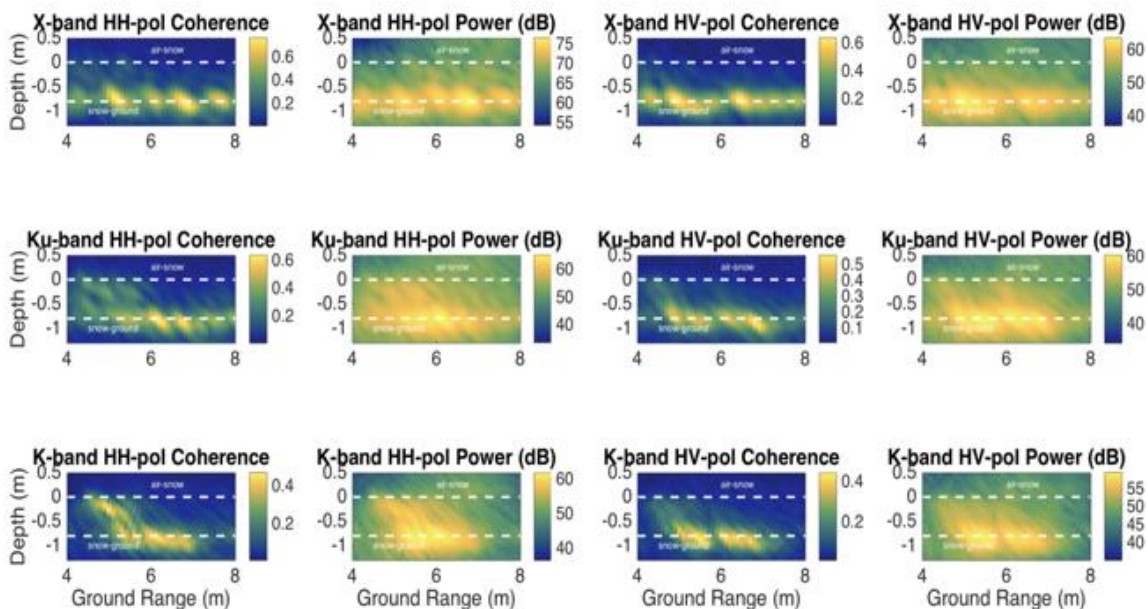

**Figure 17: TomoSAR images with co- and cross-polarization for (a) 9.6 GHz, (b) 13.5 GHz, and (c) 17.2 GHz.**

### 5.3.3 Interferometry

Interferometry can be used to measure differential path length and changes in the electromagnetic path length in signals interacting with a snow pack. Depending on the wavelength, polarization, and scattering characteristics of the air/snow, snow volume and snow/soil interfaces, the interferometric response will vary (Deeb et al., 2011; Lei et al., 2016). Hence, from a remote sensing point of view, observations of these complementary microwave measures of the snowpack can be the source of new algorithms that make use of this enhanced data set. As a result, estimates of depth, density, and SWE are all being actively explored. Interferometry can be performed either between two observations separated in space (Figure 18 (a)), or by repeat observations after changes in snow occur (Figure 18 (b)). Although phase measurements are always made modulo $2\pi$, there are a few methods, including phase unwrapping and the use of a surface DEM, that can be used to resolve the ambiguity. For standard cross-track interferometry, using a high microwave frequency or operating in wet snow where the penetration depth would be limited, we can assume that most of the signal comes from the snow surface, and the snow surface topography can be mapped. When this is differenced from a snow surface at a different time, a snow depth change can be estimated spatially. If snow depth can be estimated using a different modality (e.g. with lidar), the penetration depth would yield additional information about the snow layer.

The implementation of interferometry is achieved through the introduction in the system of a second antenna separated from the transmitting antenna by the baseline B. Such an antenna can be passive, receiving the signal of a common transmitter antenna, and the phase difference between the two antennas is measured. The advantage of such a system is that a suite of



additional measurements to the radar cross-section, polarization characteristics, and frequency dependence can be made that are relevant to the snowpack characteristics. These additional measures are the height of the scattering phase centre and the interferometric correlation magnitude, which are both sensitive to the frequency and polarization combinations as well as the

snowpack physical characteristics.

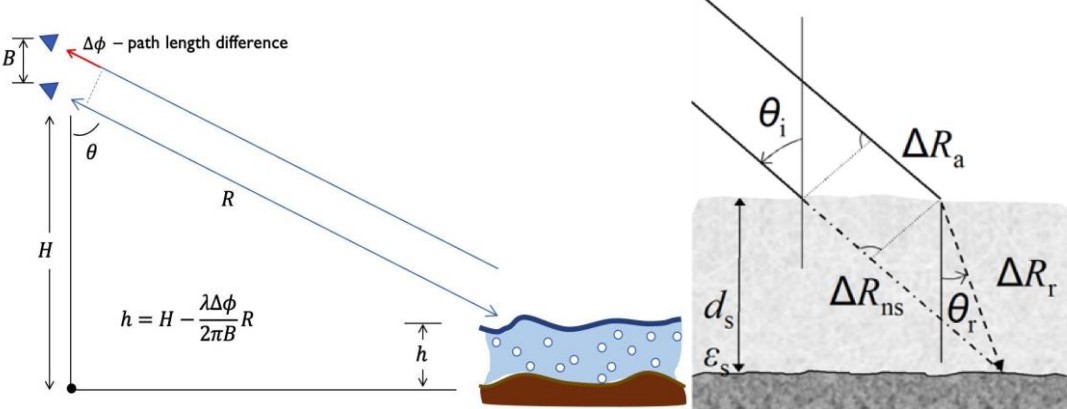

**Figure 18: (a) Illustration of an interferometric viewing geometry for a vertical baseline of length, B. Using a measured path length difference measured by the interferometric phase, $\Delta\phi$, simple trigonometric calculations are used to estimate a precise value of the look angle, $\theta$, and consequently the relative height, h, of the scattering phase center compared to the reference height, H. Calculations**
**of differential phase such as those shown above are modulo $2\pi$, and hence some care should be taken to correctly estimate the absolute phase difference. (b) When the phase centre is below the snow surface, refraction in snow must also be considered. At low microwave frequencies (e.g. L-band) in dry snow, the phase centre is often the snow-ground interface, and due to the slower electromagnetic velocity in snow, phase changes are related to changes in snow depth and density.**

To understand the source of the additional measures shown in Figure 18 (a), the observed electric field by each of the antennas
of the interferometric pair, written as E1 and E2, can be formed to estimate the complex interferometric correlation, $\gamma$, as:

$$\gamma = |\gamma|e^{j\Delta\phi} = \frac{\langle E_1 E_2^* \rangle}{\sqrt{\langle |E_1|^2 \rangle \langle |E_2|^2 \rangle}} \quad (5)$$

where the bracket symbols $< >$, indicate a spatial average and the superscript * indicates a complex conjugate. The phase indicated in the equation is related to the height of the scattering phase centres, averaged over looks, whereas the magnitude indicates the consistency of those phase centres. It can be shown that the relationship between the complex interferometric
correlations shown above is related to the Fourier transform of the extinction weighted radar cross section of the surfaces and volume components of the snow layer (Treuhaft et al., 1996; Treuhaft and Siqueira, 2000).

In the second approach, a low microwave frequency in dry snow can be used, where we can assume most of the signal comes from the snow-ground interface (Figure 18 (b)). In this case, changes in the time of flight, or electromagnetic path length, to the snow-ground interface are caused by changes in snow depth and SWE. A time series of InSAR observations at L-band can
thus potentially be used to estimate changes in snow depth and SWE. This technique was tested during the NASA SnowEx 2020 and 2021 experiments (Marshall et al., 2021). In addition, a similar approach with P-band has been demonstrated from tower-based platforms, using existing transmitted signals (Shah et al., 2017). This bistatic approach shows great promise for a much lower cost satellite system, as only a receiver is required.



At the lower frequencies of P-, L-, and C-band, interferometric observations such as these have been used for characterizing

the volume scattering components of forests (Kugler et al., 2015). Like the topography approach discussed above, when

multiple baselines are used, a vertical profile of the volume density can be estimated (Reigber and Moreira, 2000; Tebaldini

and Rocca, 2011), thus providing an even fuller picture of volume stratigraphy. While most of this development has been done

at low-frequencies for vegetation, a scaling of the volume characteristics of snow with frequency has been equally promising

(Lei et al., 2016), especially at the Ku- and Ka-bands, where the wavelengths at these frequencies are on a similar order of

magnitude as the snow volume scattering components. With the interferometry set up as described in Figure 18 (a), we can

explore possible configurations, such as launching a companion satellite only to receive the complex scattered electric field

signal from the snow medium. Such a companion would be able to complement baseline observations with additional measures

of reflectivity, interferometric phase, and correlation magnitude that could all be used to better characterize the snow volume.

## 6 Planning a Satellite Mission

The earliest synthetic aperture radar (SAR) satellite missions were focused on C-band (Envisat, Radarsat-1). Missions

developed during recent decades extended the frequency range to L- (e.g. PALSAR) and X-band (e.g. TERRASAR-X), with

planned missions at the P- (ESA-Biomass), and L- and S-bands (NASA-ISRO SAR mission). Spaceborne scatterometer

missions such as QuikScat have illustrated the wide-ranging contributions of measurements at Ku-band to applications

spanning the cryosphere (Kwok, 2007; Swan and Long, 2012), biosphere (Frolking et al., 2006), ocean (Bourassa et al., 2010),

and Ku-band cloud radar measurements have made significant contributions to precipitation-related fields through missions

like CloudSat (Stephens et al., 2002) and GPM (Skofronick-Jackson et al., 2017). To date, there have been no SAR missions

at Ku-band.

The potential for Ku-band radar to retrieve snow water equivalent (SWE) was explored as part of the NASA Snow and Cold

Land Processes Mission and supporting Cold Land Processes Experiment (Yueh et al., 2009; Cline et al., 2009). The ESA

COld REgions Hydrology High-resolution Observatory (CoReH2O; X- and Ku-bands) completed Phase A in 2013, but was

not selected for implementation (Rott et al., 2010). CLPX and CoReH2O science and mission development activities played a

major role in motivating the significant progress achieved over the past decade in measuring, understanding, and modelling

the Ku-band radar response to SWE, snow microstructure, and snow wet/dry state (as assessed in previous sections of this

review). Given this collective progress, satellite mission concept reviews subsequent to the completion of CoReH2O Phase A

(e.g. the ESA 'SnowConcepts' project completed in 2018; a Payload Analysis and Trade-off Study for snow mass funded by

the Canadian Space Agency completed in 2017) further emphasized the potential for Ku-band SAR measurements to address

a broad set of user requirements related to seasonal snow mass.

Supported by these studies and continued analysis of experimental ground-based and airborne campaigns (See Section 4; King

et al., 2018; Lemmetyinen et al., 2018; Zhu et al., 2018), a Ku-band SAR mission was selected as the most feasible approach

to meet the operational requirements (wide swath/rapid revisit/short latency) of Environment and Climate Change Canada



(ECCC) for spaceborne measurements sensitive to SWE. Since 2018, ECCC and the Canadian Space Agency (CSA) have partnered to advance the scientific and technical readiness of the "Terrestrial Snow Mass Mission" (TSMM), a Ku-band radar satellite with the primary science objectives focused on the provision of climate services related to seasonal snow, and improved operational environmental prediction including streamflow. The TSMM development effort is underpinned by

engagement with Canadian industry, academia, and international partners.

**Table 5: Summary of CoReH2O and TSMM missions.**

|  | CoReH2O | TSMM |
|---|---|---|
| **Frequencies** | X band 9.6 GHz<br>Ku band 17.2GHz | Ku band 13.5GHz<br>Ku band 17.25GHz |
| **Polarizations** | VV, VH | VV, VH |
| **Level 1 SAR image resolution** | 50m | Low resolution: 250m<br>High resolution: 50m |
| **Spatial Resolution of product** | 100m-500m | 500m |
| **Temporal** | 3 days repeat for Phase 1 | 5 days, Canada and other snow-covered area |
| **Accuracy Requirements in SWE** | 3 cm RMSE, SWE <30 cm<br>10% for SWE > 30 cm | 3 cm RMSE (non-alpine)<br>25% (alpine) |
| **NES0** | X band<br>VV <-23dB & VH <-28dB<br>Ku band<br>VV<-29dB & VH<-25dB | 13.5 GHz<br>VV&VH < -26 dB<br>17.2 GHz<br>VV&VH < -25 dB |
| **Accuracy in $\sigma_0$** | Stability <0.5 dB<br>Abs. accuracy <1 dB | Stability <0.5 dB<br>Abs. accuracy <1 dB (13.5 GHz) and <0.5 dB (17.2GHz) |
| **Incident angle** | $30 - 45°$ | $23 - 50°$ |

Because TSMM is the first spaceborne Ku-band SAR mission of its type, it falls into the 'Explorer' mission category at the CSA. Explorer scale missions must meet a specific cost cap. An Explorer mission concept was developed by industrial partners

in Canada during to advance technological innovation and prove the scientific viability with reduced overall risk. This 'TSMM-Explorer' concept meets ECCC science requirements through dual frequency (13.5 and 17.25 GHz) Ku-band radar measurements at 500-m spatial resolution, with a 50-m spatial resolution mode across a 30 km swath available for specific regions (e.g. mountains areas) and targeted events (e.g. periods of high flood risk). An imaging swath of 250 km combined



with a duty cycle of approximately 25% meets the requirement to image all of Canada and other global snow-covered areas

every 7 days.

In the CoReH2O proposal, the dual frequencies were in the X-band at 9.6 GHz and the Ku-band at 17.2 GHz. Based on the ground-based measurements and the airborne measurements, the Ku-band at 17.2 GHz has a dynamic range from -16 dB to - 6 dB and shows good correlation with SWE. The X-band at 9.6 GHz has co-polarization dynamic range from -20 dB to -14 dB. The theoretical predictions of rough surface scattering at X-band, depending on penetration through the snow layer are

from -20 dB to -12 dB. In TSMM, the X-band (9.6 GHz) is not chosen and is replaced by a low Ku-band at 13.5 GHz.

In TSMM, the dual-frequency Ku-band are at 13.5 and 17.25 GHz. This modification from CoReH2O (9.6 GHz) to TSMM (13.5 GHz) is to have less sensitivity to the underlying rough surface scattering effect. The dual frequencies are selected to fully exploit the frequency dependence of snow volume scattering, the differential sensitivities at 13.5 and 17.25 GHz, to both SWE and snow microstructure. The dual frequencies will help address the ill-posed nature of retrieving SWE from a single

Ku-band radar measurement (see Section 4c). A potential trade-off is that sensitivity to deep snow may be reduced without the X-band measurement. Further study is needed in fine-grained mountain snowpacks to better understand the limits of sensitivity to deep snow at Ku-band. It is also possible to explore the use of cross-polarization at Ku-band for larger snow depth beyond 1 meter. Table 5 shows the comparison between the proposed CoREH2O and the TSMM currently planned. The TSMM is at a planning stage and the final specifications have not been completed.

Alongside the improved understanding of the underlying physics of Ku-band's sensitivity to snow mass, work flows for mission data have been designed for implementation at ECCC. The approach is to optimally integrate spaceborne radar measurements with state-of-the-art modelling systems. For applications which require a SWE retrieval, the radar backscatter measurements will be combined with snow property initial conditions (including snow microstructure; see Section 4) produced by an advanced version of the ECCC operational land surface model forced with short-range meteorological forecasts and

precipitation analyses. Importantly, in areas without radar coverage (e.g. due to swath gaps) or where the radar-derived SWE retrieval is highly uncertain (e.g. due to wet snow or dense forest cover), the SWE analyses will be determined by numerical outputs from the land surface model. In this way, the remote sensing information is combined with modelling to create seamless coverage both in space and time, with minimized uncertainty. The Ku-band radar measurements will also be included in ECCC prediction systems even without the use of a SWE retrieval product. For land surface data assimilation needs, the Ku-band

backscatter can be directly assimilated, again with the land surface model providing the required ancillary information. This approach is analogous to how L-band radiometer measurements from the SMOS and SMAP missions have improved soil moisture analysis at ECCC through radiance-based assimilation (Carrera et al., 2019). The resulting enhanced snow analyses will then be used to initialize environmental prediction systems at ECCC, including NWP and streamflow.

While the dual-frequency Ku-band TSMM-Explorer concept meets the requirements at ECCC for enhanced snow remote

sensing, this approach alone cannot solve all snow-related observational needs. For instance, volume scattering will be negligible under wet snow conditions, hence the need to implement the approach outlined above to combine satellite and model-derived information during snow melt. There are, however, opportunities to further exploit the baseline TSMM-

Explorer concept through the development of companion satellites. This could take the form of a similar mission concept to improve coverage and revisit (e.g. a small SAR constellation). Development of a companion receive-only satellite is more

challenging but would deliver a greater reward through the potential for InSAR-based snow depth retrievals under wet snow conditions (as previously explored with airborne Ka-band measurements by Moller et al. (2017)). This InSAR approach, as described in Section 5.3, necessitates pushing the performance envelope of the TSMM-Explorer SAR (bandwidth, increased high resolution mode imaging) and spacecraft (maintenance of a very tight baseline, hence very precise orbit control) which introduces cost and complexity to the current scope of the mission. Still, the potential benefits of an InSAR capability are

notable, so further study of these options is encouraged.

With the industrial Phase 0 now complete, the TSMM-Explorer mission has advanced into a mission pre-formulation phase at CSA. Technical readiness is being advanced through industrial investment focused on the radar antenna technologies. Scientific readiness for the mission continues to be enhanced by community-wide progress in field techniques (e.g. quantitative snow microstructure measurements; Section 3.3)), physical snow modelling (Section 4.2), data assimilation, and multi-

frequency radar analysis (Section 4.4). The analysis of tower and airborne Ku-band radar datasets in collaboration with international partners is ongoing. Future airborne data acquisition plans are under development, including through the NASA SnowEx program. Collectively, these efforts serve to advance the TSMM-Explorer mission specifically, and snow-radar science in general.

## 7 Summary and Perspectives

Terrestrial snow is a critical geophysical variable for climate, hydrology and ecology science and applications. The reduction of snow in the Northern Hemisphere has been pronounced and is particularly evident in northern high latitudes. Passive microwave remote sensing has been used in global monitoring of SWE for decades, with spatial resolutions of 25 km. For applications in climate, hydrological, and ecological processes, there need to be finer resolutions of 500 m and temporal revisits of several days to monitor SWE.

Over the last decade, X- and Ku-band radar remote sensing technologies have proven to be effective for measuring SWE. The technology readiness level has significantly advanced for satellite launch. The scattering physical models have provided understanding of the contributions of volume and surface scattering and the effects of forests. Vegetation ground measurements and snow physical models have significantly advanced to be used to couple to the RTM and data analysis. The tower measurements and the airborne campaigns haven proven to be effective in providing calibration and validations of RTM, snow

physical models, and retrieval algorithms.

Retrieval algorithms have been significantly improved, leading to much reduced RMSE. In the next few years, there are plans for more tower measurements and airborne campaigns to fill information gaps such as effects of forests covers, deep snow, various substrates beneath the snow, etc. There will be more studies on coupling snow physical models to RTM so that they





can be more effectively combined in retrieval algorithms for the seven classes of snow and with computational efficiency for

real time retrieval.

We anticipate future launches of satellite missions with Ku-band radars. The Ku-band radar satellites will be followed by more

satellites of synergistic instruments of Ku-band interferometry and tomography.

**Table of Abbreviations**

|  |  |
|---|---|
| AIEM | Advanced Integral Equation Model |
| Can-CSI | Canadian CoReH2O Snow and Ice Experiment |
| CASIX | Canadian Snow and Ice Experiment |
| CF | Coniferous Forests |
| CMIP6 | Coupled Model Intercomparison Project |
| CoReH2O | Cold Regions Hydrology Observatory |
| DMRT | Dense Medium Radiative Transfer |
| DMRT-ML | Dense Medium Radiative Transfer – Multiple Layers |
| EM | Electromagnetic |
| ESA | European Space Agency |
| FMCW | Frequency Modulated Continuous Wave |
| GNSS-R | Global Navigation Satellite System Reflectometry |
| HUT | Helsinki University of Technology |
| InSAR | Interferometric SAR |
| Lidar | Light detection and ranging |
| LUT | Look-Up Table |
| LWC | Liquid Water Content |
| MEMLS | Microwave Emission Model of Layered Snowpacks |
| MIMICS | Michigan Microwave Canopy Scattering |
| NMM3D | Numerical solutions of Maxwell's Equations in 3-D |
| NoSREx | Nordic Snow Radar Experiment |
| NWP | Numerical Weather Prediction |
| POLSCAT | Polarimetric Scatterometer |
| QCA | Quasi-Crystalline Approximation |
| SAR | Synthetic Aperture Radar |
| SMAP | Soil Moisture Active and Passive |
| SMOS | Soil Moisture and Ocean Salinity |



| SMP | Snow Micropenetrometer |
|---|---|
| SMRT | Snow Microwave Radiative Transfer |
| SSA | Specific Surface Area |
| SWE | Snow Water Equivalent |
| SWESARR | Snow Water Equivalent SAR and Radiometer |
| RTE | Radiative Transfer Equation |
| RTM | Radiative Transfer Model |
| UAVSAR | Unoccupied Aerial Vehicle Synthetic Aperture Radar |
| μ-CT | Micro computed tomography |

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
