# Peer review of "Review Article: Global Monitoring of Snow Water Equivalent using High Frequency Radar Remote Sensing"

_The Cryosphere, 2021_

## Referee Comment (RC2)

***Comment on*** **"Review Article: Global Monitoring of Snow Water Equivalent using High Frequency Radar Remote Sensing" by L. Tsang et al.**

**General Comments:**

In this manuscript the authors present a comprehensive review on methods and experimental studies for snow water equivalent (SWE) measurement by means of radar-based remote sensing techniques with the focus on radar systems operating at Ku-band and X-band frequencies. A main motivation for the review is the preparation for a future satellite mission for measuring SWE, in order to close a main gap in the observation of key parameters of the global climate system. The manuscript is generally well written, comprising many technical details. According to the layout and contents it addresses primarily remote sensing experts. Some of the references are of marginal relevance for the topic of the review, in particular several of those that are reporting on studies related to vegetation. The readability would benefit from some cutback in this respect in order to improve the focus on the main issues.

For the wider snow research and monitoring communities it may be difficult capturing distinct information on the suitability of methods and tools that are relevant for their specific application. To this end a clear assessment of the performance and constraints of current methods and tools is needed, as well as a discussion how errors and uncertainties may impact the retrieval accuracy. It is also necessary to point out that by now the retrievals are relying on a few limited test cases, making the performance of models and algorithms for general application hard to assess. Furthermore, for some of the subtopics a sound description is. For these cases (in particular Sect. 5.2 and 5.3) I recommended major shortening as they are not of direct relevance for the main topic of the manuscript. Please find below details on my concerns and recommendations for revisions.

**Specific Comments:**

*References on scientific and technical activities for the ESA Earth Explorer candidate mission CoReH2O*:
In several sections of the manuscript reference is made to the CoReH2O mission. However, the references do not proceed beyond the statues of Phase 0 (first assessment studies). The Phase 0 activities were succeeded by detailed scientific and technical feasibility studies (Phase A), results of which are summarized in the (public) Report for Mission Selection (ESA, 2012). Apart from scientific and technical details, this document reports also results of performance studies and points out critical areas and risks, issues that are still of relevance and should be addressed in the review.

*L87:* The claim for a "dramatic" advancement of radar retrieval algorithms is not based on actual evidence. There is not yet any generally applicable and widely tested algorithm available for SWE retrieval. The proposed algorithms (Section 4) apply still the basic approach proposed for the CoReH2O mission that is based on constrained minimization in which iteration is performed for two free variables, SWE and a parameter related to the volume scattering albedo (ESA, 2012). By now the retrievals are relying on - and have been optimized for - a few test cases. See also the comment on Sect. 4.1 and 4.2 below.

*L96*: Please provide a reference to studies or documents specifying the spatial and temporal requirements.

*L107*: In mountain areas the spatial variability of SWE is below 100 m. See e.g. Grünewald et al., 2010.

*L162:* "volume scattering increases with snow mass" This is a cursory statement, not accounting for other factors that affect and may dominate the volume scattering signal.

*L176, Fig. 1*: Ground-volume interactions are missing.

*L225, Fig. 3 and related text*: Structural anisotropy is a characteristic feature of natural snow packs (e.g. Leinss et al., 2020). This causes changes of the phase matrix with the incidence angle. Please explain in which way incidence angle effects related to structural anisotropy are taken into account in the models for computing radar signal propagation and phase matrix.

*L257, Fig. 4 and related text:* This example is based on a very limited data base, comprising only three points out of daily NoSRex SnowScat measurements that were acquired during four winter seasons with quite different backscatter behaviour (Lemmetyinen et al., 2014; 2016). For comprehensive evaluation of the model a wider view is needed, checking data at different incidence angles and for cases with different snow structural properties. If such an analysis is not available, current limitations in this respect need to be addressed.

*L288, Fig. 5(a):* Is there any particular reason why different rms heights are used for wet and frozen soil?

*L322, Fig. 6(a):* The sample of 5 points, 4 of which show the same moisture value, is not an adequate sample for a reliable performance estimate of the soil moisture retrieval algorithm. Fig 6(b) shows also a very small sample, not matching the needs for statistically significant performance estimates. Computations at different incidence angles and comparisons with experimental data (as available from NoSRex) would provide higher confidence. Furthermore, please specify the incidence angle and the state of the soil in Fig. 6(b). Please check the allocation of the blue and red marks in the figure caption; it seems the symbols for NoSREx and SnowEx have been mixed up.

*L347ff, impact of forests:* Here it should also be mentioned that in CoReH2O Phase A the impact of forests on radar signals of snow covered ground was studied (ESA, 2012), described in detail by Montomoli et al. (2016). The forest model selected for this study accounts for scattering of trunks, branches of different size and needles, as well as for differences in the structure of vertical layers. Effects of differences in cover fraction, tree height and biomass were analysed. Consequently, this model allows for a multifaceted description of forest properties and for estimating the impact of the forest parameters on the backscatter of snow covered forests.

*L415, Fig.9:* Reference to scattering and penetration of wheat canopies is of marginal relevance for snow studies. This should better be replaced by results from studies concerned with forest canopies.

*L425ff:* The radar penetration capability in forests depends on the density and structure of the canopy and on dielectric properties. Rather than referring to lidar observations, reference should be made to studies on radar signal penetration. Kugler et al. (2014) show for various forest canopies (including coniferous) that the X-band scattering phase centre height is located well above the ground surface and the ground scattering contribution is marginal. A reasonable extrapolation from X-band to Ku-band should be possible.

*L487:* Please provide specifications on properties of the Ku-band SAR or a reference.

*L508, Table 3, Row 3:* Dates for SnowSAR campaign Finland need to be corrected: March 2011 and winter 2011-2012.

*L524ff, Section 3.3.1, field measurements:* This section presents a specific, detailed proposal on arrangement and techniques of field measurements, in its content not directly related to the topic of the review. It should better be provided as Supplement or moved to the Appendix.

Besides, the proposed arrangement requires special tools and would not be applicable on any type of terrain. In practice a trade-off between available resources and spatial coverage is needed.

*L628ff, Sections 4.1 and 4.2 (on the retrieval problem and need for a priori information):* The example algorithms apply the same basic approach as proposed for the CoReH2O mission, based on constrained minimization in which iteration is performed for two free variables, SWE and a parameter effective grain size related to the volume scattering albedo (ESA, 2012). Whereas the CoReH2O baseline version accounts for backscatter data from four channels (X-and Ku-band co- and cross-polarization) the algorithm specified in Equ. 3 uses backscatter from two channels (co-polarized) and iterates also for two free variables: total optical thickness (related to SWE after eliminating the scattering contributions) and scattering albedo (which was used as one of the free variables in the retrieval version of the CoReH2O Phase-0 studies). A critical issue is the need for accurate a priori estimates on snowpack physical properties as input for the configuration parameters of the backscatter forward model as well as for regularization. Of main concern is the parameter for describing the scattering properties (related to microstructure) for which a priori estimates within a small error bound are needed. In the performance study for CoReH2O Level-2 products the accuracy requirements of a priori data for model configuration and regularisation were quantified (ESA, 2012). This was a limited first effort. A wider view is needed for quantifying the impact of uncertainty in a priori estimates on retrieval accuracy for the different states of the global snow cover. Though addressed here between the lines, definite numbers on a priori data requirements would be needed for full traceability.

*L666, Table 4:* The generic information on typical properties of snow types in Alaska, provided in this table, is not a suitable a-priori input on snow properties, as required for inversion model initialization and regularization. This comment refers not only to specifications such as "no data" or "variable", or the contents of the last column (e.g. quoting new and wet snow for characterizing the maritime snow class), etc.

*L715:* Huang et al. (2012) refer to scattering of rough soil surfaces and not to snow.

*L775, L776:* The publications of Lemmetyinen et al. and Zhu et al. confirm the importance of reliable a priori estimates on snow microstructure (in line with the comments above on Sections 4.1 and 4.2). In these cases site-specific approaches are used for estimating snow structural parameters. In its core these retrieval algorithms (applying constrained minimization) are the same as proposed for CoReH2O and tested in the Mission Phase A with SnowScat data (ESA, 2012) and in follow-up activities with SnowSAR data (e.g. Rott et al., 2013). I was not able to locate the publication of King et al. (2019) that is cited in L776 and addressed in L784 to L790.

*L853:* Please explain the link between the co-located ground measurements and the derived a priori information.

*L888*: Please provide a reference on the difference in saturation between the co- and cross-polarized signals. NoSRex Snowscat data show similar sensitivity in terms of SWE for co- and cross-polarized Ku-band data (e.g. Lemmetyinen et al., 2014), suggesting a similar saturation limit.

*L931ff, Section 5.2, C-band:* This contribution is problematic as it provides a biased view. It expands on the statement "the volume scattering by snow grains was believed to be small at C-band" (L935). Rather than guesswork, the knowledge on C-band radar wave interaction with snow is based on careful theoretical and experimental work over years, confirming the prevalence of low backscatter intensity for seasonal snow in mountain areas. References to such studies are needed for a balanced account. Besides, the notes on some of the papers cited

in this section are questionable. For example, Pivot et al. (2012) show little change of Radarsat sigma-0 during the main part of the snow cover season (Nov. to April) at the six test sites, intermittently even a drop. Shi et al. (2000) is not included in the list of references. The data presented by Arslan et al. (2006) show similar change in in co- and cross-polarized sigma-0. There is not any statistically significant difference in the relation between either C-VV or C-VH and SWE. Bernier et al. (1999) did not use any cross-pol data. They show that C-band sigma-0 decreases with increasing SWE due to change of the backscatter contribution from ground. The Sect. 5.2 conveys the message that snow microstructure and properties of the underlying ground are of no relevance for C-band cross-polarized backscatter, in contradiction to the detailed description of the related processes in Sect. 3. Clarifying this apparent contradiction would warrant a separate publication. Therefore Sect. 5.2 should be scaled down down to a summary with some references.

*L1007*: McGrawth et al. ( 2019) report on GPR measurement of snow depth only.

*L1027 and Fig. 17*: Please provide information on the sensor and measurement site.

*L1035, Section 5.3.3, Interferometry*: The method of repeat-pass InSAR for SWE retrieval and related experiments have been described in several publications. This section does not provide any new insights and contains some errors. Therefore it is recommended to shorten this section significantly. A short summary and some key references will do. The method exploiting the phase delay in snow traces back to Guneriussen et al. (2001), not to the references cited in L1038. Further issues: L1049: SWE measurements by means of phase delay in repeat-pass InSAR do not require a second antenna. Zero baseline is optimum; in this case there is problem of ambiguity with the topographic phase is avoided. Fig. 18a: The plot and the equation are incorrect; the geometric relations are neglected. Also, it is unclear to which measurement principle the figure refers. Probably it should indicate the measurement of snow surface height by DEM differencing which can be applied in case of surface scattering (wet snow) and requires requiring single-pass InSAR. L1072ff: The paragraph on the scattering phase centre in snow is rather speculative, references report on vegetation studies, the specific conditions of interferometric radar signal propagation in snow are not taken into account (e.g. Dall, 2007). The position of the scattering phase centre in snow volumes is highly dependent snow microstructure, obscuring possible relations with snow mass (e.g. Rott et al., 2021).

*L1150:* Please provide a reference on the direct assimilation of Ku-band backscatter intensity in snow process models. The statement here, claiming that this can be done, needs a proof.

*L1174ff, Summary and Perspectives:* This review shows that significant advancements have been achieved in the fields of radar signal propagation, snow microstructure observations and backscatter modelling. On the other hand, the retrieval algorithms are still based on the same concept developed for the CoReH2O mission that requires a priori information on snow properties within comparatively narrow error bars. Major progress has been achieved in deriving such information from various sources, however by now optimized for local retrievals and tested with few confined data sets. The wider applicability and performance need to be proven. The perspectives quoted in this section are limited to short unspecific statements. Details on plans for further development would be of interest.

*List of references:* Please check the alphabetic order.

**References**

Dall, J.: InSAR Elevation bias caused by penetration into uniform volumes, IEEE Trans. Geosc. Remote Sensing, 45(7), 2319–2324, 2007.

ESA: Report for Mission Selection: CoReH$_2$O, ESA SP-1324/2 (3 volume series), European Space Agency, Noordwijk, The Netherlands, 2012. https://earth.esa.int/eogateway/documents/20142/37627/CoReH2O-Report-for-Mission-Selection-An-Earth-Explorer-to-observe-snow-and-ice.pdf

Grünewald, T. Schirmer, M., Mott, R., and Lehning M.: Spatial and temporal variability of snow depth and ablation rates in a small mountain catchment, The Cryosphere, 4, 215-225, 2010.

Guneriussen, T., Hogda, K.A., Johnson, H., and Lauknes, I.: InSAR for estimating changes in snow water equivalent of dry snow, IEEE Trans. Geosc. Rem. Sens. 39(10), 2101-2108, 2001.

Kugler, F., Schulze, D., Hajnsek, I., Pretzsch, H., and Papathanassiou, K.: TanDEM-X Pol-InSAR performance for forest height estimation, IEEE Trans. Geosc. Rem. Sens., 52 (10), 6404 – 6421, 2014.

Leinss, S., Löwe, H., Proksch, H., and Kontu. A.: Modeling the evolution of the structural anisotropy of snow, The Cryosphere, 14, 51 – 75 2020.

Rott, H., Cline, D.W., Duguay, C., Essery, R., Etchevers, P., Macelloni, G., Hajnsek, I., Kern, M., Malnes, E., Pulliainen J., and Yueh, S.H.: CoReH$_2$O, a Candidate ESA Earth Explorer Mission for snow and ice observations, Proc. of the Earth Observation and Cryosphere Science Conference, Frascati, Italy, Nov. 2012, *ESA SP-712*, European Space Agency, Noordwijk, The Netherlands, May 2013.

Rott, H., Scheiblauer, S., Wuite, J., Krieger, L., Floricioiu, D., Rizzoli, P., Libert, L., and Nagler, T.: Penetration of interferometric radar signals in Antarctic snow, The Cryosphere, 15, 4399–4419, 2021.

---

## Author Comment (AC1)

February 1, 2022

Responses to Reviewer 1's' Comments

Global Monitoring of Snow Water Equivalent using High Frequency Radar Remote Sensing
By Leung Tsang, Michael Durand, Chris Derken etal

Reviewer comments are shown in black. Proposed responses are shown in blue.

**Reviewer Comment #1**

The paper is a very thorough review of the remote sensing methodology for assessing water equivalent in the seasonal snow pack. The primary focus is on technique which utilize X to Ku band radars. There is a detailed discussion of theory concerning scattering from the evolving snow grains, scattering from rough interfaces, the effects of vegetation and complications from snow wetness. In situ and airborne data are used to illustrate generally good agreement between theory and observations.
The paper includes discussion of different retrieval algorithms and their
limitations. Suggestions for improving retrievals by augmenting with other data sets (such as passive microwave) as well as incorporation of other radar techniques (such as tomography and InSAR) are included.
The paper concludes with a discussion of two recently proposed satellite missions. Of the two, the Canadian TSSM-Explorer is an early stage of development.

I think this is a useful, stand alone review of current progress in using radar to measure snow water equivalent. The scientific justification and the rational for high resolution radar (as opposed to solely passive microwave techniques from space) are well justified.

We thank the reviewer for their summary and assessment.

One quibble I have is the short statement at the end of section 3.1.1. There, the authors briefly mention the role of layering in the snow pack on backscatter. Given the efforts by several of the authors on the impact of layering, I am surprised that there was not more discussion of a point that could present a problem for thick snow packs and well into the winter season when layering might be expected to develop.

Regarding the role of layering itself, we agree that further work ought to be done on this topic. Indeed, two new studies were published after submission of the manuscript that shed light on this topic: Thompson and Kelly (2021a) and Thompson and Kelly (2021b) present in situ measurements of Ku-band measurements of snow, along with validated forward and inverse modeling, in prairie and tundra environments, respectively. Thompson et al. (2021a) show that for a prairie site, a 2 or 3 layer model parameterization may improve results in some cases, but that a 1 layer parameterization was advantageous in other cases. Thompson et al. (2021b) illustrates the effects of a thick wind slab layer in tundra snow, and how to constrain slab

thickness in a retrieval. We will add discussion of both of these manuscripts to the discussion at the end of section 3.1.1.

More generally, papers of this sort often set up the scientific and engineering arguments for a satellite mission.  Because TSSM seems to be moving along, it is not clear to me whether the paper is designed to influence development beyond the completed Phase  0 study.  If so, then the point might be emphasized with a discussion of recommended mission and instrument requirements.  If not, then the paper losses some of its justification in my opinion.

We agree that papers such as this are often motivated by the need to articulate the rationale for a satellite mission, but we disagree that the current forward progress of TSMM minimizes the need for this paper. First, TSMM is not yet selected for full mission implementation. Second, the paper stands on its own as a summary of the notable progress made in this field (especially over the past decade), and we believe is of great value as a review even aside from the need to motivate a satellite mission. Our rationale is that there is a lot of work on this topic, indicating significant interest across the snow community. Our aim is to provide a summary of that work with significant updating of work since 2010 (last reviewed archival journal paper in H. Rott et al. *Proceedings of IEEE* 2010).  Since 2010, there have been significant amount of work in volume  and surface scattering modelling, ground and airborne measurements, retrieval algorithms etc.  which should thus also be of interest, as no review paper has yet been published. Thus, we believe the paper has adequate justification even aside from the need to motivate TSMM or any other future snow-radar mission concepts.

---

## Author Comment (AC2)

February 1, 2022

Responses to Reviewer 2's' Comments

Global Monitoring of Snow Water Equivalent using High Frequency Radar Remote Sensing
By Leung Tsang, Michael Durand, Chris Derken etal

Reviewer comments are shown in black. Proposed responses are shown in blue.

**Reviewer Comment #2: Helmut Rott**

Note – we (the authors) have numbered the general comments in order to more easily discuss them. The comments themselves are reproduced verbatim.

**General Comments:**

1. In this manuscript the authors present a comprehensive review on methods and experimental studies for snow water equivalent (SWE) measurement by means of radar-based remote sensing techniques with the focus on radar systems operating at Ku-band and X-band frequencies. A main motivation for the review is the preparation for a future satellite mission for measuring SWE, in order to close a main gap in the observation of key parameters of the global climate system. The manuscript is generally well written, comprising many technical details. According to the layout and contents it addresses primarily remote sensing experts.

We thank Dr Rott for his assessment and his helpful comments!

2. Some of the references are of marginal relevance for the topic of the review, in particular several of those that are reporting on studies related to vegetation.

We will remove some of the marginal references. We will keep many of the references however, as our purpose in that section is to show that full wave simulations of vegetation give results that are very different from the RT method for vegetation and canopy that have been in use for decades. This is further discussed in the response to Specific Comment # 11.

3. The readability would benefit from some cutback in this respect in order to improve the focus on the main issues.

We agree, and will make several cuts throughout in connection with specific recommendations below: e.g. see our response to your Specific Comment #16, #25, and #26 below.

4. For the wider snow research and monitoring communities it may be difficult capturing distinct information on the suitability of methods and tools that are relevant for their

specific application. To this end a clear assessment of the performance and constraints of current methods and tools is needed, as well as a discussion how errors and uncertainties may impact the retrieval accuracy.

We agree. The newly published Thompson and Kelly 2021a helps to further explore sources of uncertainty. We will add a summary of this paper, including the impacts of ice lenses on retrieval in prairie snowpacks, to section 4.4.3, and a quantification of the uncertainties due to vegetation and soil. Furthermore, we will summarize some of the uncertainty propagation work done for CoReH2O, which as you point out is still relevant, and summarize it in section 4.4 as well.

5. It is also necessary to point out that by now the retrievals are relying on a few limited test cases, making the performance of models and algorithms for general application hard to assess.

We agree that data are always limited for ground and airborne measurements which challenges algorithm development. There have been significant number of airborne and tower-based radar campaigns over the past decade. During the same period, the ability to objectively measure snow microstructure has advanced significantly. We will modify the abstract, in section 4.4, and in the conclusion, to this effect. We will also add citations of the recently published retrieval study, Thompson et al. 2021a.

6. Furthermore, for some of the subtopics a sound description is. For these cases (in particular Sect. 5.2 and 5.3) I recommended major shortening as they are not of direct relevance for the main topic of the manuscript. Please find below details on my concerns and recommendations for revisions.

We will significantly shorten Sect 5.2 and 5.3.

**Specific Comments**

Note – we (the authors) have numbered the general comments in order to more easily discuss them. The comments themselves are reproduced verbatim.

**References on scientific and technical activities for the ESA Earth Explorer candidate mission CoReH2O:**

1. In several sections of the manuscript reference is made to the CoReH2O mission. However, the references do not proceed beyond the statues of Phase 0 (first assessment studies). The Phase 0 activities were succeeded by detailed scientific and technical feasibility studies (Phase A), results of which are summarized in the (public) Report for Mission Selection (ESA, 2012). Apart from scientific and technical details, this

document reports also results of performance studies and points out critical areas and risks, issues that are still of relevance and should be addressed in the review.

We agree. We will add citation and discussion of the Report for Mission Selection throughout the manuscript.

2. *L87:* The claim for a "dramatic" advancement of radar retrieval algorithms is not based on actual evidence. There is not yet any generally applicable and widely tested algorithm available for SWE retrieval. The proposed algorithms (Section 4) apply still the basic approach proposed for the CoReH2O mission that is based on constrained minimization in which iteration is performed for two free variables, SWE and a parameter related to the volume scattering albedo (ESA, 2012). By now the retrievals are relying on - and have been optimized for - a few test cases. See also the comment on Sect. 4.1 and 4.2 below.

We agree that the proposed algorithms have built off the CoReH2O approach. There are similarities and differences in the cost function and a priori information as pointed out in our replies to section 4.2 and section 4.3. . In the CoREH2O cost function, there are many parameters that require a priori estimates.  We will clarify this in the revised manuscript. The algorithm in this review paper is more closely related to the NASA SMAP radar algorithm, that by using regression to electromagnetic model simulations over a wide range of parameters, the number of parameters are significantly reduced.  The strategy of this approach is to reduce the burden of a priori estimates of parameters for every scene.  Advances since the time of the 2012 report are in validating retrieval algorithms with multiple experimental datasets in peer-reviewed publications, and in reducing the dependence on a priori datasets, which we further describe in our response to specific comment #18, below.

That said, Dr Rott raises a good point that algorithms are still maturing. We will revise the manuscript at line 87 to say that the algorithms and forward models have advanced significantly, and qualify this with our reporting on multiple published studies from multiple groups with variations on the cost-function approach first proposed for CoReH2O.

3. *L96*: Please provide a reference to studies or documents specifying the spatial and temporal requirements.

The references are provided in the remaining part of the section: lines 97-154. We will add references to the CoReH2O Report for Mission Selection, as well. We will also cite the IGARSS paper Derksen etal 2021 paper concerning the requirements which are driving TSMM.

4. *L107*: In mountain areas the spatial variability of SWE is below 100 m. See e.g. Grünewald et al., 2010.

We will revise the manuscript to read "the coarse spatial resolution of existing products … is incompatible with the scale of SWE variability (<100 of meters)."

5. *L162:* ""volume scattering increases with snow mass" This is a cursory statement, not accounting for other factors that affect and may dominate the volume scattering signal.

We will revise the manuscript to read "While volume scattering is controlled by several parts of the landscape, volume scattering increases with snow mass"

6. *L176, Fig. 1*: Ground-volume interactions are missing.

We will clarify this. We provide the simple single scattering formula to help the non-expert reader. In the actual electromagnetic simulations, we do not use the simple formula. The full DMRT with boundary conditions are solved. Thus the solutions including multiple scattering are calculated. These include all orders of multiple volume scattering , ground-volume interactions and volume-snow surface interactions . We will point this out in the revised manuscript that full multiple scattering within DMRT are used to calculate the look up table .

7. *L225, Fig. 3 and related text*: Structural anisotropy is a characteristic feature of natural snow packs (e.g. Leinss et al., 2020). This causes changes of the phase matrix with the incidence angle. Please explain in which way incidence angle effects related to structural anisotropy are in the models for computing radar signal propagation and phase matrix.

We will revise to indicate that anisotropy has been included in the bicontinuous model (Tan etal 2016). There are two models in the bicontinuous model: isotropic correlation functions and anisotropic correlations functions (Tan etal 2016) . Both have been developed and simulations performed. In the retrieval, only the isotropic correlation functions versions have been used.

8. *L257, Fig. 4 and related text:* This example is based on a very limited data base, comprising only three points out of daily NoSRex SnowScat measurements that were acquired during four winter seasons with quite different backscatter behaviour (Lemmetyinen et al., 2014; 2016). For comprehensive evaluation of the model a wider view is needed, checking data at different incidence angles and for cases with different snow structural properties. If such an analysis is not available, current limitations in this respect need to be addressed.

This figure is shown only as an illustration, reproduced from Tan et al. 2015. Indeed, several years of NoSRex data were analyzed and the results of retrieval performance of the several year were illustrated in the paper by Zhu et al. 2019. We will point this out in the manuscript.

*L288, Fig. 5(a):* Is there any particular reason why different rms heights are used for wet and frozen soil?

The retrieved rms height is a single value of 1.9cm.   This is the rational of the Kim et al. algorithm that soil moisture changes but not rms height.  Wet and frozen soils have the same rms height of 1.9 cm in figure 5a. We will clarify this point in the manuscript.

9.  *L322, Fig. 6(a):* The sample of 5 points, 4 of which show the same moisture value, is not an adequate sample for a reliable performance estimate of the soil moisture retrieval algorithm. Fig 6(b) shows also a very small sample, not matching the needs for statistically significant performance estimates. Computations at different incidence angles and comparisons with experimental data (as available from NoSRex) would provide higher confidence. Furthermore, please specify the incidence angle and the state of the soil in Fig. 6(b). Please check the allocation of the blue and red marks in the figure caption; it seems the symbols for NoSREx and SnowEx have been mixed up.

Because of the dielectric contrasts, the contributions of rough surface scattering are from the snow-soil rough interface and not from the air/snow interface.   The air/snow interface have stronger scattering contribution for wet snow which is outside the domain of X band and Ku band volume scattering approach.  Significant advances of rough soil surface scattering at X and Ku band have been made by extending the previous limit of kh=3 to kh=15 where k is the wavenumber and h is the rms height.   With the availability of L band, and C band data, and full wave simulations of rough soil surface scattering, the retrievals of rms heights of the soil surfaces and the soil surface scattering component can be determined accurately.   JPL recently has released a rms height global map of soil surfaces based on the 5 months of SMAP L band radar data which further helps the estimation of rough soil surface scattering at X and Ku band using full wave simulations.  With the L band rough surface scattering algorithm tested and validated globally, figure 6 is just applying the algorithm to show that the algorithm also works for soil surface below snow by taking into  account  the changes of incidence angle  when the microwave signal enters snow from the air region. The soil moisture retrieved for the site in figure 6a also agrees with the in-situ soil measurements.

We agree that the sample size is quite small, and will add caveats on this in the manuscript.

We thank Dr. Rott for pointing out in figure 6b that the symbols SnowEx and NoSREx were mistakenly switched. We will fix the figure.

10. *L347ff, impact of forests:* Here it should also be mentioned that in CoReH2O Phase A the impact of forests on radar signals of snow covered ground was studied (ESA, 2012), described in detail by Montomoli et al. (2016). The forest model selected for this study accounts for scattering of trunks, branches of different size and needles, as well as for differences in the structure of vertical layers. Effects of differences in cover fraction, tree height and biomass were analysed. Consequently, this model allows for a multifaceted description of forest properties and for estimating the impact of the forest parameters on the backscatter of snow covered forests.

We will add further discussion of Montomoli et al. (2016) and Kugler et al. (2014). We agree that Montomoli et al. 2016 calculated the different contributions from trunks, branches and leaves. However, it is to be noted that the work is based fundamentally on the RT model which has been in use since the 1980s. Recently, since 2017, instead of using the RT models, we have used full wave simulations based on a hybrid method of combining wave multiple scattering theory (W-MST) and commercial software (HFSS and FEKO) of solving single scatterers. Results have been computed for L, S, and C Bands.  The results of full wave simulations, so far,  show large differences from that of the RT model giving much more penetration than predicted by the RT model.  The reasons are that the RT assumptions of uniform distribution of scatterers and the ignorance of gaps are poor assumptions and are not applicable to vegetation/trees. Basically, these results indicate that the RT approach, although applied for decades, is not valid when applicated to vegetation. We hesitate to extrapolate the conclusions for trees to X Band and Ku Band.  However, preliminary results running full wave simulations of needle leaves at X band and Ku band also show significant differences from the RT model. Thus, in the next few years, there will be extensive full wave simulations and new measurements to study the effects of trees and forests at X band and Ku band.

11. *L415, Fig.9:* Reference to scattering and penetration of wheat canopies is of marginal relevance for snow studies. This should better be replaced by results from studies concerned with forest canopies.

We will reduce the number of references and examples. As noted in our response to the previous comment, the key point of the wheat example is show that the RT approach for vegetation and forests which has been used for decades, is not valid.  We will further clarify the rationale for this section.

12. *L425ff:* The radar penetration capability in forests depends on the density and structure of the canopy and on dielectric properties. Rather than referring to lidar observations, reference should be made to studies on radar signal penetration. Kugler et al. (2014) show for various forest canopies (including coniferous) that the X-band scattering phase centre height is located well above the ground surface and the ground scattering contribution is marginal. A reasonable extrapolation from X-band to Ku-band should be possible.

We will clarify in the manuscript. We do believe that the point we are making with the reference to LiDAR still stands in the context of the section: please see our longer response to specific comment 11. We will include a reference to Kugler et al. in this section, and discuss it in context of the newer full wave simulations and how they may differ from past results.

We will also add discussion of a recent paper that sheds additional light on this topic, Thompson and Kelly (2019), which we will also cite.

13. *L487:* Please provide specifications on properties of the Ku-band SAR or a reference.

We will added references to each instrument listed in the section.

14. *L508, Table 3, Row 3:* Dates for SnowSAR campaign Finland need to be corrected: March 2011 and winter 2011-2012.

We will fix.

15. *L524ff, Section 3.3.1, field measurements:* This section presents a specific, detailed proposal on arrangement and techniques of field measurements, in its content not directly related to the topic of the review. It should better be provided as Supplement or moved to the Appendix.

This is a good point. We will move much of this material to an Appendix.

16. Besides, the proposed arrangement requires special tools and would not be applicable on any type of terrain. In practice a trade-off between available resources and spatial coverage is needed.

Excellent point. We will add a short discussion of these tradeoffs.

17. *L628ff, Sections 4.1 and 4.2 (on the retrieval problem and need for a priori information):* The example algorithms apply the same basic approach as proposed for the CoReH2O mission, based on constrained minimization in which iteration is performed for two free variables, SWE and a parameter effective grain size related to the volume scattering albedo (ESA, 2012). Whereas the CoReH2O baseline version accounts for backscatter data from four channels (X-and Ku-band co- and cross-polarization) the algorithm specified in Equ. 3 uses backscatter from two channels (co-polarized) and iterates also for two free variables: total optical thickness (related to SWE after eliminating the scattering contributions) and scattering albedo (which was used as one of the free variables in the retrieval version of the CoReH2O Phase-0 studies). A critical issue is the need for accurate a priori estimates on snowpack physical properties as input for the configuration parameters of the backscatter forward model as well as for regularization. Of main concern is the parameter for describing the scattering properties (related to microstructure) for which a priori estimates within a small error bound are needed. In the performance study for CoReH2O Level-2 products the accuracy requirements of a priori data for model configuration and regularisation were quantified (ESA, 2012). This was a limited first effort. A wider view is needed for quantifying the impact of uncertainty in a priori estimates on retrieval accuracy for the different states of the global snow cover. Though addressed here between the lines, definite numbers on a priori data requirements would be needed for full traceability.

Since the 2012 CoREH2O report, there have been significant airborne and ground campaigns providing much more data than was available prior to CoReH2O: see Tables 2 and 3. The algorithm of Zhu et al. 2018 builds upon the CoREH2O algorithm, but does add significant

innovation. The strategy is different from the CoREH2O algorithm where the cost function contains more parameters than the number of measurements. In CoREH2O, because the number of parameters exceed the number of measurements, there will be families of solutions . Thus, the choice of a priori estimate for every scene becomes critical for the CoReH2O algorithm. The present algorithm consists of (i) subtracting the rough surface scattering, which are significant in many cases, (ii) the remainder, volume scattering components of the measurements, (X and Ku Band co-polarization) are used to retrieve successfully the two parameters of SWE and albedo. Extensive forward simulations are performed and combined with regression to reduce to only two parameters, the SWE and effective albedo. The approach of reducing the number of parameters through regression is similar to the SMAP radar algorithm.

Because the number of parameters is equal to the number of measurements in the Zhu etal algorithm, there are not families of solutions as was the case with the CoREH2O algorithm. We will specify in the revised paper that the algorithm has binary choices of initial guesses of scattering albedo. Choosing one of the two will give a unique solution. We will include results of plotting 3D plot of the backscattering sigma 0 as a function of the two variables of SWE and scattering albedo. the This strategy of reducing the number of parameters through regression analysis, is similar that of the SMAP radar algorithm. It eliminates the burden of a priori estimate of a list of parameters for every scene

Significant rough soil surface scattering has been advanced for rough surface effects at X and Ku band as well as volume scattering models. Because of the dielectric contrasts, the contributions of rough surface scattering are largely from the snow-soil rough interface and much smaller from the air/snow interface. With the availability of L band, and C band data, and full wave simulations of rough soil surface scattering, the retrieval of rms heights of the soil surfaces and the soil surface scattering component can be determined accurately. JPL recently has released a rms height map of soil surfaces globally based on the 5 months of SMAP L band radar data which further helps the estimation of rough soil surface scattering at X and Ku band using full wave simulations.

Furthermore, Thompson and Kelly (2021a) has recently been published, which performs retrieval on a new ground-based dataset, using a similar but innovative retrieval algorithm. Multiple groups showing successful retrievals in multiple locations in the peer-reviewed literature is indeed a significant advance.

We will revise the manuscript to better highlight our point that there have been significant advances in this field.

We further clarify the role of a priori data in our responses to Specific Comment #21 below, which is on a similar topic.

18. *L666, Table 4:* The generic information on typical properties of snow types in Alaska, provided in this table, is not a suitable a-priori input on snow properties, as required for

inversion model initialization and regularization. This comment refers not only to specifications such as "no data" or "variable", or the contents of the last column (e.g. quoting new and wet snow for characterizing the maritime snow class), etc.

In fact, previous work in passive microwave remote sensing has developed a set of priors from this paper (Pan et al. 2017). We will add a mention of this in the manuscript, and discuss needed steps to go from the table to workable priors.

19. *L715:* Huang et al. (2012) refer to scattering of rough soil surfaces and not to snow.

The electromagnetic analysis in Huang's paper (2012) , although named soil for SMAP,  was carried out for general rough surface interface for relative dielectric constants  on both sides of the rough surface . The look up table  results are  applicable to both air/soil and sow/soil by using the appropriate ratio of dielectric constant.

20. *L775, L776:* The publications of Lemmetyinen et al. and Zhu et al. confirm the importance of reliable a priori estimates on snow microstructure (in line with the comments above on Sections 4.1 and 4.2). In these cases, site-specific approaches are used for estimating snow structural parameters. In its core these retrieval algorithms (applying constrained minimization) are the same as proposed for CoReH2O and tested in the Mission Phase A with SnowScat data (ESA, 2012) and in follow-up activities with SnowSAR data (e.g. Rott et al., 2013). I was not able to locate the publication of King et al. (2019) that is cited in L776 and addressed in L784 to L790.

Recent algorithm work has focused on how to reduce the importance of a priori information.

As described in response to comment 16, regression analysis were applied to volume scattering simulations with varying grain sizes and snow densities to simplify the model dependence to only two parameters:  scattering albedo and SWE. In the cost function, there are only two parameters and with two measurements of X and Ku Band co-pol.  With two equations (though non-linear), and two unknowns, we will include results in the revised manuscript that there are only two solutions, and not a family of solutions. The final choice of one of the two solutions is from one of the two initial guesses of the albedo either 0.35 or 0.65.  Thus, the algorithm in Zhu et al does not require  a priori estimate other than the binary choice of initial guess of albedo. Thus, for the present algorithm, accurate prior estimate of parameters is not a critical issue. The present algorithm is similar to the strategy for the SMAP radar algorithm in which the models are simplified to contain much fewer parameters through regression analysis.  As explained earlier, this regression strategy reduces the burden of a proiri estimates of parameters for every scene.

We are presently extending this concept to the algorithm for a two layered snow. By carrying regression analysis to a wide range of parameters of layering , densities and grain sizes, the goal is  reduce model dependence to 4 parameters. The strategy is to have the number of

parameters to match the number of measurements, which will mitigate the critical issue of a priori estimates for every scene.

The recently published Rutter et al. (2019) and Thompson et al. (2021a) also discuss the role of a priori information; we will add further discussion of these in this section.

We will add discussion of these points into the manuscript.

We will remove citation of King et al., 2019.

21. *L853:* Please explain the link between the co-located ground measurements and the derived a priori information.

We will add further clarification in the manuscript.

22. *L888*: Please provide a reference on the difference in saturation between the co- and cross-polarized signals. NoSRex Snowscat data show similar sensitivity in terms of SWE for co- and cross-polarized Ku-band data (e.g. Lemmetyinen et al., 2014), suggesting a similar saturation limit.

We will further clarify this point in the manuscript.

23. *L931ff, Section 5.2, C-band:* This contribution is problematic as it provides a biased view. It expands on the statement "the volume scattering by snow grains was believed to be small at C-band" (L935). Rather than guesswork, the knowledge on C-band radar wave interaction with snow is based on careful theoretical and experimental work over years, confirming the prevalence of low backscatter intensity for seasonal snow in mountain areas. References to such studies are needed for a balanced account. Besides, the notes on some of the papers cited in this section are questionable. For example, Pivot et al. (2012) show little change of Radarsat sigma-0 during the main part of the snow cover season (Nov. to April) at the six test sites, intermittently even a drop. Shi et al. (2000) is not included in the list of references. The data presented by Arslan et al. (2006) show similar change in in co- and cross-polarized sigma-0. There is not any statistically significant difference in the relation between either C-VV or C-VH and SWE. Bernier et al. (1999) did not use any cross-pol data. They show that C-band sigma-0 decreases with increasing SWE due to change of the backscatter contribution from ground. The Sect. 5.2 conveys the message that snow microstructure and properties of the underlying ground are of no relevance for C-band cross-polarized backscatter, in contradiction to the detailed description of the related processes in Sect. 3. Clarifying this apparent contradiction would warrant a separate publication. Therefore Sect. 5.2 should be scaled down to a summary with some references.

We agree with many of Dr Rott's points. However, this manuscript (which is devoted to high frequency radar remote sensing of SWE) cannot resolve all of the issues raised by Dr Rott, which would indeed require a separate publication as noted in the comment. Instead, our purpose here is merely to point to the fact that there are publications on C-band estimation of snow depth in the literature, and to discuss that C-band (with sensitivity in deep snow) is thus potentially highly synergistic with the higher frequency approaches described in this manuscript (for which the radar scattering signal with saturate in deep snow) . Thus, we have cut back discussion of technical C-band issues, and instead keep the result presented in the manuscript that demonstrates how C-band may work better for deep snow than for shallow snow.

24. *L1007*: McGrawth et al. ( 2019) report on GPR measurement of snow depth only.

That is correct, and is consistent with the paragraph containing this citation.

25. *L1027 and Fig. 17*: Please provide information on the sensor and measurement site.

We will  shorten this paragraph, and combine all references into one sentence about the use of ground based radar at nadir for depth and SWE.

26. *L1035, Section 5.3.3, Interferometry*: The method of repeat-pass InSAR for SWE retrieval and related experiments have been described in several publications. This section does not provide any new insights and contains some errors. Therefore it is recommended to shorten this section significantly. A short summary and some key references will do. The method exploiting the phase delay in snow traces back to Guneriussen et al. (2001), not to the references cited in L1038. Further issues: L1049: SWE measurements by means of phase delay in repeat-pass InSAR do not require a second antenna. Zero baseline is optimum; in this case there is problem of ambiguity with the topographic phase is avoided. Fig. 18a: The plot and the equation are incorrect; the geometric relations are neglected. Also, it is unclear to which measurement principle the figure refers. Probably it should indicate the measurement of snow surface height by DEM differencing which can be applied in case of surface scattering (wet snow) and requires requiring single-pass InSAR. L1072ff: The paragraph on the scattering phase centre in snow is rather speculative, references report on vegetation studies, the specific conditions of interferometric radar signal propagation in snow are not taken into account (e.g. Dall, 2007). The position of the scattering phase centre in snow volumes is highly dependent snow microstructure, obscuring possible relations with snow mass (e.g. Rott et al., 2021).

Measurement site and sensor frequencies were added. Sections 5.3.2 and 5.3.3 were significantly shortened as requested, including removing Fig 17.

The interferometery section has been shortened and missing references added.  We have made it more clear in the description of single-pass (for snow surface mapping) and repeat pass (for

depth/SWE change) InSAR. Information that can be found in other references and texts (e.g. the equation for InSAR correlation) has been removed to shorten this section.

27. *L1150:* Please provide a reference on the direct assimilation of Ku-band backscatter intensity in snow process models. The statement here, claiming that this can be done, needs a proof.

There are two main elements to enable radiance assimilation of Ku-band backscatter in environmental prediction systems: (1) implementation of a forward radar model and (2) coupling with a snow physical model to provide the required inputs (most importantly, snow microstructure).
The reviewer is correct in that further work is required to fully implement Ku-band backscatter assimilation, but the SMRT modeling framework and recent developments in snow physical modeling have supported the advancement of science readiness in both areas. Critically, this will be further advanced through the TSMM mission simulator, which will be under development soon. We will revise the text in this paragraph and add the relevant citations to clarify the current status of direct assimilation.

The study of Bateni et al. 2015 illustrates a proof-of-concept for backscatter assimilation using in situ observations; we will cite it here.

28. *L1174ff, Summary and Perspectives:* This review shows that significant advancements have been achieved in the fields of radar signal propagation, snow microstructure observations and backscatter modelling. On the other hand, the retrieval algorithms are still based on the same concept developed for the CoReH2O mission that requires a priori information on snow properties within comparatively narrow error bars. Major progress has been achieved in deriving such information from various sources, however by now optimized for local retrievals and tested with few confined data sets. The wider applicability and performance need to be proven. The perspectives quoted in this section are limited to short unspecific statements. Details on plans for further development would be of interest.

We have revised the manuscript accordingly.

29. *List of references:* Please check the alphabetic order.

We have checked and fixed references that were out of order.

**References**

Note that this includes all references in the response, including those reviewers introduce in their comments, and those that we introduce in our response.

Dall, J.: InSAR Elevation bias caused by penetration into uniform volumes, IEEE Trans. Geosc. Remote Sensing, 45(7), 2319–2324, 2007.

Derksen, C., J. King, S. Belair, C. Garnaud, V. Vionnet, V Fortin, J. Lemmetyinen, Y. Crevier, P. Plourde, B. Lawrence, H. van Mierlo, G. Burbidge, and P. Siqueira. 2021. Development of the Terrestrial Snow Mass Mission. *International Geoscience and Remote Sensing Symposium*, July, 2021.

ESA: Report for Mission Selection: CoReH2O, ESA SP-1324/2 (3 volume series), European Space Agency, Noordwijk, The Netherlands, 2012.

https://earth.esa.int/eogateway/documents/20142/37627/CoReH2O-Report-for-Mission-Selection-An-Earth-Explorer-to-observe-snow-and-ice.pdf

Grünewald, T. Schirmer, M., Mott, R., and Lehning M.: Spatial and temporal variability of snow depth and ablation rates in a small mountain catchment, The Cryosphere, 4, 215-225, 2010.

Guneriussen, T., Hogda, K.A., Johnson, H., and Lauknes, I.: InSAR for estimating changes in snow water equivalent of dry snow, IEEE Trans. Geosc. Rem. Sens. 39(10), 2101-2108, 2001.

Kugler, F., Schulze, D., Hajnsek, I., Pretzsch, H., and Papathanassiou, K.: TanDEM-X Pol-InSAR performance for forest height estimation, IEEE Trans. Geosc. Rem. Sens., 52 (10), 6404 –_6421, 2014.

Leinss, S., Löwe, H., Proksch, H., and Kontu. A.: Modeling the evolution of the structural anisotropy of snow, The Cryosphere, 14, 51 –75 2020.

Pan, J., Durand, M. T., Jagt, B. J. V., & Liu, D. (2017). Application of a Markov Chain Monte Carlo algorithm for snow water equivalent retrieval from passive microwave measurements. *REMOTE SENSING OF ENVIRONMENT*, *192*, 150 165. https://doi.org/10.1016/j.rse.2017.02.006

Rott, H., Cline, D.W., Duguay, C., Essery, R., Etchevers, P., Macelloni, G., Hajnsek, I., Kern, M., Malnes, E., Pulliainen J., and Yueh, S.H.: CoReH2O, a Candidate ESA Earth Explorer Mission for snow and ice observations, Proc. of the Earth Observation and Cryosphere Science Conference, Frascati, Italy, Nov. 2012, *ESA SP-712*, European Space Agency, Noordwijk, The Netherlands, May 2013.

Rott, H., Scheiblauer, S., Wuite, J., Krieger, L., Floricioiu, D., Rizzoli, P., Libert, L., and Nagler, T.: Penetration of interferometric radar signals in Antarctic snow, The Cryosphere, 15, 4399–4419, 2021.

Tan S. , C. Xiong, X. Xu and L. Tsang " Uniaxial Effective Permittivity of Anisotropic Bicontinuous Random Media Using NMM3D", IEEE Geoscience and Remote Sensing Letters, 2016

Thompson, A, R.E.J. Kelly (2019) Observations of coniferous forest at 9.6 and 17.2 GHz: Implications for SWE retrievals, *Remote Sensing*, 11(6), doi:10.3390/rs11010006

Thompson, A. and R.E.J. Kelly (2021a) Radar retrieval of snow water equivalent for mid-latitude agricultural sites, *Canadian Journal of Remote Sensing*. 47(1): 119-142. https://doi.org/10.1080/07038992.2021.1898938.

Thompson, A. and R.E.J. Kelly (2021b) Estimating wind slab thickness in a tundra snowpack, *Remote Sensing Letters*. 12:11, 1123-1135. https://doi.org/10.1080/2150704X.2021.1961174.

---

## Author Response (AR2)

**Editor's Remarks**

Dear Authors,
thank you again for the revisions of your manuscript. I am glad that the reviewer agrees with your changes, and has provided more suggestions for further improvements and clarifications. Please consider them carefully in your next revisions.
Looking forward to receiving your manuscript, best regards
Christian Haas

**Review by Helmut Rott**

General Comment:
I wish to thank the authors for their efforts in revising the manuscript and preparing the detailed response to the reviews. The revised version thoroughly addresses and clarifies issues raised in the review. In particular the revisions and material added to the sections on radar signal interaction, the retrieval approach and on the test cases, as well as related discussions are essential contributions for comprehensive understanding and are enhancing the impact of the topic addressed in the article. However, there are a still some issues that need to be checked and clarified, as addressed below.

1. Line 362ff: The RMSE value and correlation coefficient are lacking statistical significance, being based on a very limited sample with only two different soil moisture values (Fig. 6a).

   We have removed the RMSE value and correlation coefficient.

2. Fig. 9 and related text: Rather than an example for wheat (which is usually harvested many weeks before the first snowfall) the readership of a review paper on snow remote sensing techniques would expect reference or discussion on the impact of vegetation as prevalent during the snow cover season.

   The example of wheat is shown to illustrate that RTE/DBA underestimates the transmission through vegetation and has a higher frequency dependence when compared to the hybrid method as shown in Gu et al., 2021, 2022. There are not many new published results using NMM3D for different types of vegetation to be shown in this review paper. However, there has been tremendous improvement recently in computational EM efficiency and we expect new NMM3D results in near future.

3. L552 and Fig 10: Only a short statement refers to this figure, lacking a message on the observed SnowSAR backscatter signatures. For example, as follow-up to the extensive discussion on the impact of forest on the backscatter signals and the sensitivity to SWE,

information on differences in sigma-0 between open land and forested areas would be of interest for the reader. The add-on of forest cover map along the track (e.g. forest density, as shown in the Appendix to ESA 2012), or an outline of the forested areas, would be useful.

In the revised paper, it is now stated on page 20: "The differences between open area and forested area has been addressed and illustrated in Montomoli et al., 2016 paper using the models of classical radiative transfer. Results of full wave simulations are currently being studied."

4. Fig, 15b: This figure was replaced by a less informative one (showing only binary scattering albedo values) whereas the original one shows observed and simulated values (which is of interest).

The method to compute the "observed" single-scattering albedo is quite complex. After considering, we removed the observed values from the graph to avoid the need for a lengthy discussion of how they were computed. Thus, we have kept this graphic as is, even though it is possibly less informative, as we are more concerned with confusing readers than the information contained in this graph.

5. Fig. 16 and Fig. 17: The data base and message of these figures, which were not shown in the original version of the paper, are unclear. Besides, in Fig. 16 the assignment of the two point clouds at each frequency, displayed with the same symbol, is not obvious.

The purpose of this figure is to visually illustrate that the backscatter is dependent on two variables, SWE and single-scattering albedo. However, we agree that the purpose of the figures was perhaps not as clearly explained as it might have been.

We have revised this by removing one frequency, and producing plots only at Ku-band, and doing some analytical work to show that a linear fit based on both SWE and single-scattering albedo has less residual error than fitting on SWE alone. This helps visually and quantitatively illustrate the point that Ku is dependent on both SWE and single-scattering albedo.

6. Table 4: Please provide also the mean SWE values for each data set, in addition to the SWE range.

We have updated Table 4 with mean SWE values.

7. L1040: Please provide a reference on observed difference in saturation between the co- and cross-polarized signals, and specify the Ku-band frequency, incidence angle and snow type for the 300 mm saturation limited quoted here. Depending on microstructure, the saturation limit may vary by one order of magnitude. NoSRex

Snowscat data show similar sensitivity in terms of SWE for co- and cross-polarized Ku-band data (e.g. Lemmetyinen et al., 2014), suggesting a similar saturation limit.

It is now stated on page 40: "The Ku band cross polarization saturation can be shown using bi-continuous DMRT. The data for deep snow cross polarization at Ku band is quite limited."

8. Section 5.2, C-band SAR: The revision of this section does not adequately address the issues mentioned in the comments to the first version of the paper. As motivation for including this section it is mentioned " that some recent studies have demonstrated the possibility of snow depth estimation for deep snow using C-band radar …". The material and reference presented in this section suggest that the snow microstructure does not play a role for deducing the depth of deep snow from C-band backscatter data. A balanced account, as to be expected for a review paper, should present a wider view, referring also to publications on the impact of snow microstructure. An in-depth study on this respect is reported by Naderpour et al. (2022), showing a dense time series on L- to K-band backscatter (including C-band and Ku-band) for Alpine snow and discussing the impact of microstructure on the observed signatures and SWE.
We agree that snow microstructure likely plays a role in the response of the C-band cross-polarized radar signal to SWE, and did not intend to imply otherwise. We have added a statement clarifying this at line X:

"Thus, we would expect that snow microstructure plays a role in governing C-band response to SWE."

Naderpour et al. (2022) uses ESA's WBSCAT instrument, which indeed has cross-pol measurement capability. However, Naderpour et al. (2022) analyze only the co-pol signals. We have added a citation of Naderpour et al.:

"This hypothesized mechanism may explain why cross-polarized C-band radar is sensitive to SWE (Lievens et al., 2019), while co-pol is relatively insensitive to SWE (Naderpour et al., 2022). Further field-based studies of the investigations of the effects of microstructure on C-band radar signals are ongoing."

9. Fig. 20A: Please check the equation for computing the height and the geometric properties of an interferometric baseline. The height above a reference surface is proportional to the wavelength, the phase difference, and the radar range times the sin of the incidence angle and inversely proportional to the perpendicular baseline. Besides, it is not obvious that this figure is needed, considering that similar graphics have been shown in many publications and textbooks over the years.

Good catch. We have revised the equation. Thank you.

10. L1203: Radarsat and Envisat (a multi-sensor mission) were not the first C-band SAR mission, ERS was years ahead.

   It is true that ERS was launched before Radarsat-1 and Envisat. We have changed the statement.

11. L2010: CryoSat-2 has a Ku-band sensor featuring a SAR mode and a SAR Interferomeric (SARIn) mode.

   We have removed the statement on page 46 "To date, however, there have been no SAR missions at Ku-band."

12. L1238 to 1241: " Based on the ground-based measurements and the airborne measurements, the Ku-band at 17.2 GHz has a dynamic range from -16 dB to -6 dB " , "The X-band at 9.6 GHz has co-polarization dynamic range from -20 dB to -14dB". I wonder where these numbers come from and to which snow type, background medium and incidence angle they refer. For example, at the snow-covered Alpine valley and the tundra site the SnowSAR data show at 40 deg. inc. angle KuVV sigma-0 values up to -4 dB, and X-VV up to -6 dB.

   This is a fair point. We have deleted these sentences.

13. L1239: " … Ku-band 17.2 GHz …. shows good correlation with SWE ". This is a simplified statement, can be omitted. For deep, fine-grained snow the sensitivity is rather low, as for example evident from the SnowSAR data of the Alpine tundra site.

   We have removed this statement.

14. L1247 -L1249: Please provide a reference to experimental data on the sensitivity of Ku-band cross-pol sigma-0 for snow depth below one meter.

   We have removed this statement.

15. L1284: " ….seasonal snow melt is a commodity of the utmost importance for human health and well-being, supports nearly all sectors of the economy …" This seems to be slightly exaggerated.

   We have revised "nearly all sectors" to "many sectors", and have added three citations that support the point.

   Naderpour, R., Schwank, M., Houtz, D., Werner, C. and Mätzler, C. 2022: Wideband backscattering from Alpine snow cover: A full-season study, IEEE Trans. Geosc. Rem. Sens., vol. 60, pp. 1-15, 2022, Art no. 4302215, doi: 10.1109/TGRS.2021.3112772, (date of publication October 5, 2021; date of current version January 21, 2022).

---

## Author Response (AR3)

**Editor's Remarks**

Dear Authors,

Thank you for the revisions of your manuscript, which I am happy to accept! I have only one minor remaining remark, and that is that I would be glad if you could address one of the reviewer's comments related to Figure 9 (comment 2). Please include the justification for keeping the wheat example in your text as well.

Thanks a lot

Christian Haas

Comment 2 from reviewer:

Fig. 9 and related text: Rather than an example for wheat (which is usually harvested many weeks before the first snowfall) the readership of a review paper on snow remote sensing techniques would expect reference or discussion on the impact of vegetation as prevalent during the snow cover season.

In line 488 (page 18), we have added the justification for keeping the wheat example as: "The example of wheat is shown to illustrate that RTE/DBA underestimates the transmission through vegetation and has a higher frequency dependence when compared to the hybrid method as shown in Gu et al., 2021, 2022. There are not many new published results using NMM3D for different types of vegetation to be shown in this review paper. However, there has been tremendous improvement recently in computational EM efficiency and we expect new NMM3D results in near future."